EMBO
Molecular Medicine

# Repression of apelin Furin cleavage sites provides antimetastatic strategy in colorectal cancer

Béatrice Demoures[1], Fabienne Soulet[1], Jean Descarpentrie[1], Isabel Galeano-Otero [1], José Sanchez Collado[1], Maria Casado[1,2], Tarik Smani [3], Alvaro González [1], Isabel Alves[4], Fabrice Lalloué [5], Bernard Masri [6], Estelle Rascol[4], Jean-William Dupuy [7,8], Cyril Dourthe[1,8], Frédéric Saltel [1,8], Anne-Aurélie Raymond [1,8], Iker Badiola[2], Serge Evrard[1,9], Bruno Villoutreix [10], Simon Pernot[1,9], Géraldine Siegfried [1✉] & Abdel-Majid Khatib [1,9✉]

## Abstract

**The adipokine apelin has been directly implicated in various physiological processes during embryogenesis and human cancers. Nevertheless, the importance of the conversion of its precursor proapelin to mature apelin in tumorigenesis remains unknown. In this study, we identify Furin as the cellular proprotein convertase responsible for proapelin cleavage. We explore the therapeutic potential of targeting proapelin cleavage sites in metastatic colorectal cancer by introducing apelin-dm, a modified variant resulting from alteration in proapelin cleavage sites. Apelin-dm demonstrates efficacy in inhibiting tumor growth, promoting cell death, suppressing angiogenesis, and early colorectal liver metastasis events. Proteomic analysis reveals reciprocal regulation between apelin and apelin-dm on proteins associated with clinical outcomes in colon cancer patients. Apelin-dm emerges as a modulator of apelin receptor dynamics, influencing affinity, internalization, and repression of apelin signaling linked to various protein kinases. Pharmacokinetic and toxicity assessments confirm the specificity, safety, and stability of apelin-dm, as well as its facile hepatic metabolism. These findings position targeting proapelin cleavage as a promising therapeutic strategy against metastatic colorectal cancer, paving the way for further clinical exploration.**

**Keywords** Apelin; Furin; Colon Cancer; Liver Metastasis; Safety Pharmacology
**Subject Categories** Cancer; Post-translational Modifications & Proteolysis; Vascular Biology & Angiogenesis

## Introduction

Colorectal cancer (CRC) stands as the second leading cause of cancer-related mortality worldwide and ranks third in global cancer diagnoses (Morgan et al, 2023). Despite the efficacy of surgical resection and adjuvant treatments in early-stage disease, relapse and metastasis remain common. The prognosis of CRC is significantly influenced by local tumor staging, with liver metastasis being the predominant occurrence, followed by the lung. Approximately one in five CRC patients present with liver metastases at diagnosis, and up to half will develop liver metastasis during the disease course (Siriwardena et al, 2014). Early detection of colorectal liver metastasis (mCRC) is paramount for improving survival rates, enabling the selection of candidates for curative liver surgery and guiding treatment decisions.

Apelin, a bioactive peptide initially discovered in the gastrointestinal tract, plays a key role in regulating various physiological functions through its interaction with the G-protein-coupled apelin receptor (Read et al, 2019). Apelin stands at the crossroads of physiological responses, exhibiting heightened expression under hypoxic conditions. Beyond its established roles in promoting epithelial cell growth, proliferation, and migration (Bernier-Latmani et al, 2022a; Wang et al, 2004), apelin has emerged as a key player in the pathophysiology of metabolic and cardiovascular diseases (Palmer et al, 2022; Humphries and Wright, 2008; Adam et al, 2016). This peptide's involvement in cancer is expansive, with documented expressions in various cancers. Apelin overexpression notably correlates with increased tumor growth and microvessel density across diverse cancer types (Bernier-Latmani et al, 2022a; Zuurbier et al, 2017; Lv et al, 2022; Gourgue et al, 2020). In colorectal cancer, apelin concentration becomes a predictive indicator, influencing responses to bevacizumab therapy and affecting susceptibility to apoptosis-inducing agents (Zuurbier et al, 2017).

[1]University of Bordeaux, Bordeaux Institute of Oncology (BRIC)-UMR1312, Bordeaux, France. [2]Department of Cell Biology and Histology, University of the Basque Country, B° Sarriena sn, 48940 Leioa, Spain. [3]Institute of Biomedicine of Seville, University Hospital of Virgen del Rocío/University of Seville/CSIC, Avenida Manuel Siurot s/n, 41013 Seville, Spain. [4]Univ. Bordeaux, CNRS, Bordeaux INP, CBMN, Bordeaux, France. [5]EA3842- CAPTuR, GEIST, Faculté de Médecine, Université de Limoges, 2 rue du Dr Marcland, 87025 Cedex Limoges, France. [6]Institut Cochin, INSERM U1016, CNRS UMR 8104, Université Paris Cité, 75014 Paris, France. [7]Bordeaux Protéome, F-33000 Bordeaux, France. [8]Oncoprot Platform, TBM-Core US 005, Bordeaux, France. [9]Institut Bergonié, Bordeaux, France. [10]Université de Paris, Inserm UMR 1141, Robert-Debré Hospital, 75019 Paris, France. ✉E-mail: Geraldine.siegfried@inserm.fr; majid.khatib@inserm.fr

The cDNA of apelin encodes a 77-amino-acid (aa). The apelin peptides isolated from different tissues and cell types suggest that they are processing products derived from the C-terminal portion of this peptide precursor. The main active forms of apelin are apelin-13 aa, apelin-17 aa, and apelin-36 aa (or proapelin). Apelin-36 is directly generated from preproapelin (55 aa) by not well-known proteases (s) (Lee et al, 2000; Masri et al, 2002). Subsequently, apelin-36 is rapidly cleaved within the two dibasic conserved sequence motifs RRK60F and RR64QR to generate apelin-17 and apelin-13, respectively (Figs. 1A and EV1), suggesting the involvement of proprotein convertases (PCs) in this process (Seidah and Prat, 2012; Siegfried et al, 2020; Scamuffa et al, 2008; Bontemps et al, 2007; Descarpentrie et al, 2022). This study reveals Furin as a unique cellular apelin precursor cleaving enzyme, uncovering the potential clinical significance of unprocessed proapelin, designated apelin-dm, obtained by Furin cleavage sites mutations. When expressed in colon cancer cells or administered pharmacologically to mice with colon cancer tumors, apelin-dm represses the malignant and metastatic phenotype of cancer cells. Apelin-dm altered apelin receptor internalization and signaling and directly competed with apelin and Elabela for apelin receptor binding and activation.

Investigation of apln-dm peptide in a panel of Absorption, Distribution, Metabolism, Excretion (ADME), toxicity assays, and pharmacokinetic profiles demonstrated its good safety profile, specificity, and stability in mice and human serum. This highlights apelin-dm as a promising and safe therapeutic strategy for colorectal cancer and its associated metastatic liver events, paving the way for targeted clinical interventions.

# Results

## Apelin precursor is specifically cleaved by the proprotein convertase Furin

To identify proprotein convertases (PCs) cleaving proapelin, coding region of proapelin was cloned from HUVEC and tagged with a V5 sequence for detection (Fig. 1A). Co-transfection of LoVo cells, a PC-deficient cell line (Takahashi et al, 1993) with apelin precursor and various PC-encoding vectors revealed Furin's exclusive role in reducing apelin-55 and apelin-36 levels while generating apelin-17 and apelin-13 (Fig. 1B). Co-transfection of CHO-FD11, cells with reduced Furin activity (Gordon et al, 1997; Sfaxi et al, 2014) with proapelin and Furin resulted in decreased apelin-17 and increased apelin-13 secretion (Fig. 1C).

To assess the efficiency of PC inhibitors on proapelin processing by endogenous Furin, we treated HEK293A cells with the synthetic Furin inhibitor decanoyl-Arg-Val-Lys-Arg-chloromethyl-ketone (CMK, 10 µM) (Seidah and Prat, 2012; Bontemps et al, 2007) or expressed in these cells the Furin inhibitors, including the Furin prodoamine (P-p-Furin) (Scamuffa et al, 2014) and α1-PDX (Descarpentrie et al, 2022; Jean et al, 1998) (Fig. 1D). As demonstrated by Western blot analysis, transfection of HEK293A cells with a vector encoding proapelin (Control) resulted in ~95% processing. Treatment of cells with CMK or co-transfection with Furin inhibitors revealed that processing of proapelin is significantly blocked by CMK (~10% processing), α1-PDX (~50% procession) and P-p-Furin (~30% processing). In contrast, wild-type α1-antitrypsin failed to inhibit proapelin processing (Fig. 1D). In vitro enzymatic assays (Sfaxi et al, 2014; Basak et al, 2010; López et al, 2021) confirmed increased enzymatic activity with PCs expression and its reduction by Furin inhibitors expression (Appendix Fig. S1), supporting Furin as primary PC mediating proapelin cleavage, producing apelin-17 and apelin-13.

## Apelin, apelin receptor, and Furin expression is altered in colorectal cancer and colorectal liver metastases

In a healthy colon, crypts contain stem cells responsible for the constant renewal and differentiation of the epithelial lining. While in cancerous tissue, these processes are disrupted, leading to uncontrolled cell proliferation, aberrant differentiation, and loss of normal crypt architecture (Humphries and Wright, 2008). Given the co-expression of apelin, apelin receptor and Furin, that we identified in colon tissues, colon carcinoma cells, and endothelial cells (Appendix Fig. S2), we sought to evaluate their expression and colocalization in both normal and colon cancer tissues. To do this, we first conducted immunohistochemical staining on a set of matched tissue sections, including normal human colon mucosa, primary colon tumors, and corresponding colorectal liver metastases from 35 patients. All these proteins were expressed in non-cancerous colon tissues, primarily localized to the base of the colon crypts (Fig. 1E,F). The expression of apelin, apelin receptor and Furin in the colon crypts highlights their potential roles in maintaining the dynamic environment of the intestinal epithelium. In cancerous tissues, the loss of crypts was associated with a modified expression pattern of apelin (Fig. 1H), apelin receptor (Fig. 1G), and Furin (Fig. 1G,H), suggesting that these molecules may play roles in tumor progression and possibly in the tumor microenvironment. Quantitative analysis of immunohistochemistry staining in primary colorectal cancer (CRC) and metastatic colorectal cancer (mCRC) tumors, along with their respective normal colon tissues, revealed a consistent increase in the average staining levels of these proteins in the analyzed tumors (Appendix Fig. S3). Moderate correlation coefficients (ranging from 0.3 to 0.5) were observed between the expression levels of apelin and KI-67, apelin receptor and KI-67, as well as Furin and KI-67 in primary CRC (Fig. 1I) and their corresponding metastatic liver lesions (Fig. 1J). These positive correlations underscore the potential roles of apelin, apelin receptor, and Furin in CRC progression and metastasis. Although APLN mRNA was overexpressed in CRC tumors and their corresponding metastatic livers, the average tumoral/normal (T/N) ratio of apelin expression in hepatic metastases was almost similar to that of colon primary tumors (Fig. 1K). Furthermore, APLNR (Fig. 1L) and Furin (Fig. 1M) were overexpressed in metastatic tumors, with levels higher than those in matched healthy and primary colon tumors.

Hypoxia is a common feature of the microenvironment in almost all solid tumors and is frequently associated with angiogenesis and cancer growth, including CRC and has been reported to induce APLN expression in endothelial cells (Andersen et al, 2011) and Furin during regenerative angiogenesis (Khatib et al, 2016). Analysis of Furin expression in endothelial cells cultured under normoxic (21% $O_2$) and hypoxic (1% $O_2$) conditions revealed that hypoxia is also a strong inducer of Furin mRNA in endothelial cells, with increased expression observed at all tested hypoxia exposure times ranging from 4 to 24 h (Fig. 1N).

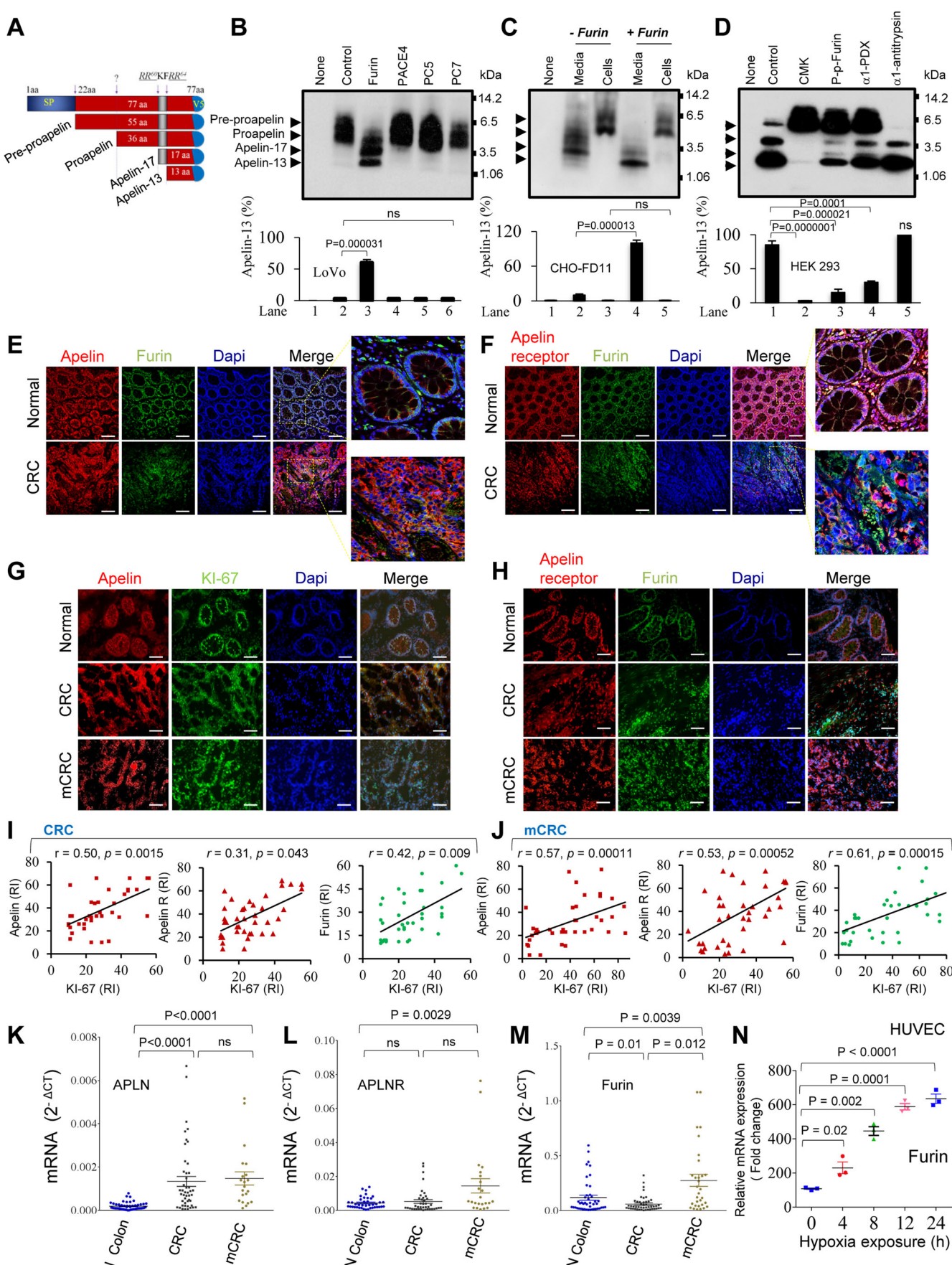

**Figure 1. Specific cleavage of apelin precursor (proapelin) by Furin and altered expression of apelin, Furin and apelin receptor in colon cancer and colorectal liver metastasis.**

(A) Schematic representation of the primary structure of the 77-amino acid (aa), 55 aa and 36 aa of human apelin precursors and the maturated apelin-17 and apelin-13 aa peptides. The signal peptide (SP), the cleavage site of apelin-55 to apelin-36, and the two proprotein convertase (PCs)-processing sites are indicated (arrows). (B) Proapelin processing by PCs was analyzed by immunoblotting in media of LoVo cells, a PCs activity-deficient cell line, that were co-transfected with empty vector (None), or vectors containing proapelin (Control) and indicated PCs cDNAs. (C) Proapelin processing by Furin in media and lysates from CHO-FD11 cells with reduced Furin activity co-transfected with empty vector (None), or vectors containing preproapelin and/or Furin constructs, was analyzed by western blotting. (D) Inhibition of proapelin processing was assessed by Western blot analysis in media of HEK293A cells co-transfected with empty vector (None), or vectors containing preproapelin (Control) and indicated Furin inhibitors, or treated with the PC inhibitor decanoyl-Arg-Val-Lys-Arg-chloromethyl-ketone (CMK, 10μM). (B–D) The bars show the corresponding percentage of band intensities deduced from the ratio of apelin/(proapelin + apelin) and indicate the level of mature apelin accumulation (%). Data are representative of independent experiments ($n = 3$), and all values represent the mean ± s.e.m. $P$ values were determined by a two-tailed Unpaired $t$ test. ns not significant. (E, F) Representative immunofluorescence images of colon cancer samples (CRC) and adjacent healthy tissues (Normal) from patients ($n = 35$) that were stained with anti-apelin (red) and anti-Furin (green) (E) or with anti-apelin receptor (red) and anti-Furin (green) (F). (G, H) Representative images of apelin (red) and KI-67 (green) staining (G) and apelin receptor (red) and Furin (green) (H) staining in 35 CRC, mCRC and corresponding normal tissues. (I, J) Plots of the correlation between KI-67 and apelin, KI-67 and apelin receptor, and KI-67 and Furin expression in CRC and mCRC that was determined by immunofluorescence following staining intensity analysis of (G, H) data, as measured on ImageJ and Spearman's correlation analysis. (K–M) Relative expression of APLN (K), APLNR (L) and Furin (M) mRNA in human normal colon (N colon, $n = 49$ patients), corresponding CRC ($n = 46$ patients) and mCRC livers ($n = 30$ patients). Significant differences P were determined by Unpaired $t$ tests using mean ± s.e.m. values. (N) Kinetics of Furin mRNA accumulation in HUVEC cells cultured in normoxia (21% $O_2$) or hypoxia (1% $O_2$) for various time periods as indicated and data are representative of three independent experiments with values that represent the mean ± s.e.m. Significant differences $P$ were determined by two-way ANOVA. Scale bar, 100 μm. Source data are available online for this figure.

Taken together, these findings indicate that the increased co-expression levels of apelin, the apelin receptor, and/or Furin are associated with CRC tumors, suggesting a potential role for these proteins in colon cancer progression and metastasis.

## Alteration of proapelin cleavage sites contributes to the reduction in colorectal tumor growth and liver metastasis

To investigate the functional significance of apelin precursor processing by Furin in colon cancer, we mutated the proapelin cleavage sites from RR60KFRR64QR to SS60KFSS64QR, creating the apelin double mutant (apelin-dm) (Figs. 2A and EV1). Lentiviral vectors carrying mock, APLN, and APLN-DM were then introduced into colon cancer cell lines CT-26 and MC-38 (Appendix Fig. S4A). Despite Furin presence, the mutations prevent apelin-dm processing (Fig. 2A). Wild-type APLN expression in cancer cells led to twofold increased anchorage-independent growth, while APLN-DM reduced proliferation and showed >80% decrease (Appendix Fig. S4B, C). APLN expression prevented induced apoptosis, unlike APLN-DM (Appendix Fig. S4D). In vivo tumor growth assay in Rag2/γc mice showed apelin-increased growth, while apelin-dm significantly reduced it (Fig. 2B,C). Apelin-dm also inhibited tumor growth in syngeneic BALB/c and C57BL/6 mice (Fig. 2D,E), using both CT-26 and MC-38 cells. The expression of APLN-DM in these cells seems to affect their ability to mediate tumor growth with varying efficacy. Indeed, a greater inhibitory effect was observed in CT-26 cells compared to MC-38 cells, suggesting the potential implication of genetic background differences in the mice used and/or differences in the tumorigenic mediators secreted by CT-26 and MC-38 in their tumor microenvironments. In addition, MC-38 cells produce more apelin compared to CT-26 cells (Appendix Fig. S4A), suggesting the potential involvement of one or more of these differences in the lower anti-tumor efficacy of APLN-DM expression in MC-38 cells compared to CT-26 cells. Similarly, overexpression of APLN in MC-38 and CT-26 cells increased their ability to induce hepatic metastases and incidence, which is repressed by APLN-DM expression (Fig. 2F,G). Notably, APLN-DM-expressing cell-derived tumors exhibited a significant decrease in KI-67+ cells

(Fig. 2H–J). Taken together, these findings suggest that apelin-dm represses tumor growth, angiogenesis, and liver metastasis that are otherwise promoted by apelin.

## Apelin-dm inversely regulates a protein subset induced by apelin, impacting clinical outcomes in colon cancer

To identify the pathways mediated by apelin-dm involved in the repression of the malignant phenotype of colon cancer cells, we performed proteome analysis of cancer cells and the same cells expressing APLN and APLN-DM. Out of the 1825 cellular proteins identified, the expression of APLN and APLN-DM in cancer cells significantly increased the expression of only 14 and 153 proteins, and decreased 20 and 67 proteins, respectively (Fig. 2K). Further analysis revealed that the most significantly dysregulated proteins in response to APLN and APLN-DM expression in cancer cells were related to extracellular matrix regulation, metabolic activity, cytokine production, cell death, and growth (Dataset EV1). In cells expressing APLN-DM, the most induced proteins are single-stranded interacting protein 2 (Rbms2) and RNA-binding motif (RBP). Rbms2 is a tumor suppressor gene that inhibits cell proliferation by positively regulating the stability of P21, mediates cell apoptosis, and enhancing sensitivity to several cancer treatments (Xu et al, 2022; Sun et al, 2018). The protein most repressed in apelin-dm-expressing cells was the oncogene GTPase NRas. Importantly, we found that 28 proteins were inversely regulated in cells expressing APLN and APLN-DM (Fig. 2L). Several of these proteins were linked to both good and poor prognosis in colon cancer patients. Indeed, Galectin-3-binding protein (LGALS3BP), gelsolin (Gsn), and Spartin (Spg20), which are associated with a favorable clinical outcome in colorectal carcinoma are downregulated in APLN-expressing cells and upregulated in APLN-DM-expressing cells. In contrast, ATP-dependent RNA helicase (DHX36), Synembryn-A (Ric8a), Protein CYR61 (Cyr61), Tubulin alpha-1A (Tuba1a), and probable 28S rRNA (cytosine-C(5))-methyltransferase (NSUN5), which are associated with poor prognosis in colon cancer, are upregulated in APLN-expressing cells and downregulated in APLN-DM-expressing cells. Further analysis revealed that LGALS3BP

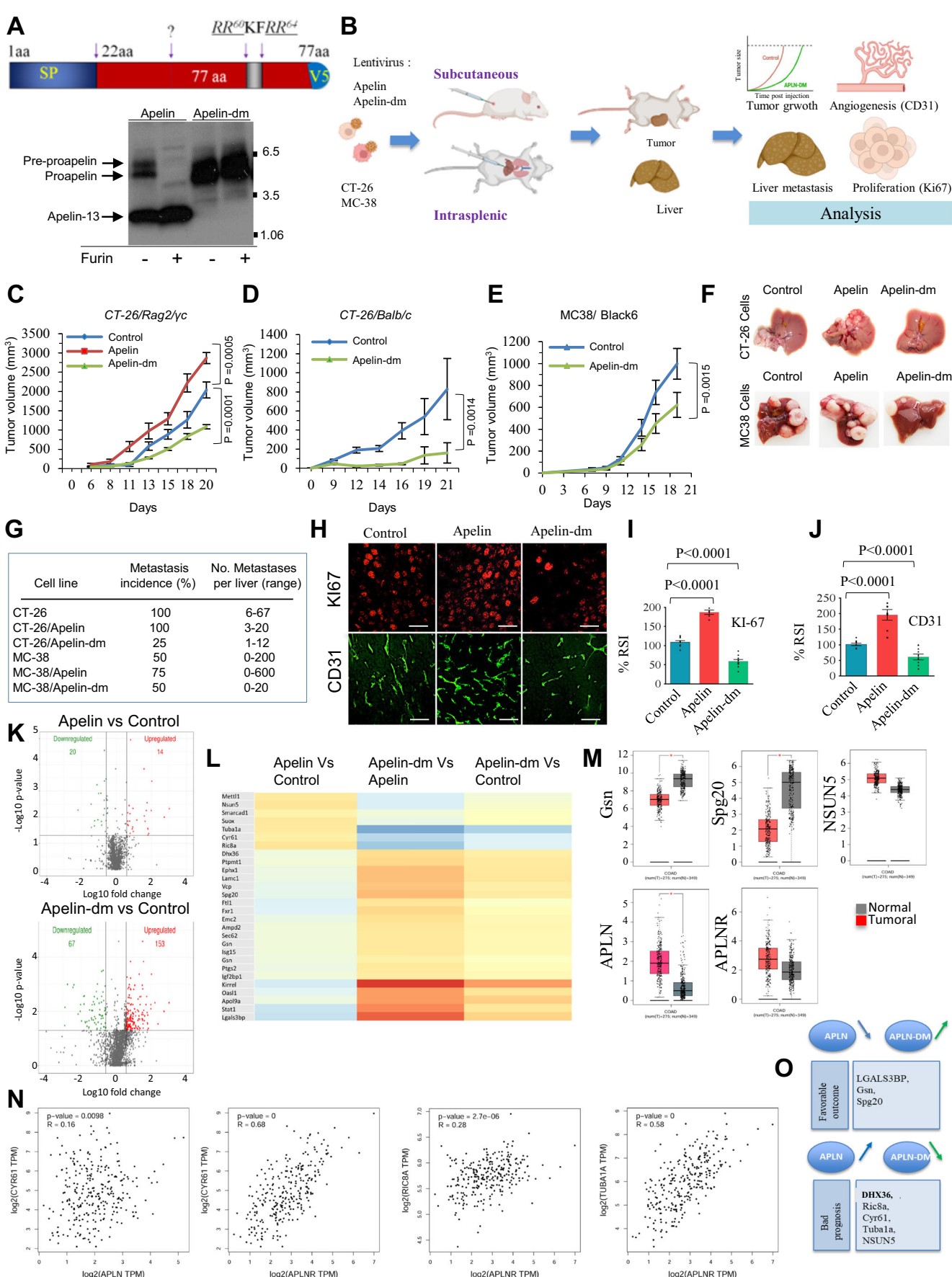

**Figure 2. Alteration of proapelin cleavage sites represses colorectal tumor growth and colorectal liver metastasis.**

(A) Colon cancer cells CT-26 and MC-38 infected with lentivirus for stable expression of apelin and apelin-dm generated by mutation of the indicated PCs cleavage sites RR$^{60}$KFRR$^{64}$ (arrows) and their processing were analyzed by Western blot analysis in the absence and presence of Furin (+/−). (B) Experimental design of in vivo experiments, indicating subcutaneous and intrasplenic injected mice with CT-26 and MC-38 cells expressing APLN and APLN-DM and analysis of developed tumors. (C–E), Tumor growth curves over time from representative experiments of subcutaneous inoculation of control CT-26 and MC-38 cells (10$^{-6}$) or the same cells stably expressing APLN and APLN-DM into Rag2/$\gamma_c$ (C), Balb/c (D), or Black6 (E) mice (n = 8 tumors per group/experiment, 3 experiments). (F) Representative images of livers from Black6 and BALB/c mice injected intrasplenically with control CT-26 and MC-38 cells or the same cells stably expressing APLN and APLN-DM (n = 7 tumors per group/experiment, 3 experiments), respectively. (C–E) Error bars indicate s.e.m. center values indicate mean (Mann–Whitney *U* test). (G) Incidence of liver nodules (%) and the number of metastases per liver (median) 15 days post-intrasplenic injection are shown. (H) Representative immunofluorescence images of liver tumor sections derived from control CT-26 cells or the same cells stably expressing APLN and APLN-DM that were stained with anti-Ki67 (proliferative index, red) and anti-CD31 (angiogenic index, green). Scale bar, 100 μm. (I, J) The percentage of relative staining intensity (% RSI) of KI67, and CD31 for control CT-26 tumors and CT-26 tumors expressing apelin and apelin-dm. Data are representative of three independent experiments as mean ± s.e.m. P values by two-tailed Unpaired *t* test. (K) Volcano plot of proteins in APLN and APLN-DM-expressing cells vs. Controls. Green dots indicate downregulated proteins; and red dots, upregulated proteins, Statistical significance was determined using Welch's *t* test, with a P value threshold of 0.05 (n = 3). (L) Heatmaps plotting expression of the 28 proteins inversely regulated between APLN and APLN-DM-expressing cells. (M) Expression level of GSN, Spg20, NSUN5, APLN, APLNR in COAD (n = 275) or non-cancerous tissues (n = 349) from Gene Expression Profiling Interactive Analysis (GEPIA) datasets (central band, boxes, and whiskers of the boxplot represent the median, first quartile, third quartile, minimum, and maximum values, respectively). Four-way ANOVA, using sex, age, ethnicity, and disease state, was used to calculate statistical significance. (N) Data were derived from GEPIA (Tang et al, 2017) indicating scatter plot graphs of the Spearman correlation analysis of CYR61 and APLN, CYR61 and APLNR, RIC8A and APLN, and RIC8A and APLNR in colon adenocarcinoma. (O) Schematic representation that summarizes APLN and APLN-DM regulated proteins associated with clinical outcome of colorectal cancer patients. Source data are available online for this figure.

expression was most affected by APLN-DM (~9.5-fold increase, Dataset EV1). LGALS3BP has anti-tumor activity in colon cancer by suppressing Wnt signaling (Piccolo et al, 2015) and also plays a role in preventing and treating inflammatory diseases by suppressing TAK1-dependent NF-κB activation (Hong et al, 2019). Therefore, based on the ability of APLN-DM to regulate the expression of proteins involved in both good and poor prognosis of colon cancer patients, we used the web server GEPIA (http://gepia.cancer-pku.cn/) (Tang et al, 2017) to analyze their expression in colon cancer patients. We found that Gsn and Spg20 mRNA expression was significantly lower, and NSUN5 mRNA expression was significantly higher in colon tumor tissues than in normal tissues (Fig. 2M). The same analysis also revealed upregulated expression of APLN and APLNR in colon cancer tumors (Fig. 2M). We next explored the correlation between the expression of APLN and APLNR with genes that are dysregulated by APLN or APLN-DM expression in cancer cells, and involved in prognosis of colon cancer patients. Among the dysregulated molecules, we observed varying degrees of correlation with APLN and APLNR. Specifically, we found a low positive correlation between APLN and CYR61 (r = 0.16), a stronger correlation between APLNR and CYR61 (r = 0.68), a low correlation between APLNR and RIC8A (r = 0.20), and a moderate correlation between APLNR and TUBA1A (r = 0.58) in tumor patients (Fig. 2N). Collectively, these results indicate that APLN-DM induces the expression of proteins associated with a favorable clinical outcome and limits the expression of proteins induced by APLN that are involved in the poor prognosis of colon cancer (Fig. 2O).

## Apelin-dm peptide represses apelin-mediated vascular network and tumor cells malignant phenotype

We next synthesized the apelin-dm peptide to explore its therapeutic potential. In apelin-36, the sequence "*RRKFRR*" has been substituted with "*SSKFSS*", where the R residues have been replaced by S. R is positively charged at neutral pH with approximate molecular (MW) of 174.2 Da. Whereas S is neutral with MW of 105.1 Da. This difference in molecular weight and

charge properties likely contributes to the lower migration of apelin-dm peptide on SDS-PAGE than apelin-36 (Fig. 3A,B). First, the impact of varying concentrations of apelin-dm on in vitro angiogenesis was assessed through a tube-like structure formation assay (Appendix Fig. S5A,B). This assay, which evaluated both the number of junctions and tubule length, revealed that apelin-dm disrupted the formation of capillary-like structures in HUVECs in a concentration-dependent manner, with 100 nM being the lowest concentration that elicited a significant effect (Appendix Fig. S5A,B). Comparative analysis with Bevacizumab revealed similar inhibitory effects, with EC50 values of 0.130 μM for Bevacizumab and 0.083 μM for apelin-dm, respectively (Appendix Fig. S5C). Previously, Bevacizumab was shown to exhibit inhibitory effects at concentrations around 100 nM, both in vitro and in CAM assays (Ljoki et al, 2022; Ademi et al, 2021), suggesting a similar anti-angiogenic effect for apelin-dm and Bevacizumab at this concentration. In the presence of apelin, HUVECs formed a network of capillary-like tubes, evidenced by a significant increase in the number of junctions and tube length, and this effect was counteracted by apelin-dm (Fig. EV2A,B). In addition, we found that apelin-dm inhibited both HUVEC and SMC proliferation and migration, especially those induced by apelin-13 or apelin-36 (Fig. EV2C–E). Real-time PCR showed the expression of apelin and its receptor in SMCs and HUVECs (Fig. EV2F). Analysis of the apelin receptor at the protein level revealed its high expression in HUVEC cells compared to CT-26 and MC-38 cells (Fig. EV2G). Further analysis revealed high expression of Furin in SMCs (5193 copies/cell) and HUVECs (6013 copies/cell), while it was lower in cancer cells, with 3667 and 3713 copies/cell for CT-26 and MC-38, respectively (Fig. 3C).

Angiogenesis is a complex intercellular process involving the organization of various cells. To evaluate the effect of apelin-dm in the context of whole tissue, we performed the aortic ring assay (Fig. EV2H). While apelin induced sprouting of vessel-like structures in the aortic ring assay, apelin-dm repressed this effect. We examined whether apelin-dm could interfere with VEGF-induced angiogenic sprouting in endothelial cells. Our findings indicate that apelin-dm prevents sprouting in endothelial cells

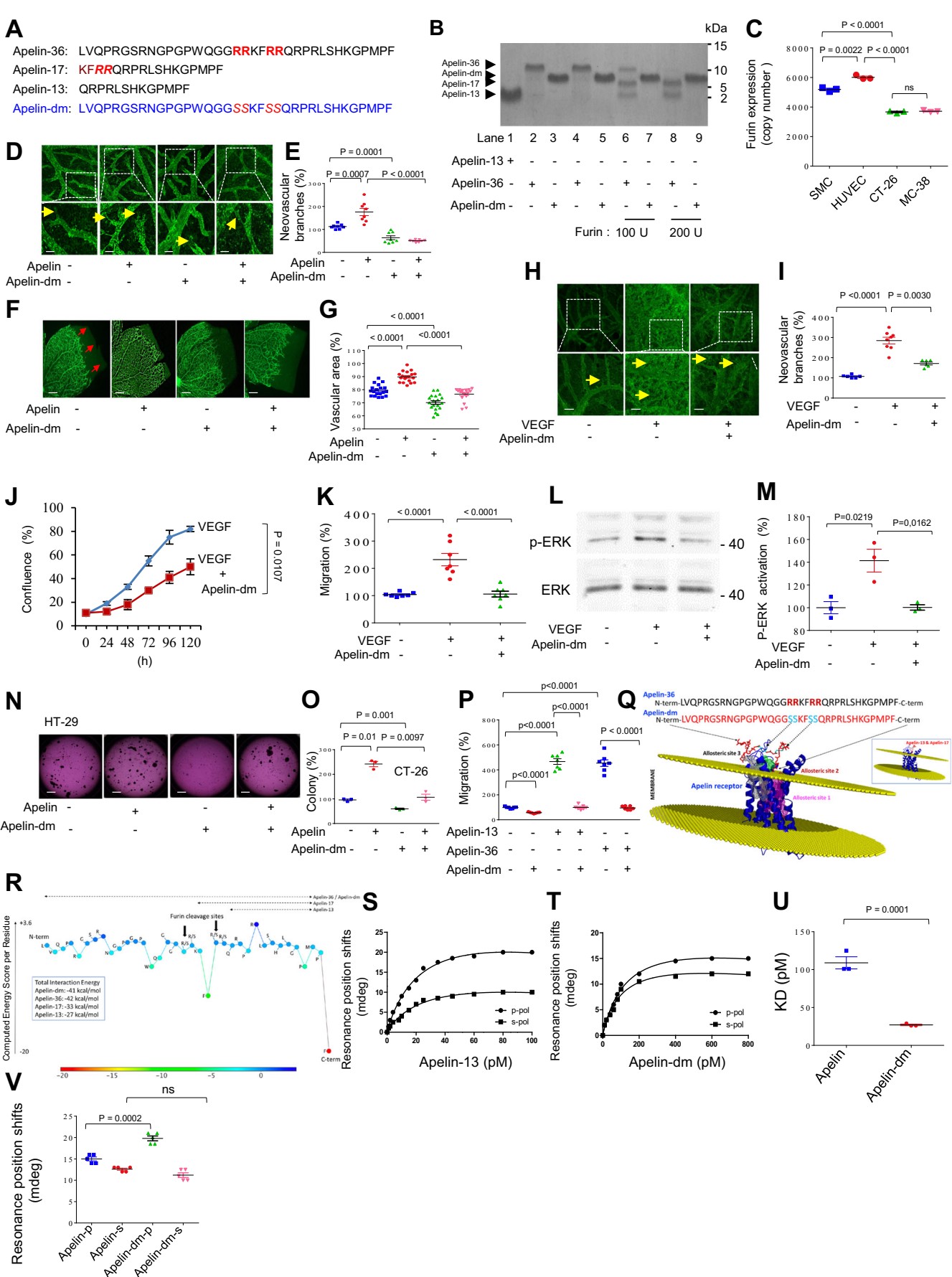

**Figure 3.  Apelin-dm peptide represses apelin-mediated vascular network and tumor cells malignant phenotype: Apelin-dm and apelin competition for apelin receptor binding.**

(A) Amino acid sequences of apelin-13, apelin-17, apelin-36 and apelin-dm peptides. (B) SDS gel analysis of wild-type proapelin and apelin-dm peptide cleavage by Furin. Apelin-13 was added for comparaison. The difference in molecular weight and charge properties of the aa R and S in apelin-dm contributes to the lower migration of apelin-dm peptide than apelin-36. (C) Furin expression analysis expressed as copy number per cell in smooth muscle cells (SMC), endothelial cells (HUVEC), and cancer cells CT-26 and MC-38 using droplet digital polymerase chain reaction (ddPCR) assay. (D) Representative images of chick chorioallantoic membrane treated for 48 h with apelin (100 nM) or/and apelin-dm (100 nM). Vessel density quantification is presented in (E), with $n = 8$ CAM per group. (F) Representative image of Isolectin B4 staining in retinal whole mounts from 100 nM apelin, apelin-dm or apelin and apelin-dm-treated mice on P5. (G) Quantification of retina vascularization, $n = 6$ total animals per group, from three different experiments. Red arrows indicate the direction of vessel formation. (H) Representative images of CAM ($n = 8$ per group) treated for 48 h with VEGF (30 ng/ml) and/or apelin-dm (100 nM), with small formed vessels indicated by arrowheads. (I) Small formed vessels density quantification in CAM ($n = 8$ per group). (J) Cells were plated at low confluence for time-lapse phase-contrast video microscopy using an IncuCyte microscope, and cell proliferation in the presence of VEGF (30 ng/ml) or apelin-dm (100 nM) was monitored by automated confluence analysis at set intervals after plating (means, $n = 6$ wells per group, three independent experiments). (K) Effect of apelin-dm (100 nM) on VEGF (30 ng/ml)-induced migration of HUVECs. (L, M) Effect of apelin-dm (100 nM) on VEGF (30 ng/ml)-induced ERK phosphorylation in HUVEC cells analyzed by immunoblotting (L). Quantification of ERK phosphorylation relative to control untreated HUVEC assigned 100% (M). (N) Anchorage-independent colony formation assay was performed on colon cancer cells HT-29 in the presence of apelin and/or apelin-dm. (O) The number of colony >100 μm were counted and the results were presented as the percentage of the developed colonies. (P) Effect of apelin peptides and/or apelin-dm on the migration of colon cancer cells CT-26. All data are representative of three independent experiments. All values represent the mean±s.e.m. Significant differences P were determined by two-way ANOVA. scales, 500 μm. (Q) Docking peptides into apelin receptor. A representative binding poses for apelin-dm (red) is shown, with the C-terminal region inserted in the main binding pocket (allosteric site 1), while the N- terminal region seemed to wrap around the receptor. The predicted structure of the apelin-36 and apelin receptor is highly similar and is also shown. The two Furin cleavage sites are shown. All apelin peptides are predicted to interact with another allosteric pocket (allosteric site 2) while only apelin-36 or apelin-dm is likely to contact yet another allosteric binding pocket (allosteric site 3). Colored spheres (yellow) show the predicted position of the membrane. Inset, apelin-17 and apelin-13 were also docked into the receptor and numerous contacts between these peptides and the protein are lost as compared to apelin-dm and apelin-36. (R) Apelin-apelin receptor docking scores per-residue and global binding energy scores. The per-residue decomposition of apelin-dm (or apelin-36) docking score is shown (the per-residue scores between the two peptides are very similar). Each dot represents an amino acid, the color code is assigned according to the predicted binding score with apelin receptor for each residue. Only some residues are predicted to contribute significantly to the binding with the receptor (e.g., the C-term F). The approximate total interaction energy values for each peptide and the receptor are also shown. (S–U) Plasmon waveguide resonance (PWR). Binding curve for apelin (S), and apelin-dm (T) interaction with apelin receptor in HEK-apelin receptor. (U) Plots showing the results of KD values ($n = 3$) corresponding to (S, T). (V) Conformational changes of apelin receptor in response to apelin and apelin-dm. Spectral changes induced by apelin and apelin-dm binding to apelin receptor with two polarizations (p-pol and s-pol) ($n = 3$). P p-(perpendicular to the sensor surface) polarized light, S s-(parallel to the sensor surface). (U, V) Data are representative of three independent experiments and shown as the mean ± s.e.m. Significant differences P determined by one-way ANOVA with Tukey's multiple comparison tests. Source data are available online for this figure.

caused by VEGF throughout the treatment periods (Fig. EV2J,K). Under these conditions, apelin-dm was comparable in effect to Bevacizumab. We also tested the anti-angiogenic effect of apelin-dm using the chick chorioallantoic membrane (CAM) assay. Apelin-dm treatment impaired both basal CAM angiogenesis and CAM angiogenesis induced by apelin. Examination of the CAM vasculature by isolectin B4 staining showed that while apelin induced vessel formation, apelin-dm mainly affected the formation of small vessels (Fig. 3D,E, yellow arrows, and (Fig. EV3A). In addition, we used the neonatal mouse retinal model as an in vivo assay to examine the effect of apelin-dm on neoangiogenesis. At birth, the retina is avascular, and a superficial vascular plexus grows from the center to the periphery during the first week after birth (Selvam et al, 2018). Isolectin B4 staining of the retinal vasculature at P5 revealed that apelin administration significantly increased vascular sprouting in mice compared to controls (Fig. 3F,G). In both untreated and apelin-treated mice, sprouting vessels were reduced by apelin-dm administration. In addition, apelin-dm hindered VEGF-induced endothelial cell sprouting, and disrupted angiogenesis in the CAM assay (Figs. 3H,I and EV3B).

To investigate the mechanism by which apelin-dm represses VEGF-induced angiogenesis, we first analyzed its effect on HUVEC proliferation (Fig. 3J) and migration (Fig. 3K) in response to VEGF. As indicated, while VEGF significantly promoted HUVEC proliferation and migration, the presence of apelin-dm suppressed these effects. Furthermore, analysis of ERK activation, a signaling pathway mediated by VEGF and involved in HUVEC proliferation and migration, showed that while VEGF induced ERK activation, treatment with apelin-dm attenuated this effect (Fig. 3L,M). This suggests that apelin-dm interferes with the VEGF signaling

pathway, which may explain the reduced angiogenesis mediated by VEGF in the presence of apelin-dm.

In soft-agar assays, apelin-dm significantly inhibited the clonogenicity of colon cancer cells, countering the apelin-induced increase in colony formation (Fig. 3N,O). Apelin-13 and apelin-36 stimulated cancer cell migration, while apelin-dm mitigated this effect, suggesting interference with apelin's autocrine action (Fig. 3P; Appendix Fig. S6). Indeed, all the analyzed colon cancer cells express both apelin and the apelin receptor (Appendix Fig. S2B).

## Computational docking analysis of apelin peptides and apelin-dm with apelin receptor: predicting binding modes and affinity

Aligning apelin-13 with the experimental structure of the AMG3054 peptide mimetic in the presence of the active apelin receptor cryo-EM structure suggests that the C-terminal region of apelin-13 inserts into a deep pocket (Fig. 3Q). Additional CABS-flex simulations induced minor changes in the N-terminal region of apelin-13, while the C-terminus remained deeply anchored compared to the starting position of the peptide. Apelin-dm, apelin-36, and apelin-17, docked using CABS-dock, wrapped around the apelin receptor, showing additional favorable interactions in the peptide C-terminal region compared to apelin-13, particularly for apelin-dm and apelin-36. Models of the apelin receptor with apelin-dm exhibited higher total binding energy values than models with apelin-13 or apelin-17, indicating superior binding of apelin-dm. The predicted binding energy values between apelin-36 or apelin-dm and the receptor were similar (Fig. 3R). APOP identified allosteric sites, with the top site matching the peptide-binding site in the C-terminal region (allosteric site 1) (Fig. 3Q). Another allosteric site, accessible to

apelin-13, apelin-17, apelin-36, and apelin-dm, was identified and labeled allosteric site 2. A third allosteric pocket (allosteric site 3) can also interact with the N-terminal region of apelin-36 or apelin-dm (Fig. 3Q). The predicted binding energy scores favored apelin-dm or apelin-36 over apelin-13 or apelin-17 (Fig. 3R).

## Apelin-dm peptide-driven changes in apelin receptor conformation

Apelin and apelin-dm's effects on apelin receptor affinity and conformation were assessed using plasmon waveguide resonance (PWR) (Soulet et al, 2020). The sensor, pretreated with polylysine, captured cell fragments through electrostatic interactions. After capturing fragments, ligand incrementation-induced spectral changes tracked for both polarizations. Ligand affinity analyses showed apelin-dm's highest affinity (KD ~30 pM) for cell membrane fragments, surpassing apelin (KD ~100 pM) (Fig. 3S–U). Spectral changes under ligand binding revealed distinct conformational changes induced by apelin-dm and apelin-13, with apelin-dm causing more substantial and anisotropic shifts (Fig. 3V). These differences indicated ligand-induced distinct receptor conformational states. Notably, using membranes from HEK cells lacking apelin receptor under the same conditions did not induce significant conformational changes in the receptor (Appendix Fig. S7).

## Apelin-dm exhibits high-affinity competition with apelin for apelin receptor binding

The ability of apelin-dm to repress various biological functions of apelin suggests that apelin-dm probably acts as an inhibitor of endogenous and exogenous apelin in terms of receptor binding. Previously, apelin was reported to cause clathrin-mediated apelin receptor internalization (He et al, 2016; Reaux et al, 2001; El Messari et al, 2004a) and translocation of β-arrestin to the cell surface, indicating translocation to the phosphorylated apelin receptor (Lee et al, 2010). After apelin-induced internalization, the apelin receptor can either be recycled to the cell surface or be degraded in lysosomes (Lee et al, 2010). Therefore, we compared the effects of apelin and apelin-dm on the cellular recycling of the apelin receptor. To facilitate the functional analysis of receptor internalization, we used U2OS cells stably co-expressing the human apelin receptor and β-arrestin 2-GFP. Stimulation of these cells with Tamara-apelin for 30 min induced receptor internalization, as visualized by the appearance of a punctuated pattern of red (Tamara-apelin) and green (apelin receptor/arrestin 2-GFP) fluorescence (Fig. 4A). In contrast, cells treated with apelin-dm showed reduced Tamara-apelin receptor internalization (Fig. 4A). The use of the same cells revealed that apelin-dm, like apelin, induces receptor internalization that colocalizes with clathrin (Fig. 4B). Similar to apelin, apelin-dm dose-dependently increases the Bioluminescence Resonance Energy Transfer (BRET) signal between the apelin receptor and Rab5 (Fig. EV4), confirming the ability of both apelin and apelin-dm to promote apelin receptor internalization. Using HEK293A cells stably expressing apelin receptor-EGFP fusion protein, we found that six hours after apelin-13 washout, the apelin receptor seemed to be mainly returned to the cell surface. In contrast, in cells treated with apelin-dm, the intracellular vesicles of the apelin receptor remained internalized (Fig. EV5), suggesting delayed or blocked recycling of the apelin receptor in the presence of apelin-dm. To further validate that

apelin-dm functions by directly competing with apelin for receptor binding, competitive binding assays were performed to verify the binding affinities of the radiolabeled peptide, [125I]-(Pyr1)-apelin-13, in the presence of apelin-13 and apelin-dm. As shown in Fig. 4C, both peptides potently inhibited the specific binding of the [125I]-(Pyr1)-apelin-13 peptide to the apelin receptor in a concentration-dependent manner. However, the apelin-dm peptide showed an IC50 about 2.5 times smaller than that of apelin-13 (0.45 nM versus 1.13 nM), suggesting that lower concentrations of apelin-dm are required to compete with apelin-13 for apelin receptor binding. These results indicate that apelin-dm competes with apelin for apelin receptor binding and represses its activation and internalization.

## Apelin-dm modulates Elabela activity and its receptor binding

Alongside apelin, Elabela can activate the apelin receptor, boasting a distinct amino acid sequence from apelin (Read et al, 2019). It colocalizes with apelin in endothelial cells and circulates in plasma. Elabela shares functional roles with apelin in angiogenesis and cancer (Yang et al, 2017; Soulet et al, 2020; Nys et al, 2024). To investigate whether apelin-dm affects the interaction between Elabela and the apelin receptor, we conducted PWR and ligand affinity analyses. Our findings showed that apelin-dm competes with Elabela's binding to the apelin receptor, as evidenced by an increased KD from 27 ± 1 to 56.6 ± 2.6 in the presence of apelin-dm (Fig. 4D). In addition, ERK activation analysis in HEK-apelin receptor cells revealed that apelin-dm inhibited Elabela-induced ERK activation (Fig. 4E). These results suggest that apelin-dm competes with apelin and Elabela for the apelin receptor and influences their functions.

## Apelin-dm targets apelin and non-apelin-relayed signaling pathways

To evaluate the impact of apelin-dm on apelin receptor down-stream signaling, we initially explored its potential effect on the ERK and AKT pathways, known to be activated by apelin (Masri et al, 2004). Our findings revealed that while apelin treatment activated ERK and AKT in HUVECs, SMCs, CT-26 cells, and HEK-293 cells, apelin-dm significantly inhibited these processes (Fig. 4F,G; Appendix Fig. S8). For a more comprehensive assessment of the effect of apelin-dm on various kinase activities, we conducted a screening of diverse kinase activities using the PamGene technology (Versele et al, 2009) that showed substantial differences in kinase activity profiles in control and treated cells. Apelin treatment affected phosphorylation in 60 PTK and 58 STK peptides, whereas apelin-dm induced lower changes in phosphorylation in 15 PTK and 37 STK peptides (Fig. 4H–K; Appendix Fig. S9). In addition, 22 PTK and seven STK peptides exhibited significantly dysregulated phosphorylation between apelin and apelin-dm-treated cells. Apelin treatment indicated significantly higher activity for Src family kinases (Src, Lyn, Lck, HCK, BLK, Fyn) and insulin receptor family (InsR, IRR) among PTKs, as well as heightened activity in the ERK (ERK1 and ERK2) and CDK (CDK11, CDK9) families within STKs (Fig. 4L,M; Appendix Figs. S10 and 11). Conversely, in apelin-dm-treated cells, Src family kinases (Lyn, Lck, HCK, BLK) and CDK family (CDK11,

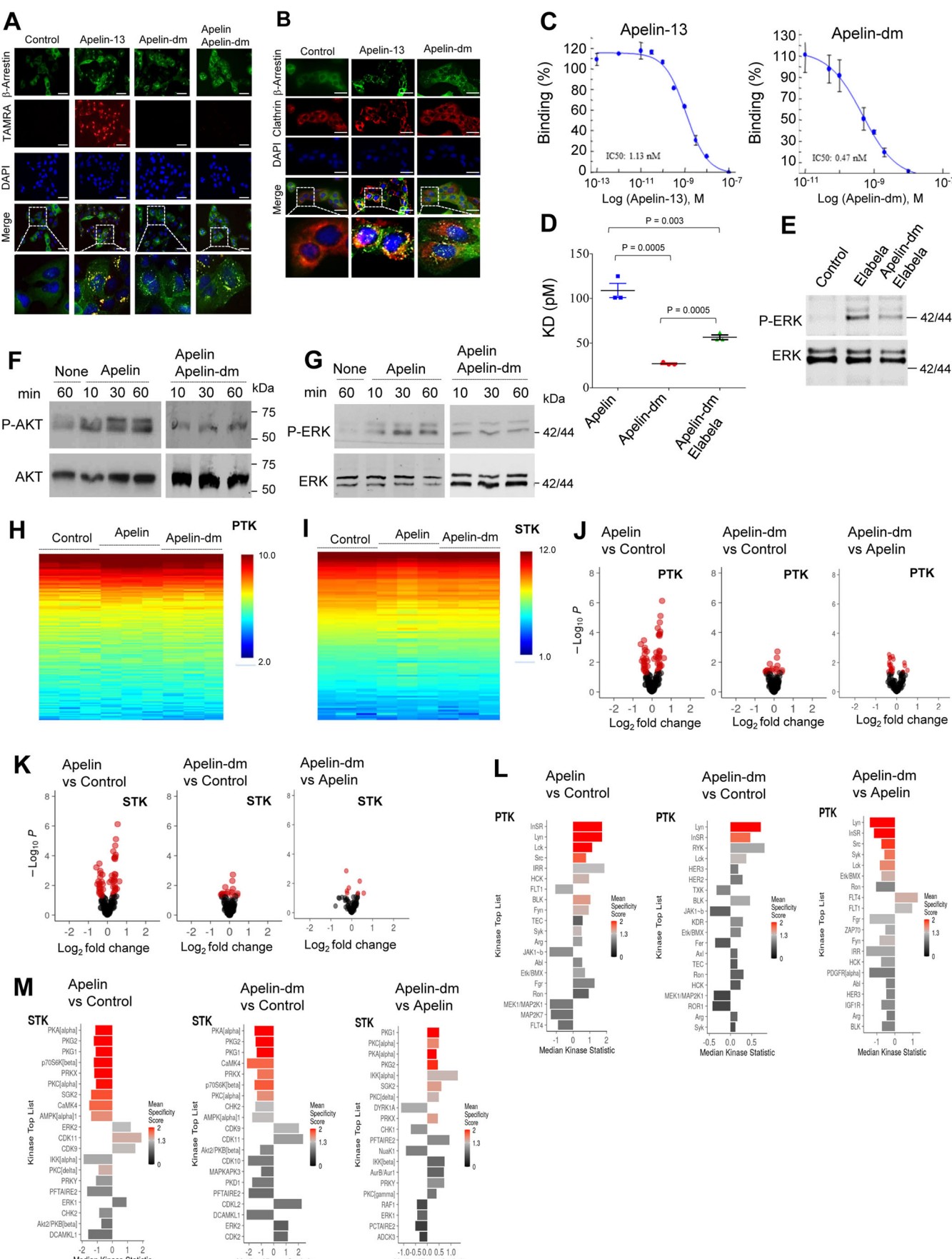

Figure 4. Effect of apelin-dm on apelin receptor internalization and signaling, and Kinome profiling.

(A) Representative images of Fluorescence microscopy used to visualize β-arrestin 2-GFP recruitment to the human apelin receptor in U2OS cells stably co-transfected with apelin receptor and pEGFP-β-arrestin 2 in the absence and presence of apelin-Tamara and/or apelin-dm. (B) Representative images of clathrin staining (red) and β-arrestin 2-GFP (green) recruitment to the human apelin receptor in U2OS cells in the absence and presence of apelin and apelin-dm. Scale bar, 50 μm. (C) Radioligand binding assay. Apelin receptor membrane extracts were incubated with [125I]-(Pyr1)-apelin-13 in the presence of various concentrations of apelin-13 or apelin-dm. (D) KD values ($n = 3$) of apelin, apelin-dm and apelin-dm/Elabela generated by plasmon waveguide resonance (PWR). (E) Effect of apelin-dm (100 nM) on ERK phosphorylation in HEK cells-expressing apelin receptor induced by Elabela (100 nM). (F, G) Phosphorylation of AKT (F) and ERK (G) analyzed by immunoblotting of HUVEC cells treated with apelin and apelin-dm. Data are representative of three independent experiments and data shown as the mean ± s.e.m. Significant differences $P$ determined by one-way ANOVA with Tukey's multiple comparison tests. (H, I) Heatmaps showing log2-transformed signal intensities for 196 PTK (H) peptide substrates and 144 STK (I) peptide substrates phosphorylated of Control, apelin and apelin-dm-treated HEK cells-expressing apelin receptor. The signals were sorted from high (red) to low (blue) intensity/phosphorylation. (J, K) Changes in peptide phosphorylation in apelin and apelin-dm-treated HEK cells-expressing apelin receptor, analyzed by a two-group comparison depicted as a volcano plot (effect size <0: less phosphorylation in treated cells ($n = 3$); significance score (log2) > 1.3 indicates significant changes, dotted line). Statistical significance was determined using ANOVA followed by a post-hoc test. (L, M) Upstream kinase analysis of PTK (L) and STK (M) of control, apelin and apelin-dm-treated HEK cells-expressing apelin receptor showing the top 20 ranked kinases (normalized kinase statistic (log2) < 0: less kinase activity in treated cells; specificity score (log2) > 1.3; white to red bars: statistically significant changes). Source data are available online for this figure.

CDK9) exhibited weaker activity compared to apelin-treated cells, indicating repression by apelin-dm. Mean kinase statistics and mean kinase scores for branches and nodes in the phylogenetic tree of the human protein kinase family were also calculated. In addition, the top upstream kinases from the significantly altered PTK/STK peptides by apelin-dm were predicted (Appendix Fig. S12A–C).

## Therapeutic efficacy of apelin-dm peptide

To assess the tumor-suppressive potential of apelin-dm, mice with subcutaneously injected CT-26 and MC-38 cells were treated intraperitoneally with apelin-dm (30 mg/kg), apelin receptor antagonist ML221 or a saline vehicle control (Fig. 5A). Tumor growth, apoptosis, proliferation, and/or the angiogenic index of the developed tumors were first evaluated. Apelin-dm treatment significantly reduced tumor growth compared to controls, as observed in daily monitored tumor volumes for both mice tumor models (Fig. 5B,C). The anti-tumor effect of apelin-dm was comparable to the effect of the apelin receptor antagonist ML221 (Fig. 5D). The CD31 and KI-67 staining of tumor sections revealed decreased angiogenic (Fig. 5E,H–J) and proliferative indices (Fig. 5F,H,K,L) in apelin-dm-treated mice inoculated by CT-26 and MC-38 tumor cells, respectively. Immunohistochemistry analysis showed increased pro-apoptotic protein BIM levels in apelin-dm-treated mice tumors (Fig. 5G,M,N), indicating enhanced cell death. Co-staining for KI-67 and CD31 in CT-26 and MC-38 tumors (Fig. 5H) derived from apelin-dm-treatd mice showed that the reduced KI-67 staining was predominantly observed in the cancer cells within the tumors. Apelin-dm also attenuated tumor growth (Fig. 5O,P) and angiogenesis (Fig. 5O,Q) induced by cancer cells in the CAM assay, and reduced tumor blood flow in mice with subcutaneous HCT116 colon cancer tumors (Fig. 5R,S) and other cancer cell types (MDA-MB-231) (Fig. 5T). Apelin-dm's anti-angiogenic and anti-tumorigenic activity was comparable to bevacizumab, and no obvious toxicity in treated mice was observed (Appendix Fig. S13).

## Apelin peptide represses early event of colorectal liver metastasis

In liver colorectal metastasis, metastatic cancer cells trigger the adhesion molecule E-selectin expression in liver endothelial cells, facilitating tumor cell arrest and metastasis (Scamuffa et al, 2008; Khatib et al, 1999, 2005). We investigated apelin's role by exposing

human hepatic endothelial cells (LSEC) to 100 nM apelin (Fig. 6A). E-selectin mRNA levels increased at 2 h, peaked at 6 h, and remained elevated even after 8 h of apelin treatment. At the protein level, E-selectin expression peaked at 24 h, followed by a decrease at 32 h (Fig. 6B). Apelin also induced E-selectin expression in endothelial cells HMEC-1 and HUVEC (Fig. 6C,D). However, the dynamics of apelin's effect differed in these endothelial cells compared to LSECs. While E-selectin expression initially increased after 2 h of incubation with apelin, it subsequently declined but remained elevated compared to baseline levels after 6 h. This divergence may arise from inherent variations in the regulatory mechanisms governing E-selectin expression, suggesting potential differences in the regulatory pathways and signaling mechanisms among these cell types in response to apelin. Apelin-dm peptide inhibited apelin-induced E-selectin, as did the apelin receptor antagonist MM54 in LSECs (Fig. 6E,F). The effect of apelin-dm on E-selectin induction was also assessed in mice intrasplenically/portal-inoculated with CT-26 cells. While, liver colonillizing CT-26 cells induced E-selectin expression, apelin-dm significantly repressed hepatic E-selectin (Fig. 6G,H). MM54 also repressed E-selectin induction mediatd by cancer cells (Fig. 6I). Apelin-dm and MM54-treated groups exhibited reduced hepatic metastases formation two weeks post-treatment (Fig. 6J,K). Taken together, these results highlight apelin's role in liver colonization and apelin-dm's ability to repress these processes (Fig. 7).

## Pharmacological profile of apelin-dm peptide

The pharmacological profile of the apelin-DM peptide was evaluated through a comprehensive series of studies (Fig. 8A). Acute toxicity was first assessed in Balb/C mice at doses ranging from 50 to 1000 mg/kg, with no adverse effects observed (Fig. 8B; Appendix pharmacological study 1). The maximum tolerated dose (MTD) was found to be >1000 mg/kg via intraperitoneal administration (Fig. 8B; Appendix pharmacological study 1). Repeated administration at a dose of 300 mg/kg/day for 14 days showed no safety concerns, with hematology, blood chemistry, and histological analyses confirming its safety (Fig. 8B; Appendix pharmacological study 2). In vivo stability was tested, revealing rapid systemic circulation and detection in serum (0.65 ± 0.3 μM at 5 and 10 min) with a half-life of 20 min (Fig. 8B; Appendix pharmacological study 3). In-solution properties showed 54% and 26% protein binding for human and mouse plasma, the solubility of 187.3 μM in PBS at 2.10–4 M, and a half-life of 1121 min (>18 h) in human plasma (Fig. 8B; Appendix

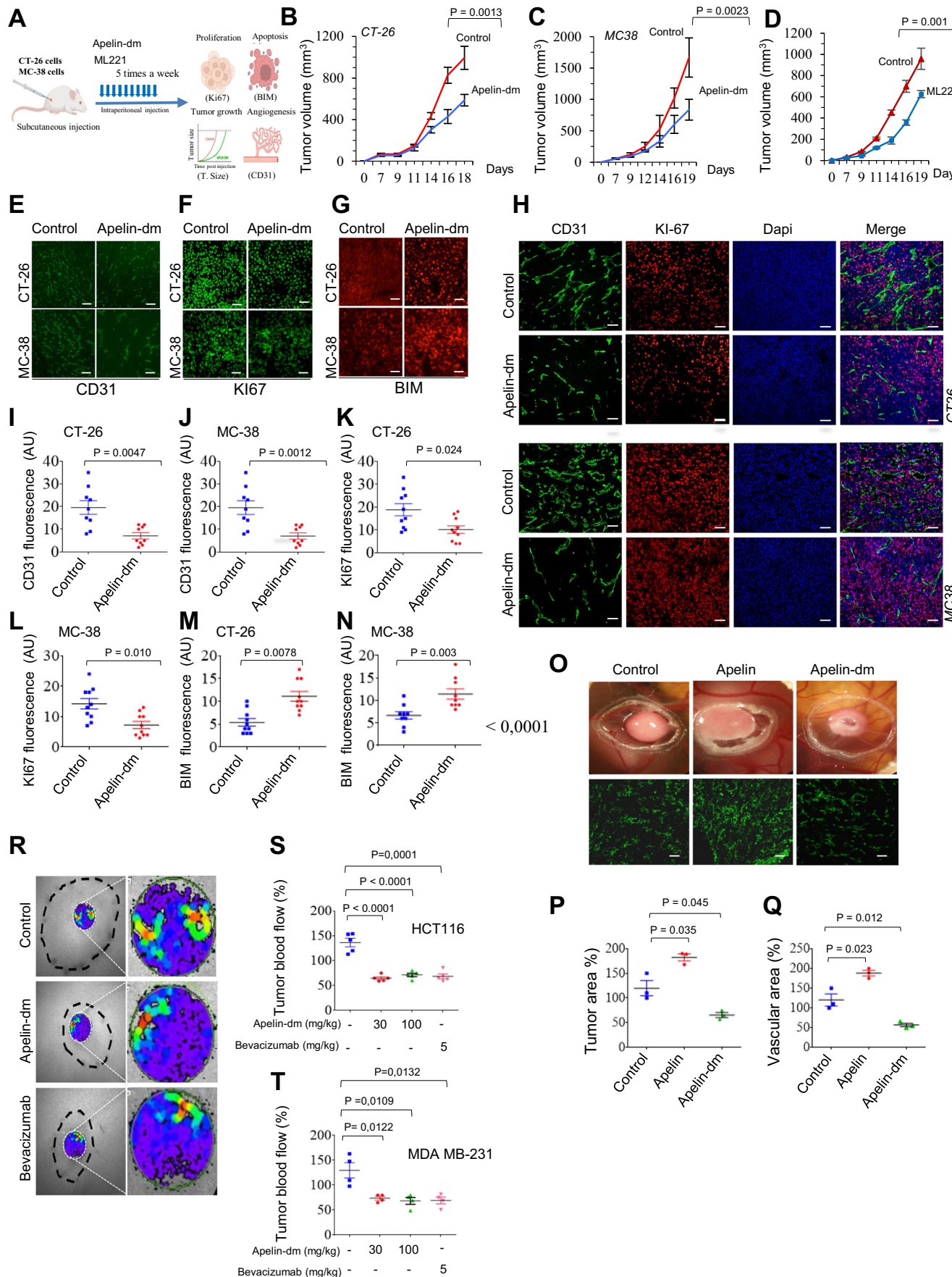

**Figure 5. Therapeutic efficacy of apelin-dm.**

(A) Scheme and workflow of apelin-dm preclinic studies. (B–D) Tumor growth analysis of subcutaneously injected mice ($n = 8$ tumors per group/experiment, three experiments) with CT-26 (B) and MC-38 (C) colon cancer cells treated with apelin-dm (30 mg/Kg), as shown in (A). The apelin receptor antgonoist ML221 (30 µg/kg), was used for comparaison (D). (B–D) Error bars indicate s.e.m. center values indicate mean (Mann–Whitney $U$ test). (E–H) Representative immunofluorescence staining of tumor sections derived from (B, C) shows CD31 (E, H green); KI-67 (F (green); H (red)); and BIM (apoptotic index, G, red). The percentage of relative staining intensity corresponding to CD31, Ki67, and BIM expression in CT-26 and MC-38 tumors of treated mice are shown in (I–N). Scale bar, 100 µm. Data represent mean ± s.e.m. of 3–4 independent experiments. $P$ values were calculated with the Student $t$ test in GraphPad. (O) Representative images of apelin and apelin-dm peptide effect on tumor growth and capillaries-treated CAMs implanted with CT-26 colon cancer cells ($n = 3$ CAM per group). Quantification of tumor area (P) and vascularized area (Q) relative to control untreated CAMs ($n = 3$ CAM per group) assigned 100%. Data represent mean ± s.e.m. $P$ values were calculated with the Student $t$ test in GraphPad. (R) color-coded images of blood flow analyzed by laser Doppler imager in control, apelin-dm (30 mg/Kg), and Bevacizumab (5 mg/Kg)-treated mice tumors. (S, T) The average blood flow expressed in % relative to blood flow measured in HCT113 (S) and MDA-MB231 (T) developed tumors. Data represent mean ± s.e.m. of 3–4 independent experiments. $P$ values were calculated with the Student $t$ test in GraphPad. Source data are available online for this figure.

pharmacological study 4). Intrinsic clearance in liver microsomes was <115.5 µL/min/mg (Fig. 8B, Appendix pharmacological study 4). Of the 16 CYP isoforms tested, apelin-dm inhibited CYP3A4 and induced 10.9%, 5.2%, and 5.0% inhibition in the hERG cardiac toxicity assay at concentrations of $10-5$ M, $10-6$ M, and $10-7$ M, respectively (Fig. 8B; Appendix pharmacological study 4). When compared to various reference compounds, used at the clinical setting in all these assays, apelin-dm emerges as a safe, stable, and specific potential drug candidate overall.

# Discussion

Metastasis stands as a primary contributor to mortality among patients grappling with diverse cancer types, notably colon cancer (Morgan et al, 2023; Buccafusca et al, 2019). The insidious dissemination of colon cancer can manifest in both the advanced stages of cancer progression and the pre-invasive phases, influenced by a spectrum of systemic factors secreted by the tumor or its microenvironment (Morgan et al, 2023; Buccafusca et al, 2019; Wang et al, 2023). Apelin, an emerging systemic factor, assumes a pivotal role in tumor advancement, angiogenesis, and the establishment of a premetastatic niche in potential metastatic organs through its intricate interaction with apelin receptor (Feng et al, 2016). Consequently, the early identification and prevention of metastasis, whether through traditional treatments or interventions in the apelin/ apelin receptor interaction, emerge as crucial strategies in mitigating cancer-related pathologies and fatalities. Within this study, we discerned apelin as a distinctive substrate of Furin, exhibiting resistance to cleavage by other proprotein convertases (PCs). Our findings furnish a mechanistic understanding of the anti-tumorigenic, antimetastatic, and anti-angiogenic effects attributed to apelin-dm, which arises from targeted modulation of the Furin cleavage site in the apelin precursor.

The dysregulation of apelin and its receptor, marked by overexpression and aberrant activation, plays a pivotal role in the onset and progression of numerous human malignancies, leading to reduced survival rates and compromised chemo-radiation sensitivity (Feng et al, 2016; Gourgue et al, 2020; Zuurbier et al, 2017; Bernier-Latmani et al, 2022a; Berta et al, 2010; Cazzato et al, 2015). In our study, we observed the co-expression of apelin and its receptor in both primary and metastatic tissues of colon cancer patients. This observation indicates that colon tumors sustain the capability to produce apelin and remain responsive not only in primary tumors, but also across various stages of tumor progression, including metastasis. This suggests that the autocrine loops mediated by apelin play a pivotal role not only in the growth and survival of primary tumors but also in facilitating the metastatic dissemination of cancer cells and angiogenesis. Previous studies have reported the expression of apelin and its receptor in endothelial cells (del Toro et al, 2010), smooth muscle cells (SMC) (Pitkin et al, 2010), and cancer cells (Williams et al, 2024). Comparative analysis of these protein expressions revealed higher levels of apelin and its receptor in endothelial and SMC cells compared to cancer cells. Similarly, Furin expression was also found to be higher in endothelial and SMC cells compared to CT-26 and MC-38 cancer cells. We also found that, like apelin, Furin is induced by hypoxia in endothelial cells.

Upon activation, apelin receptor triggers various pathways crucial to the growth, survival, and migration of both cancer and endothelial cells (Read et al, 2019; Falcão-Pires et al, 2010; Picault et al, 2014a). Consequently, our focus in this study centered on unraveling the impact of apelin-dm on these critical processes within colon cancer and endothelial cells. Through a series of in vitro and in vivo models simulating tumor progression and colorectal liver metastasis via intrasplenic injection of colon CT-26 and MC-38 cancer cells expressing apelin-dm, we unearthed compelling evidence of the anti-cancer and anti-angiogenic properties of apelin-dm. Notably, similar effects were replicated using purified apelin-dm peptide. This peptide not only hindered angiogenesis driven by apelin but also curtailed VEGF-stimulated angiogenesis by repressing VEGF-induced proliferation, migration, and signaling pathways. In addition, apelin and apelin receptor expression was reported to be increased in endothelial and non-endothelial cells following VEGF stimulation (Kojima and Quertermous, 2008; Takano et al, 2018), suggesting VEGF's involvement in the enhanced autocrine action of apelin in endothelial cells. This implies that apelin-dm can repress VEGF-induced proliferation and migration through this autocrine action. We also found that apelin-dm interferes with the interaction between Elabela and the apelin receptor and represses Elabela-induced receptor activation. Apelin-dm exhibited also a pronounced inhibitory effect on colony formation and migration of colon cancer cells, endothelial cells, and smooth muscle cells. These observations provide plausible mechanistic insights into how this inhibitor may effectively impede tumor growth and angiogenesis.

Our investigation unveiled that the inhibition of proapelin processing appears to heighten the affinity of the apelin receptor for apelin-dm. This observation was substantiated through comprehensive docking computations and structural analyses, which were further validated by radioligand binding assays. Constructing the

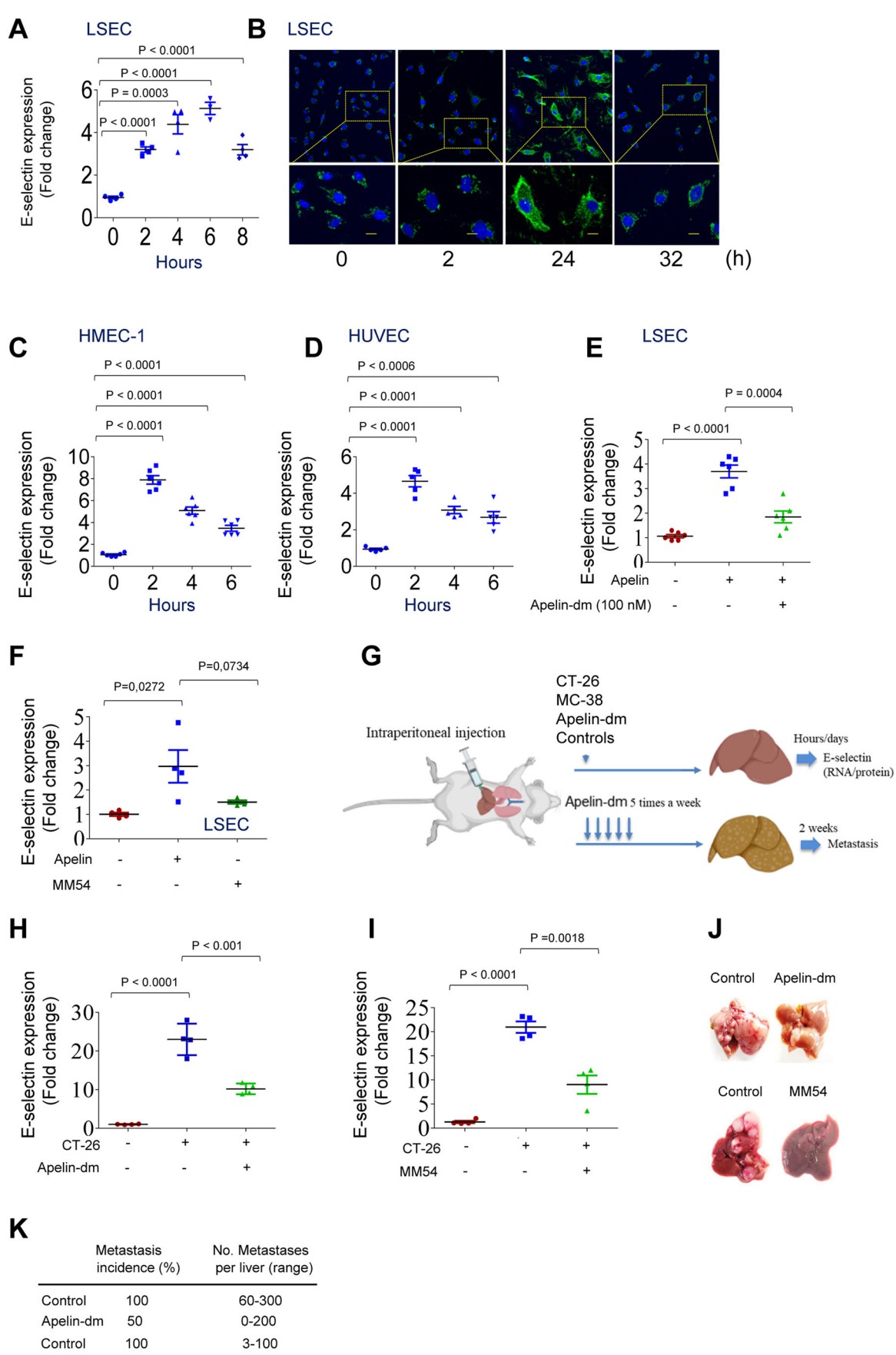

**Figure 6.  Apelin-dm peptide represses early events of colorectal liver metastasis.**

RT-PCR and immunostaining analyses of E-selectin expression in human liver sinusoidal endothelial cells (LSEC) (**A, B**), HMEC-1 (**C**), and HUVEC (**D**) treated with 100 nM each of apelin, apelin-dm (**E**), and/or MM54 (**F**) ($n = 3$ per group, three independent experiments). Values represent the mean ± s.e.m., and significant differences (*P*) were determined by two-way ANOVA. (**G**) Scheme and workflow of apelin-dm effect analysis on E-selectin expression and metastasis induced by colon cancer cells CT-26 and MC-38 ($10^6$). (**H, I**) Mice were injected through the intrasplenic/portal route with $10^6$ CT-26 cells and treated with apelin-dm (**H**) or MM54 (**I**) (30 mg/Kg), as shown in (**G**). Their livers were then removed for E-selectin mRNA expression analysis ($n = 6$ per group). The results shown are representative of three experiments. Values are mean ± s.e.m. *P* values were calculated using the Student's *t* test in GraphPad. (**J**) Representative images of livers from mice ($n = 7$ tumors per group/experiment) injected intrasplenically with control CT-26 treated with apelin-dm and MM54 (30 mg/Kg), as shown in (**G**). The number and incidence of liver nodules 2 weeks after tumor cell injection are indicated (**K**). Scale bar for (**B**), 100 μm. Source data are available online for this figure.

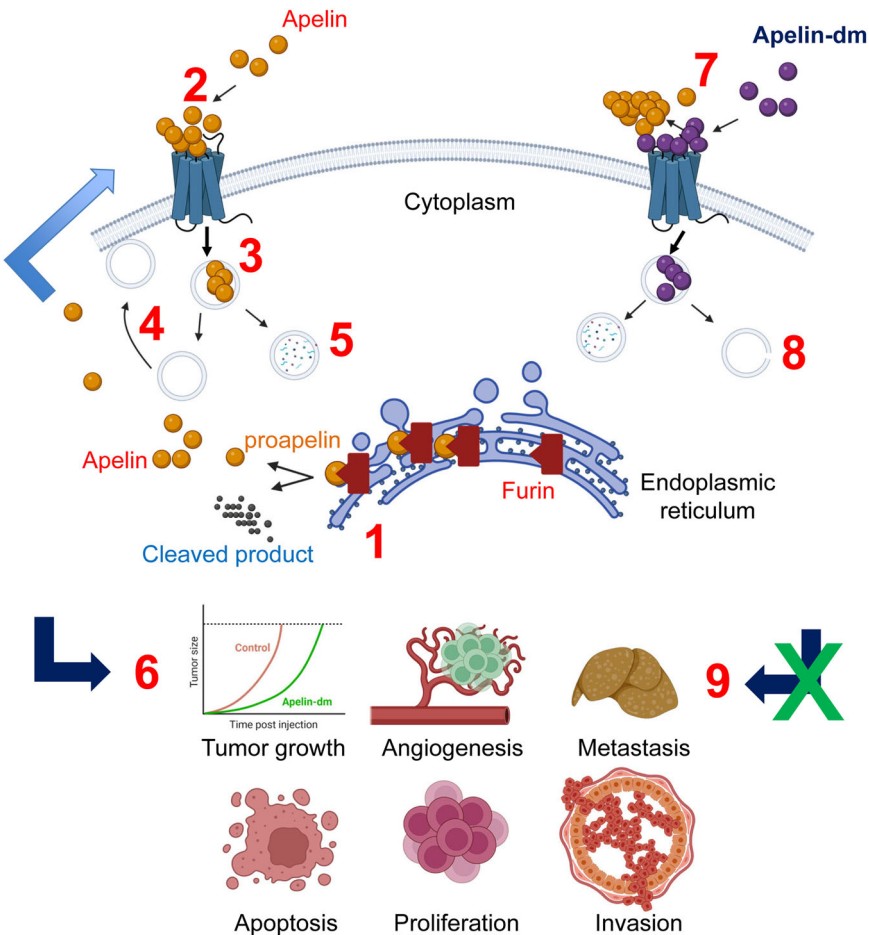

**Figure 7.  A model depicting apelin-dm interaction with the apelin receptor, leading to the repression of colon tumor angiogenesis and metastasis.**

Proapelin cleavage by Furin, which may occur at the tumor cell surface or in the Trans-Golgi Network (1), generates secreted mature apelin that activates the apelin receptor (2), inducing its internalization (3) and recycling (4) or degradation (5), leading to tumor growth, angiogenesis, survival, invasion, and metastasis (6). In contrast, apelin-dm interferes with the interaction and signaling between apelin and the apelin receptor (7), delaying or preventing apelin receptor recycling at the cell surface (8) and inhibiting these processes (9). All model diagrams were created using BioRender.com.

apelin-dm-apelin receptor complex based on the X-ray structure of the apelin receptor-apelin peptide mimetic AMG3054 revealed a distinctive wrapping of apelin-dm (or apelin-36) around the apelin receptor, with a notable concentration of major binding energy in the C-terminal region of the peptide, particularly at the last Phe residue. Intriguingly, the residues cleaved by Furin exhibited a tendency to point away from the receptor, forming limited contact points with the protein. The predicted binding scores, favoring apelin-dm (and apelin-36) over the shorter apelin-13, aligned with experimental data, supporting the additional favorable contacts for the longer 36-residue peptides. The predicted interaction energy values are relatively similar between apelin-36 and apelin-dm, but apelin-dm is more stable and not cleaved as compared to the wild-type apelin-36, which is rapidly cleaved upon synthesis. This allows

**A**

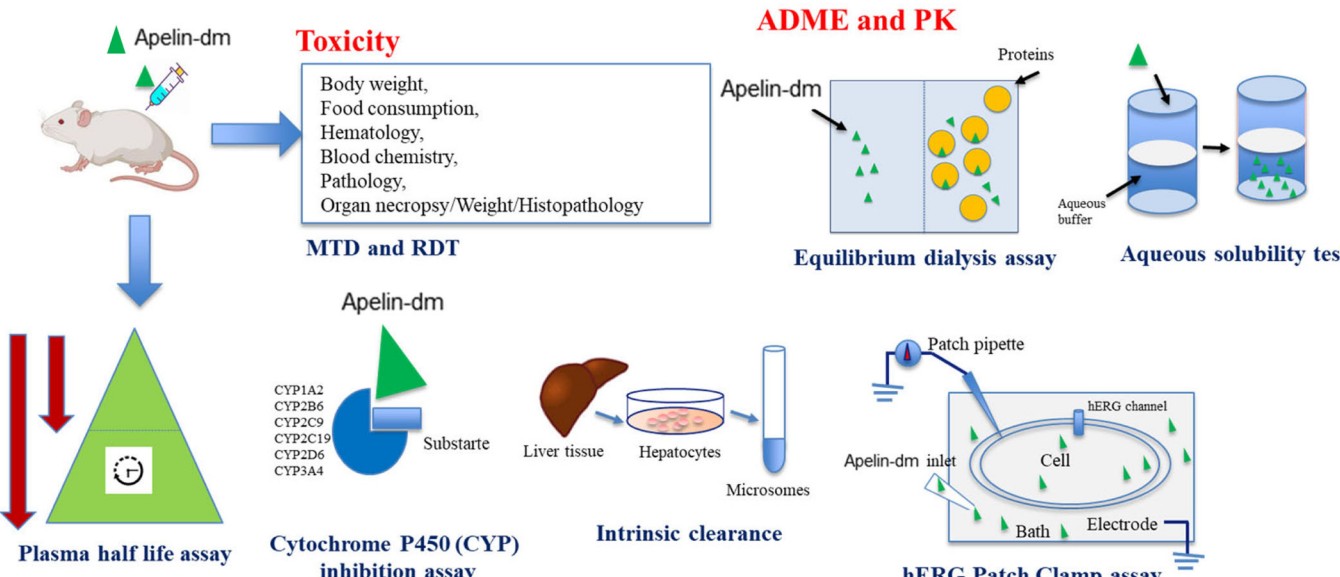

**B**

| Pharmacological evaluation | Calculated values | Detailed study in: |
| --- | --- | --- |
| **MTD**<br>  Acute toxicity<br>  Repeated administrations | >1000 mg/kg<br>300 mg/kg/day for 14 days | Appendix pharmacological study 1<br>Appendix pharmacological study 2 |
| **Half-life**<br>*In vivo* in Mice<br>In human plasma | 20 min<br>1121 min | Appendix pharmacological study 3 |
| **Protein bound assay ($10^{-5}$ M )**<br>In human plasma<br>In mice plasma | 54%<br>26% | Appendix pharmacological study 4 |
| **Solubility ($10^{-4}$ M )** | 187.3 µM | Appendix pharmacological study 4 |
| **Intrinsic clearance ($10^{-7}$M)** | <115.5µL/min/mg | Appendix pharmacological study 4 |
| **CYP inhibition** | CYP3A4 isoform | Appendix pharmacological study 4 |
| **Inhibition of the hERG current**<br>$10^{-5}$ M<br>$10^{-6}$ M<br>$10^{-7}$ M | 10.9%<br>5.2%<br>5.0% | Appendix pharmacological study 4 |

**Figure 8.  ADME, pharmacokinetics, and toxicity analysis of apln-dm.**

Workflow (**A**) and analysis (**B**) of the Absorption, Distribution, Metabolism, Excretion (ADME), Pharmacokinetics (PK) and Toxicity of apelin-dm (details are provided in Appendix pharmacological study 1–4). MTD The maximum tolerated dose.

for major competition on the apelin receptor to occur between mature apelin rather than apelin-36. The binding pocket crucial for interaction with the C-terminal region of apelin was predicted to be allosteric, corroborating with observed flexibility and structural changes in this region of the receptor. While apelin-13 was predicted to engage two allosteric sites, apelin-dm (or apelin-36) was anticipated to establish further contacts with an additional allosteric site situated on the receptor's side. This suggests the possibility that the two peptides (e.g., apelin-13 and apelin-dm) induce slightly different structural changes upon binding. The precise 3D predictions of such changes remain challenging, acknowledging the inherent flexibility of the N-terminal of apelin-dm, yet the overall position of the peptides on the apelin receptor seems reasonable. It is important to note that the exact high atomic resolution of apelin-dm remains unknown. Utilizing a plasmon waveguide resonance (PWR) assay, we discerned that apelin-dm and apelin-13 elicited distinct conformational changes in the receptor, indicative of potential differences in apelin receptor affinity and in the likely interactions with the predicted allosteric sites for apelin-13 versus apelin-dm.

Furthermore, our findings unveiled that, while apelin triggered the phosphorylation of various kinases associated with cell proliferation, survival, and migration encompassing members of the Src, InsR, ERK, and CDK family kinases apelin-dm exhibited a diminished effect. Notably, apelin-dm not only attenuated the stimulatory effects of apelin but also inhibited the ability of apelin to induce various kinase activities. This divergence in kinase activation profiles further emphasizes the distinct functional impact of apelin-dm compared to its processed peptide apelin.

Protein half-life exhibits significant variability spanning several orders of magnitude (Toyama and Hetzer, 2013) with longer-lived proteins typically serving as "housekeeping" proteins, while shorter-lived proteins often function as phosphorylation targets and signaling molecules. apelin, a key player in physiological processes, has a notably short in vivo half-life (<5 min) (Vickers et al, 2002), implicating its rapid turnover and degradation. Initial studies proposed the involvement of metalloproteases, specifically angiotensin-converting enzyme II (ACE2) and proline carboxy peptidase (PRCP), in apelin degradation, particularly targeting the Pro12-Phe13 residues (Kehoe et al, 2016). ACE2, in particular, efficiently hydrolyzes apelin-13, apelin-36, and apelin-36 (Vickers et al, 2002). Interestingly, the generated fragments from ACE2 cleavage did not induce apelin receptor internalization in vitro, suggesting that these cleavages may induce a distinct conformational state of the apelin receptor compared to full-length peptides (El Messari et al, 2004b). Another identified cleavage site between Leu5-Ser6 residues was attributed to neprilysin (NEP) (McKinnie et al, 2016). Our study revealed that modifying the cleavage site of the apelin precursor significantly increased its half-life. This implies that potential conformational changes introduced by cleavage site mutations may prevent the degradation of apelin-dsm by proteases such as ACE2 and NEP. Ligand–receptor

dynamics often involve trafficking to lysosomes upon binding, where the ligand is released and degraded while the receptor undergoes recycling or complex degradation (Bonecchi et al, 2008). The extended half-life of certain receptors, previously associated with "slow" recycling between endosomes and the cell surface, has been reported for various receptors, including decoy receptors and membrane-localized vascular endothelial growth factor receptor-1 (mVEGFR1) (Boucher et al, 2017). The stability and trafficking of mVEGFR1 are regulated by the palmitoylating enzyme DHHC3 and depalmitoylating enzyme APT1. Impaired mVEGFR1 palmitoylation results in increased trafficking to lysosomes for degradation, disrupting vascular morphogenesis. Our findings highlight the impact of ligand half-life in delaying receptor recycling and trafficking upon binding, subsequently altering receptor function and leading to reduced angiogenesis and tumor progression.

Our investigation into the pharmacokinetic properties of apelin-dm, which revealed ~50% plasma protein binding, presents several advantages that warrant consideration in therapeutic contexts. These binding properties strike a balance between a rapid onset of action and a prolonged duration of therapeutic effect. While a portion of the drug is bound to plasma proteins, ensuring a reservoir for sustained release, an equally substantial fraction remains unbound and pharmacologically active. This equilibrium allows for the swift attainment of therapeutic concentrations in the bloodstream while maintaining a steady supply of bioavailable drug molecules over an extended period. Consequently, apelin-dm may offer a favorable pharmacokinetic profile suitable for conditions requiring both immediate intervention and sustained therapeutic coverage. Highly protein-bound drugs may be susceptible to drug interactions and alterations in pharmacokinetics due to competition for plasma protein binding sites, potentially leading to suboptimal therapeutic outcomes or an increased risk of toxicity. Conversely, drugs with minimal protein binding may exhibit rapid clearance and reduced bioavailability, necessitating frequent dosing regimens or higher doses to maintain therapeutic efficacy. Apelin-dm's 50% plasma protein binding confers a degree of flexibility, striking a harmonious balance between efficacy and safety, and minimizing the likelihood of pharmacokinetic fluctuations that could compromise treatment efficacy. Previously, peptides with similar properties have been well tolerated with few side effects, suggesting that apelin-dm is likely to be translatable into clinical practice (Davenport et al, 2020).

We elucidate the crucial role of apelin precursor cleavage by Furin and identify apelin-dm as a peptide with anti-tumorigenic and antimetastatic functions. Importantly, this peptide exhibits favorable pharmacological profiles. Overall, we present apelin-dm as a potent, selective, non-toxic inhibitor of the apelin receptor interaction, demonstrating efficacy in a mouse model of colon cancer and colorectal liver metastasis. Our data strongly support apelin-dm as a promising lead compound for further exploration in medicinal chemistry and drug development.

# Methods

### Reagents and tools table

| Reagent/resource | Reference or source | Identifier or catalog number |
|---|---|---|
| **Experimental models** | | |
| Human Peripheral Blood Mononuclear Cells (hPBMCs) | Institute Bergonié, Bordeaux or EFS, Bordeaux | NA |
| Colorectal cancer patients' sample | Institute Bergonié, Bordeaux | NA |
| HT-29 cells (*H. sapiens*) | ATCC | HTB-38 |
| LoVo cells (*H. sapiens*) | ATCC | CCL-229 |
| HEK-293 cells (*H. sapiens*) | ATCC | CRL-1573 |
| U2OS cells (*H. sapiens*) | ATCC | HTB-96 |
| MDA-MB-231 cells (*H. sapiens*) | ATCC | HTB-26 |
| MC-38 cells (*M. musculus*) | Kerafast | CVCL_B288 |
| CT-26 cells (*M. musculus*) | ATCC | CRL-2638 |
| CHO-FD11 cells (*C. griseus*) | Gordon et al, 1997 | NA |
| C57BL6/J (*M. musculus*) | Charles Rivers Laboratories | N/A |
| Rag2/γc (*M. musculus*) | The Jackson Lab | NA |
| BALB/c (*M. musculus*) | Charles Rivers Laboratories | NA |
| **Recombinant DNA** | | |
| pIRES-APLN | This study | NA |
| pIRES-APLN-V5 | This study | NA |
| pIRES- APLN | This study | NA |
| pIRES-apelin-dm | This study | NA |
| APLNR-pEGFP-N1 | This study | NA |
| **Antibodies** | | |
| Anti-apelin | Eurogentec | NA |
| Anti-apelin receptor | Abcam | ab84296 |
| Anti-Ki67 | Cell Signaling | 9129T |
| Anti-c-caspase 3 | Cell Signaling | 96617 |
| Anti-CD31 | BD PharmingenTM | 553370 |
| Anti-phospho ERK | Cell Signaling | 9106S |
| Anti-ERK | Cell Signaling | 4695 |
| Anti-β-actin | Cell Signaling | 4967 |
| Anti–E-selectin | Abcam | ab2497 |
| Anti- human Clathrin | Abcam | ab2731 |
| Anti-V5 | Invitrogen | NA |
| Secondary anti-rabbit | Cell Signaling | 7074 |
| Secondary anti-mouse | Cell Signaling | 7076 |
| **Chemicals, enzymes, and other reagents** | | |
| Apelin peptides | Clinisciences | NA |
| pERTKR-AMC Fluorogenic Peptide Substrate | R&D system | ES013 |
| M-PER lysis buffer | Thermo Fisher Scientific | 78503 |
| Halt Protease and Phosphatase inhibitors | Thermo Fisher Scientific | 87786 and 78420 |
| STK reagent kit | PamGene | 32201 |
| PTK reagent kit | PamGene | 32112 |
| Bovine serum albumin (BSA) | Euromedex | 04-100-812-C |
| [125I]-MIP-3b | Perkin Elmer | NA |

| Reagent/resource | Reference or source | Identifier or catalog number |
|---|---|---|
| MIP-3b | R&D Systems | 361MI |
| FITC Annexin V Apoptosis Detection Kit I | BD PharmingenTM | NA |
| **Software** | | |
| Protein identification | Proteome Discoverer 1.4 Mascot 2.5 | NA |
| | Sequest HT | NA |
| | Percolator algorithm | NA |
| Software Docking | CABS-dock flexible docking engine | Kurcinski et al, 2019 |
| | PULCHRA | Rotkiewicz and Skolnick, 2008 |
| | MMTK | http://dirac.cnrs-orleans.fr/MMTK.html |
| | pyDockEneRes | Romero-Durana et al, 2020 |
| | PPM server | Lomize et al, 2012 |
| | APOP | Kumar et al, 2023 |
| | Fpocket | Le Guilloux et al, 2009 |
| | UCSF ChimeraX | Pettersen et al, 2021 |
| | PyMOL | https://www.schrodinger.com |
| ImageJ/Fiji | https://imagej.nih.gov/ij | NA |
| BioNavigator V63 | PamGene | BN63 |
| **Primers** | | This study |
| Mutant | Sense | Primer sequence 5' > 3' |
| pIRES-APLN-M1 | Sense | 5'-GGCAGGGAGGTTCGAGTAAATTCCGCCGCCAGCGGCCC-3' |
| | Antisense | 5'-GGGCCGCTGGCGGCGGAATTTACTCGAACCTCCCTGCC-3' |
| pIRES- APLN-M2 | Sense | 5'-GGCAGGGAGGTCGGAGGAAATTCAGCAGCCAGCGGCCC-3' |
| | Antisense | 5'-GGGCCGCTGGCTGCTGAATTTCCTCCGACCTCCCTGCC-3' |
| pIRES-APLN-DM | Sense | 5'-GGCAGGGAGGTTCGAGTAAATTCAGCAGCCAGCGGCCC-3' |
| | Antisense | 5'-GGGCCGCTGGCTGCTGAATTTACTCGAACCTCCCTGCC-3' |

| Species | Gene | Sens | Primer sequence 5' > 3' |
|---|---|---|---|
| Human | Furin | Sense | 5'-GCCCAGAATTGGACCACAGT-3' |
| | | Antisense | 5'-TCCCGATGTCTTTGGGCTC-3' |
| | | Probe | 5'-CAGCGGAAGTGCATCATCGACATCC-3' |
| | APLN | Commercial | Hs00936329_m1 (Thermo Fisher Scientific) |
| | APLNR | Sense | 5'-TGACTTTGACCTCTTCCTCATGAAC-3' |
| | | Antisense | 5'-GGGTTGAGGCAGCTGTTGAC-3' |
| | | Probe | 5'-TCTTCCCCTACTGCACCTGCATCAGC-3' |
| | GAPDH | Sense | 5'-CAAATTCCATGGCACCGTC-3' |
| | | Antisense | 5'-CCCACTTGATTTTGGAGGGA-3' |
| | | Probe | 5'-CCCATCACCATCTTCCAGGAGCGAG-3' |
| Mouse | Furin | Sense | 5'-TGAGCCATTCGTATGGCTACG-3' |
| | | Antisense | 5'-TGCGCACCTCTAGCCGTT-3' |
| | | Probe | 5'-TGGTGGAACCCAAGGACATCGGC-3' |
| | APLN | Sense | 5'-CCACTGATGTTGCCTCCAGAT-3' |
| | | Antisense | 5'-TCACCAGGTAGCGCATGCT-3' |
| | APLNR | Sense | 5'-GCATGCCTGGAAGGACTCTAA-3' |
| | | Antisense | 5'-GGATTGGCTTGAACCTCAGAGTA-3' |
| | HPRT1 | Commercial | Mm01545399_m1 (Thermo Fisher Scientific) |
| **Other** | | | |
| BD AccuriTM C6 Cytometer | | | |
| Nikon C2si Eclipse Ti-S | | | |
| Tecan Infinite® F200 PRO, Tecan Group Ltd. | | | |

## Patient samples

Samples from 35 colorectal cancer patients, along with corresponding liver metastatic tissues and adjacent normal regions, were collected from frozen and formalin-fixed paraffin-embedded (FFPE) tissues. The specimens were clinically and histopathologically diagnosed at Institut Bergonié, Bordeaux. The tissues, acquired after resection, were promptly placed on ice post-surgery and snap-frozen in liquid nitrogen for subsequent analysis. All human specimens were obtained following written informed consent approved by Bergonié Institute, Bordeaux, France. Patient consent forms for all samples were obtained at the time of tissue acquisition. Biopsies were de-identified. All the performed experiments are conformed to the principles set out in the WMA Declaration of Helsinki and the Department of Health and Human Services Belmont Report.

## Mice

The mouse strains used in this study included C57BL/6, BALB/c, and Rag2/γc, all obtained from the Jackson Laboratory. Both male and female mice, aged 6–8 weeks, were used in the experiments, with age matching applied wherever possible. All animal procedures were reviewed and approved by the Institutional Animal Care and Use Committees (IACUC) of the University of Bordeaux and were conducted under license V7#10362. Mice were maintained under specific pathogen-free (SPF) conditions, with a 12-h light/dark cycle, and provided with ad libitum access to standard chow and water. To ensure environmental enrichment and support animal welfare, cages were supplemented with chew sticks, handling tubes, and domed houses. All experiments complied with the ARRIVE guidelines, as applicable.

## Analysis of public datasets

The comparison between colon tumor and normal tissues regarding *GSN, Spg20, NSUN5, APLN, APLNR* mRNA expression was performed using GEPIA (Tang et al, 2017). The correlation between CYR6, APLN, and APLNR mRNA expression, and RIC8A, APLN, and APLNR mRNA expression in colon adenocarcinoma (COAD) was also determined using GEPIA.

## Cloning and mutagenesis

pIRES-APLN and pIRES-APLN-V5 constructs were generated by amplifying the coding region of human apelin by PCR using specific primers (Appendix Reagents and Tools Table). Mutagenesis was carried out by an oligonucleotide-directed mutagenesis system (Quick Change site-directed mutagenesis kit, Stratagene) according to the manufacturer's recommendation using pIRES-APLN construct. The oligonucleotides used to generate the mutations for pIRES-APLN-DM generation are listed in "Appendix Reagents and Tools Table". APLNR cDNA from human colorectal cancer tissue was amplified using PCR and specific primers and the PCR product was subcloned into the pEGFP-N1 vector to generate the APLNR-pEGFP-N1 vector. All constructs were verified by sequencing.

## Cell transfection, transduction, and culture

LoVo, HT-29, CT-26, MC-38, U2OS, and MDA-MB-231 cells were procured from ATCC (Manassas, VA, USA). CHO-FD11 cells with

reduced Furin activity were characterized previously (Gordon et al, 1997). All cells were maintained in modified Eagle's medium (MEM), Dulbecco's modified Eagle's medium (DMEM) or RPMI-1640 with 10% fetal calf serum, 100 units/ml penicillin, 100 µg/ml streptomycin, and 2 mM L-Glutamine (Dutscher), and incubated at 37 °C with 5% $CO_2$. Authenticity and mycoplasma screening were routinely performed by PCR (every two weeks and before each animal experiment for mycoplasma screening). For hypoxic experiments, endothelial cells were serum-starved and placed in a sealed, humidified chamber maintained at 1% $O_2$ and 5% $CO_2$ for different time periods ranging from 4 to 24 h.

Transfections used Effectene transfection reagent (Qiagen, Germany). Lentiviral vectors with wild-type APLN or APLN-DM cDNA were prepared using pIRES plasmids, cloned into a self-inactivating lentiviral vector with a tdTomato reporter gene (pRRLsin-MND-hPGK-tdTomato-WPRE) under myeloproliferative sarcoma virus enhancer control. All constructs were sequenced, and lentiviral vector production was conducted by the "Vect'UB" service platform at the TMB-Core of the University of Bordeaux.

## Ribonucleic acid (RNA) extraction, real-time PCR and droplet digital PCR (ddPCR)

For real-time PCR, total RNA was extracted using an RNA isolation kit (Macherey-Nagel) including DNase treatment (Qiagen), according to the manufacturer's instructions, and qPCR data were acquired with the StepOnePlusTM Real-Time PCR System (Applied Biosystems, Courtaboeuf, France), as previously described (Scamuffa et al, 2008). For ddPCR, cDNA was synthesized from 2 µg of total RNA using Maxima Reverse Transcriptase (Fisher Scientific) and primed with oligo-dT primers (Fisher Scientific) and random primers (Fisher Scientific). PCR were performed in a final volume of 22 µl containing the required QX200 ddPCR EvaGreen Supermix (Bio-Rad) with a final concentration of 150 nM of each primer sets and 2 µl cDNA equivalent to 4 ng total RNA input. Each ddPCR assay mixture (22 µl) was loaded into a disposable droplet generator cartridge (Bio-Rad). Then, 70 ml of droplet generation oil (Bio-Rad) was distributed into each of the eight oil wells of the cartridge. The cartridge was then placed inside the QX200 droplet generator (Bio-Rad). When droplet generation was completed, the droplets were transferred to a 96-well semi-skirted ddPCR plate (Bio-Rad) heatsealed with foil in a PX1 PCR Plate Sealer (Bio-Rad) and amplified with a Mastercycler Nexus Gradient Thermal Cycler (Eppendorf). Thermal cycling conditions for EvaGreen assays were as follows: 95 °C for 5 min, followed by 45 cycles of 95 °C for 30 s and 61 °C for 1 min, followed by 4 °C 5 min and a final inactivation step at 90 °C for 5 min. A no-template control and a negative control for each reverse transcription reaction were included in the assay. After thermal cycling the sealed plate was placed in the QX200 Droplet Reader (Bio-Rad) for data acquisition. The resulting data was analyzed using QuantaSoft Software (version 1.7; Bio-Rad).

## Production of apelin-13, apelin-36 and apelin-dm peptides

Peptides apelin-36 (LVQPRGSRNGPGPWQGGRRKFRRQRPR LSHKGPMPF), apelin-13 (QRPRLSHKGPMPF) and apelin-dm (LVQPRGSRNGPGPWQGGSSKFSSQRPRLSHKGPMPF) were synthesized by Neo Biotech (Clinisciences, France) and

resuspended in water or PBS to a 1–5 mM working solution. Peptides were added directly into the medium of the cells or injected into mice and fertilized eggs for the chick chorioallantoic membrane assay at the indicated concentrations.

## Immunoblotting

Western blot analysis was conducted as previously described (Scamuffa et al, 2008), with modifications to detect various apelin isoforms. After their respective treatments, cells were lysed at 4 °C with lysis buffer (20 mM HEPES at pH 7.5, 150 mM NaCl, 0.5% Triton X-100, 6 mM β-octylglucoside, 10 μg/mL aprotinin, 20 μM leupeptin, 1 mM NaF, 1 mM DTT, and 100 μM sodium orthovanadate). Lysates were loaded on a 13.3% SDS-PAGE acrylamide-glycerol gel, followed by transfer onto a nitrocellulose membrane (Protran 0.2 μm, Amersham). The membrane was fixed with 2.5% glutaraldehyde (Sigma G7651). Membranes were incubated with indicated primary antibodies that were revealed by HRP-conjugated secondary antibodies (Amersham Pharmacia Biotech) and enhanced chemiluminescence (ECL Plus, Amersham Pharmacia Biotech) according to the manufacturer's instructions. After imaging, band quantification was performed using ImageJ software (NIH), and protein levels were normalized to actin.

## Measurement of PCs activity

The PCs activity in cells and media was assessed by the evaluation of the enzymes ability to digest the universal PC substrate, the fluorogenic peptide pERTKR-MCA as described previously (Sfaxi et al, 2014; Scamuffa et al, 2014). Extracts were incubated with pERTKR-MCA (100 μM) during the indicated time periods in the presence of 25 mM Tris (pH 7.4), 25 mM methyl-ethane-sulfonic acid, and 2.5 mM CaCl$_2$, at 37 °C, and the fluorometric measurements were performed using a spectrofluorometer (Tecan Infinite® F200 PRO, Tecan Group Ltd, France).

## Internalization assay

U2OS cells stably co-expressing human apelin receptor and β-arrestin 2-GFP were seeded on poly D-lysine-coated glass slides in 12-well dishes ($3 \times 10^5$ cells/well). After 12 h, the medium was replaced with fresh medium containing apelin-(5-carboxytetra-methylrhodamine: apelin-13-TAMRA (100 nM) (Proteogenix, Oberhausbergen, France), apelin-dm, or apelin-13-TAMRA and apelin-dm (100 nM) for 45 min. In another experiment, these cells were treated with apelin (100 nM) or apelin-dm (100 nM) and stained with mouse anti-human clathrin heavy chain antibody (Abcam). For washout experiments, cells were incubated for 30 min with apelin peptides (100 nM), replaced with serum-free medium, and fixed 2 h after washout. Analysis was performed using a Zeiss LSM-780 confocal microscope.

## Bioluminescence resonance energy transfer (BRET) measurement

For the BRET assays, phenol red-free medium was removed from HEK293T cells transiently transfected with apelin receptor-Rluc and Rab5-YFP, and replaced with PBS containing calcium and magnesium. The assay was initiated by adding 10 μl of the cell-permeant Renilla luciferase substrate, coelenterazine h, to achieve a final concentration of 5 μM. Five minutes later, apelin and apelin-dm were added to assess their activity. Plate readings were taken 15 min after substrate addition. BRET signals were collected using a Mithras LB940 instrument, which integrates signals sequentially in the 465–505 nm and 515–555 nm windows using appropriate bandpass filters, managed by MicroWin 2000 software. Net BRET signals were calculated by subtracting the BRET signal from cells expressing only Rluc-tagged apelin receptor from those co-expressing both Rluc-tagged apelin receptor and YFP-tagged Rab5.

## Radioligand binding assay

Radioligand binding assay (Euroscreen catalog: FAST-020B) was performed in 96-well plates (Master Block, Greiner, 786201) containing apelin receptor membrane extracts (2 μg protein/well), 0.02 nM [125I]-MIP-3b (Perkin Elmer, custom product) and apelin-13 or apelin-dm at various concentrations. Nonspecific binding was determined using a 200-fold excess of MIP-3b (R&D Systems, 361MI). Dose-response curves were established and the estimated IC$_{50}$ values were calculated for each compound.

## Soft agar, proliferation, and apoptosis assays

Anchorage-independent colony formation assay was performed as previously described (Sfaxi et al, 2014; Scamuffa et al, 2014). Proliferation assays were performed using an IncuCyte live-cell microscopy incubator (Essen Bioscience) as previously described (Soulet et al, 2020). For the apoptosis assay, tumor cells were grown to 70% confluency, washed repeatedly to remove serum, and then incubated for the indicated time periods in serum-free media. Cells were washed and stained with FITC-labeled Annexin V using the FITC Annexin V Apoptosis Detection Kit I (BD Pharmingen™), according to the manufacturer's instructions. Cells were analyzed by flow cytometry (BD Accuri™ C6 Cytometer), as previously described (Scamuffa et al, 2014).

## Tube-like formation assay

HUVECs were seeded at a density of $25 \times 10^3$ cells/well onto chamber slides (Labtek) previously coated with Geltrex (Gibco) at 37 °C for 30 min to allow polymerization. Appropriate medium-containing vehicles, apelin, apelin-dm or bevacizumab were added at various concentrations. The plates were examined for tube formation under an inverted light microscope at various time points. Photographs of the tubular network in the wells were captured using a digital camera attached to an inverted microscope.

## Aortic ring assay

Six-week-old C57Bl/6 male mice were anesthetized with isoflurane for 5 min before being sacrificed. The descending aorta was isolated, cleared of adventitia, and placed in serum-free Opti-MEM (Gibco) supplemented with 1× antibiotic/antimycotic (Gibco, 100× stock solution). The aortas were sectioned into ∼25 rings of 0.5 mm (about 15–20 per aorta) thickness and placed in fresh serum-free Opti-MEM with 1× antibiotic/antimycotic for 1 h at 37 °C. The embedded rings

were incubated with apelin (100 nM), apelin-dm (100 nM), Bevacizumab (100 nM), and/or VEGF (30 ng/ml). For each ring, the microvessels emerging from the main ring and individual branches arising from them were quantified.

## Chick chorioallantoic membrane (CAM) angiogenesis assay

Fertilized eggs were allowed to mature ex ovo. A small incision was made in the eggshell on day 3 of development, and seven days later, apelin and/or apelin-dm peptide was added to the CAM tissue (100 nM). In other experiments, $1 \times 10^6$ control tumor cells or the same cells stably expressing wild-type APLN or APLN-DM were directly allowed to grow on the CAM tissue. Immunohistology in toto was performed for vessel staining. Images of CAM vessels were acquired using a Zeiss Axiophot microscope.

## Retina angiogenesis assay

Postnatal Day 5 (P5) mice were intraperitoneally injected daily for 6 days with apelin (10 mg/kg), apelin-dm (10 mg/kg), or both (10 mg/kg each), via intraperitoneal (IP) injection. Following the treatment period, the mice were humanely euthanized, and their eyes were harvested. The eyes were immediately transferred to 4% paraformaldehyde (PFA) for fixation for 10–15 min, after which they were transferred to cold phosphate-buffered saline (PBS) on ice in a Petri dish for dissection. The retinas were carefully isolated and fixed in 4% PFA for 2 h at room temperature or overnight at 4 °C. After fixation, the retinas were blocked and permeabilized in a solution containing 1% bovine serum albumin (BSA) and 0.5% Triton X-100 overnight at 4 °C. Subsequently, the retinas were washed three times with a buffer solution composed of 0.5% Triton X-100, 1 mM CaCl$_2$, 1 mM MgCl$_2$, and 1 mM MnCl$_2$ in PBS, pH 6.8. The retinas were then incubated with FITC-labeled Isolectin B4 (1:1000; Vector Laboratories) to label blood vessels. Following staining, the retinas were embedded in Tissue-Tek OCT Compound, and 10-μm cryosections were cut. Confocal immunofluorescence images were acquired with an inverted Nikon C2si Eclipse Ti-S microscope and analyzed using NIS-ElementsAR software (Nikon Instruments Europe B.V.).

## Maximal tolerated dose (MTD) and repeated dosing toxicity study

Apelin-dm was administered at the maximum tolerated dose to six groups of BALB/c mice ($n = 6$/group; 3 males, 3 females). The control group received saline, while apelin-dm doses were 200, 500, 1000, 1500, and 2000 mg/kg. A single intraperitoneal injection on day 1 was followed by a 14-day observation period. Parameters monitored included mortality, morbidity, clinical signs, and body weight. At the study's end, gross pathology and organ necropsy were conducted. For the repeated dosing toxicity study, 40 Balb/C mice (5 males, 5 females/group) were injected 5 days/week for 2 weeks with saline or Apelin-dm (50, 150, 300 mg/kg). Evaluations encompassed clinical observations, body weight, food consumption, hematology, blood chemistry, gross pathology, selected organ necropsy, and weight, and histopathology examinations. Comprehensive study details are in Appendix pharmacological studies-1 and -2.

## The absorption, distribution, metabolism, excretion (ADME) and pharmacokinetics (PK) studies

For in-solution property studies, plasma-binding protein assays were performed using the equilibrium dialysis technique, and an aqueous solubility test was performed using a shake flask system. For in vitro metabolism studies, the half-life and intrinsic clearance of the apelin-dm peptide were evaluated in human plasma and human and mouse liver microsomes, respectively. Cytochrome P450 (CYP) inhibition was assayed by fluorimetry using recombinant human CYP. Cardiac toxicity was assessed using the hERG-automated whole-cell patch-clamp technique. For each assay, various reference compounds were used for comparison. Details of the complete study are provided in Appendix pharmacological studies-3 and -4.

## Preclinical efficacy experiments

Tumor growth effects were assessed by injecting $1 \times 10^6$ control or APLN/APLN-DM expressing CT-26/MC-38 cells subcutaneously into BALB/c, Rag2/γc, or C57BL/6 mice. Additional experiments involved injecting $1 \times 10^6$ control CT-26/MC-38/MDA-MB231 cells into BALB/c, C57BL/6, or nude mice, respectively. Apelin-dm treatment (10 mg/kg, i.p.) or saline control commenced when tumors reached ~60 mm$^3$. LASER Doppler imaging assessed total blood flow, with FlowR® Hemodynamics visualization used for analysis. Results were expressed as % relative to pre-treatment tumor blood flow. Experimental liver metastases were induced as previously described (Sfaxi et al, 2014; Scamuffa et al, 2008) by intrasplenic/portal injection of $1 \times 10^6$ CT-26 or MC-38 cells, followed by randomization into treatment (10 mg/kg, i.p.) or saline groups after 5–6 days.

## Immunostaining and confocal microscopy

Sections derived from colon cancer and metastatic liver patients and their corresponding non-cancerous tissues, or from tumors induced in C57BL/6, BALB/c, or Rag2/γ$_c$ mice, were stained with unconjugated antibodies and fluorochrome-associated secondary antibodies. All samples were mounted with Prolong containing DAPI (Invitrogen), and confocal immunofluorescence images were taken using the inverted microscope Nikon C2si Eclipse Ti-S with NIS-ElementsAR software (Nikon Instruments Europe B.V.).

## Proteomic analysis

Three independent biological replicates of total protein extracts from control CT-26 cells stably expressing APLN and APLN-DM were compared to control CT-26 cells by label-free protein quantification. Proteins were loaded on a 10% acrylamide SDS-PAGE gel and visualized by Colloidal Blue staining. Migration was stopped when the samples had just entered the resolving gel and the unresolved region of the gel was cut into only one segment. Sample preparation and protein digestion by trypsin and NanoLC-MS/MS analysis were performed as previously described with modification (Henriet et al, 2017) and briefly detaile in supplementary Material and Methods. NanoLC-MS/MS analysis was performed using an Ultimate 3000 RSLC Nano-UPHLC system (Thermo Scientific, USA) coupled to a nanospray Q-Exactive hybrid quadruplole-

Orbitrap mass spectrometer (Thermo Scientific, USA). Each peptide extract was loaded onto a 300 μm ID × 5 mm PepMap C18 pre-column (Thermo Scientific, USA) at a flow rate of 20 μL/min. After a 5 min desalting step, peptides were separated on a 75 μm ID × 25 cm C18 Acclaim PepMap® RSLC column (Thermo Scientific, USA) with a 4–40% linear gradient of solvent B (0.1% formic acid in 80% ACN) for 108 min. The separation flow rate was set at 300 nL/min. The mass spectrometer was operated in the positive ion mode at a 1.8 kV needle voltage. Data were acquired using Xcalibur 3.1 software in data-dependent mode. MS scans ($m/z$ 350–1600) were recorded at a resolution of R = 70000 (at $m/z$ 200) and an AGC target of $3 × 10^6$ ions was collected within 100 ms. Dynamic exclusion of selected precursors was set to 30 s and the top 12 ions were selected from fragmentation in the HCD mode. MS/MS scans with a target value of $1 × 10^5$ ions were collected with a maximum fill time of 100 ms and a resolution of R = 17,500. In addition, only +2 and +3 charged ions were selected for fragmentation. Other settings were as follows: no sheath and auxiliary gas flow, heated capillary temperature, 200 °C; normalized HCD collision energy, 27%; and isolation width, 2 $m/z$. Protein identification was performed using Proteome Discoverer 1.4. Mascot 2.5 and Sequest HT algorithms were used for protein identification in batch mode by searching against a UniProt Mus musculus protein database (49 821 entries, release October 2016; https://www.uniprot.org/ website). The mass tolerances in MS and MS/MS were set to 10 ppm and 0.02 Da. Oxidation (M) and acetylation (K) were searched for dynamic modifications, and carbamidomethylation (C) was used as a static modification. Peptide validation was performed using Percolator algorithm (Käll et al, 2007) and only "high confidence" peptides were retained corresponding to a 1% False Positive Rate at peptide level. Raw LC-MS/MS data were imported into Progenesis LC-MS (version 4.1; Nonlinear Dynamics, Waters) for feature detection, alignment, and quantification. All sample features were aligned according to retention times by manually inserting up to 50 landmarks, followed by automatic alignment to maximally overlay all two-dimensional ($m/z$ and retention time) feature maps. Singly charged ions and ions with more than six charge states were excluded from the analysis. All remaining features were used to calculate a normalization factor for each sample, which was corrected for experimental variation. Peptide identifications (with FDR < 1%; see above) were imported into Progenesis software. Only nonconflicting features and unique peptides were considered for quantification at the protein level. Quantitative data were considered for proteins, quantified by a minimum of two unique peptides, a fold change above 2, and a statistical $P$ value (Welch's $t$ test) lower than 0.05.

## PamGene kinase assay

Following treatment with APLN an APLN-DM, the cells were washed and lysed in M-PER lysis buffer (Thermo Fisher Scientific, 78503) supplemented with Halt Protease and Phosphatase inhibitors (87786 and 78420, Thermo Fisher Scientific), following PamGene's instructions. The lysates were centrifuged at 4 °C for 15 min at 15,000 × $g$, and the supernatant was stored at –80 °C until further analysis. For the PamGene kinase assay, Each PamChip® contains four identical porous arrays spotted with distinct 13-amino acid-long peptide substrates with phosphosites (196 for the protein tyrosine kinase (PTK) and 144 for the serine–threonine kinase (STK) arrays). Phosphorylation of phosphosites is then used to predict one or multiple upstream kinases (protein tyrosine kinases for the PTK PamChip® and serine–threonine kinases for the STK PamChip®). Therefore, STK and PTK microarray assays were performed according to the manufacturer's instructions (PTK assay on PamStation 12 User Manual, Version 3.0; Serine Threonine Kinase assay on PamStation 12 User Manual, Version 5.1) (ref). In brief, cell lysates were mixed with reaction solutions generated with the STK reagent kit (PamGene, 32201) or PTK reagent kit (PamGene, 32112) and ampuwa ultrapure water (Fresenius Kabi) and placed on STK PamChips (PamGene, 32501) and PTK PamChips (PamGene, 32508), respectively. Triplicates of the cell lysates derived from the apelin- and apelin-dm-treated cells were processed within the same assay run to allow for correct comparisons. Fluorescently labeled anti-phospho-antibodies were used to detect the phosphorylation activity of kinases present in the sample during and after the pumping of lysates through the 3-dimensional surface of the array. During the assay, the samples were pumped through the porous membrane, and images of each array were taken at several exposure times using a camera in the workstation. Images were later used by BioNavigator® software to calculate the signal values for each phosphosite. The data workflow consisting of image quantification, quality control, statistical analysis, visualization, and interpretation was performed using BioNavigator® software. Analysis of PamGene was performed using the BioNavigator v63 software (BN63; PamGene® International, The Netherlands). After excluding several peptides (undetectable or without kinetics (PTK) and quality control (QC)), 125 of 196 PTK substrates and 120 of 144 STK substrates on the PamChip® kinome array were included in the final analysis. For data normalization, the ComBat batch-effect correction method (Johnson et al, 2007) was used after the $\text{Log}_2$ transformation of the S100 signals. The ratios of the normalized signals of the treated samples and controls were used to calculate the log-fold change for each peptide. For generating peptide phosphorylation heatmaps and compare the phosphorylation levels across all samples, normalized $\text{Log}_2$ S100 signals were used. Statistical significance was tested using Unpaired $t$ tests, and the results are represented by volcano plots generated using BN63 (BN63; PamGene® International, The Netherlands). Peptides with a $P$ value < 0.05 were considered a significant change in the degree of phosphorylation of a peptide in the two groups.

## Docking computations and structural analyses

The crystal structure (PDB 5VBL) of apelin receptor in complex with apelin peptide mimetics (AMG3054) (Ma et al, 2017; Yue et al, 2022) was used for the docking of apelin-13 and apelin-dm and the generation of the apelin receptor-apelin-13 or apelin receptor-apelin-dm complex. The ten best apelin receptor-apelin-dm models were generated using the standalone version of the CABS-dock flexible docking engine (Kurcinski et al, 2019). The flexibility of the docked apelin receptor-apelin-dm peptide models and the apelin receptor-apelin-13 transposed model were investigated using the standalone version of CABS-flex (Kurcinski et al, 2019). The CABS Python engine tools use a medium-resolution coarse-grained representation of a protein structure (i.e., not all atoms are present but only the C-alpha and beta atoms, while a pseudo-atom is placed

in the center of mass of the remaining side chain atoms) to accelerate the simulation (~3–4 orders of magnitude faster than all-atom molecular dynamics simulation). The full atom structural models were reconstructed using the Protein Chain Reconstruction Algorithm (PULCHRA) (Rotkiewicz and Skolnick, 2008). Additional short energy minimization of the peptides and modeled receptor-peptide complexes was performed with the molecular modeling toolkit MMTK (http://dirac.cnrs-orleans.fr/MMTK.html) to refine the geometry of the system. The per-residue decomposition (of the various energy-minimized apelin receptor-apelin-13 or apelin-dm structural models) of docking energy was computed using pyDockEneRes (Romero-Durana et al, 2020). The computed binding energy score obtained from pyDockEneRes was a linear combination of three energetic terms: electrostatic, desolvation, and van der Waals. The orientation of the receptor structure with respect to the lipid bilayer was predicted using the PPM server (Lomize et al, 2012). The search for allosteric pockets on the receptor was performed using the open-source code, APOP (Kumar et al, 2023). This algorithm predicts binding pockets (via the use of Voronoi tesselation and alpha shapes) and local hydrophobic density using Fpocket (Le Guilloux et al, 2009) and perturbs the predicted pockets by stiffening pairwise interactions in the elastic network of the residues lining the investigated pocket to emulate ligand binding. Structural analysis was performed using UCSF ChimeraX (Pettersen et al, 2021) and the figure was prepared using PyMOL (https://www.schrodinger.com).

## Plasmon waveguide resonance (PWR)

PWR was used to follow apelin receptor conformational changes upon apelin and apelin-dm addition to HEK-apelin receptor cell membrane fragments overexpressing apelin receptor immobilized on the sensor surface. PWR measurements were performed in a homemade instrument functioning at a fixed wavelength of 632 nm and variable incident angle with an angular resolution of about 0.5 millidegrees, as previously described (Soulet et al, 2020). The polarization angle of the incident light was 45° to allow both p-polarized (parallel to the incident light and perpendicular to the sensor surface) and s-polarized (perpendicular to the incident light and parallel to the sensor surface) light resonances to be obtained within a single angular scan. The sensor consists of a BK-7 prism coated with silver and silica to support the waveguide. All the measurements were performed at 22 °C.

## Statistical analysis

Unless otherwise indicated, data are presented as mean ± s.e.m. A two-tailed $t$ test and one- or two-way ANOVA with Tukey's multiple comparisons test were used to analyze the data. Pearson's coefficient was calculated to determine the correlation between normally distributed protein expression in human tumors. Differences between mice groups were analyzed using the Mann–Whitney test in GraphPad Prism (GraphPad Software). No inclusion or exclusion criteria were applied to the human samples or animals included in the analysis, and the statistical significance level is illustrated with $P$ values. Blinding and randomization were applied when pertinent. Specifically for experiments involving animal models and patient-derived samples, that were randomly assigned to experimental groups to minimize selection bias.

### The paper explained

#### Problem
Colorectal cancer stands as the second leading cause of cancer-related mortality worldwide and ranks third in global cancer diagnoses. Despite the efficacy of surgical resection and adjuvant treatments in early-stage disease, relapse and metastasis remain common and the underlying mechanisms involved in these processes remained elusive. Thus, the discovery of a novel efficient regulator may help elucidate the potential mechanisms of the colorectal liver metastasis pathway and provide potential therapeutic strategies for colorectal liver metastasis treatment.

#### Results
We report apelin-specific cleavage and activation by Furin as a key player in colon cancer and early events of colon cancer and liver cells interaction leading to metastasis. Alteration of apelin cleavage sites generates apelin-dm peptide that effectively represses the malignant and metastatic phenotype of cancer cells and angiogenesis through modulation of apelin receptor dynamics, affinity, internalization, and diverse apelin signaling pathways. Pharmacokinetic and toxicity assessments confirm the specificity, safety, and stability of apelin-dm.

#### Impact
Through multi-level studies of cell culture, pathological specimens, and animal models, we identify the importance of Furin in apelin activity in colon cancer and liver metastasis and apelin-dm peptide as a potential therapeutic strategy, paving the way for further clinical exploration.

Statistical $P$ was set than 0.05. The relative staining intensity (RSI) was calculated using ImageJ by measuring the integrated density of staining (I stain) within manually selected ROIs, subtracting the background staining intensity (I bg) and normalizing by the area of the ROI (A stain). The Relative Staining Intensity (RSI) was calculated using the following formula: RSI = Istain − Ibg/Astain.

## Data availability

The sequencing data were deposited in the ProteomeXchange consortium via the PRIDE (Perez-Riverol et al, 2019) partner repository. The proteomic data from this publication have been deposited to the Proteomic analysis of Control, APLN, APLN-DM-expressing cells database https://www.ebi.ac.uk/pride/login and assigned the identifier PXD045256. The reviewer account details are Username: reviewer_pxd045256@ebi.ac.uk and Password: XiQWy1dj.

The source data of this paper are collected in the following database record: biostudies:S-SCDT-10_1038-S44321-025-00196-5.

## Peer review information

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

## Acknowledgements

The authors thank Vect'UB and One Cell of the TMB-Core of the University of Bordeaux for the lentiviral vector construction and production. This work was supported by Institute Bergonié, by Conseil Régional Aquitaine (Région Nouvelle-Aquitaine), by INSERM.

## Author contributions

**Béatrice Demoures**: Conceptualization; Data curation; Formal analysis; Validation; Investigation; Methodology; Writing—original draft. **Fabienne Soulet**: Formal analysis; Investigation; Methodology. **Jean Descarpentrie**: Software; Formal analysis; Investigation; Methodology. **Isabel Galeano-Otero**: Software; Formal analysis; Investigation; Methodology. **José Sanchez Collado**: Software; Formal analysis; Investigation; Methodology. **Maria Casado**: Formal analysis; Investigation; Methodology. **Tarik Smani**: Resources; Supervision; Investigation. **Alvaro González**: Formal analysis; Investigation; Methodology. **Isabel Alves**: Formal analysis; Investigation; Methodology. **Fabrice Lalloué**: Formal analysis; Investigation; Methodology. **Bernard Masri**: Formal analysis; Investigation; Methodology. **Estelle Rascol**: Formal analysis; Investigation; Methodology. **Jean-William Dupuy**: Software; Investigation; Methodology. **Cyril Dourthe**: Software; Investigation. **Frédéric Saltel**: Software; Investigation; Methodology. **Anne-Aurélie Raymond**: Software; Investigation; Methodology. **Iker Badiola**: Formal analysis; Supervision; Investigation; Methodology. **Serge Evrard**: Resources; Funding acquisition; Investigation. **Bruno Villoutreix**: Software; Methodology. **Simon Pernot**: Resources; Supervision; Funding acquisition; Investigation. **Géraldine Siegfried**: Conceptualization; Resources; Data curation; Formal analysis; Funding acquisition; Validation; Project administration; Writing—review and editing. **Abdel-Majid Khatib**: Conceptualization; Resources; Data curation; Formal analysis; Funding acquisition; Validation; Project administration; Writing—review and editing.

Source data underlying figure panels in this paper may have individual authorship assigned. Where available, figure panel/source data authorship is listed in the following database record: biostudies:S-SCDT-10_1038-S44321-025-00196-5.

## Disclosure and competing interests statement

The authors declare no competing interests.

# Expanded View Figures

Human.AAF25815.1
MNLRLCVQALLLLWLSLTAVCGGSLMPLPDGNGLEDGNVRHLVQPRGSRNGPGPWQGG*RR*KF*RR*QRPRLS HKGPMPF

Gorilla.XP_018874376.1
MNLRLCVQALLLLWLSLTAVCGGSLMPLPDGNGLEEGNVRHLVQPRGSRNGPGPWQGG*RR*KF*RR*QRPRLS HKGPMPF

Rat. AAF25814.1
MNLSFCVQALLLLWLSLTAVCGVPLMLPPDGKGLEEGNMRYLVKPRTSRTGPGAWQGG*RR*KF*RR*QRPRLS HKGPMPF

Mouse. NP_038940.1
MNLRLCVQALLLLWLSLTAVCGVPLMLPPDGTGLEEGSMRYLVKPRTSRTGPGAWQGG*RR*KF*RR*QRPRLS HKGPMPF

Zebrafish. NP_001159596.1
MNVKILTLVIVLVVSLLCSASAGPMASTEHSKEIEEVGSMRTPLRQNPARAGRSQRPAGW*RRRR*PRPRLSHKGPMPF

**Figure EV1. Comparison of apelin precursor sequences from various species.**

Represented are the putative peptide sequences of human, Gorilla, rat, mouse and zebrafish. The positions of the convertase general motif (K/R)–(X)n–(K/R)↓, where $n = 0$, 2, 4 or 6 and X (green and in italics) are conserved. The National Center for Biotechnology Information (NCBI) accession number for each sequence is given.

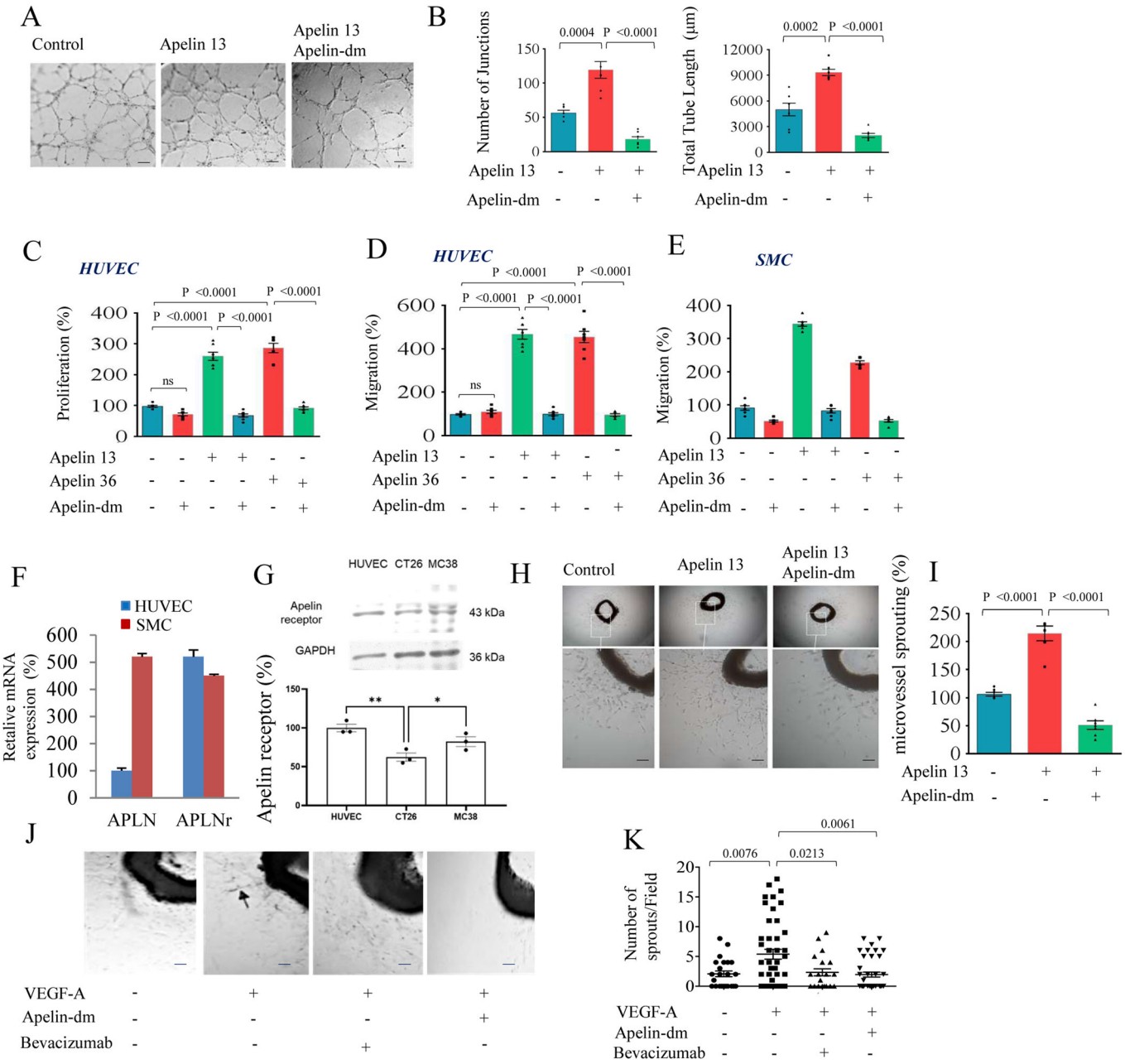

**Figure EV2. Effects of apelin-dm on endothelial tube formation, cell migration, proliferation, and aortic ring microvessel sprouting.**

(A) Representative images of apelin-dm and/or apelin peptides effect on tube-like structure formation by HUVEC. Scale bar indicates 250 μm. (B) Effect of Apelin-dm on apelin-induced tube-like structure formation measured by the number of junctions and tubule length. (C, D) Apelin-dm inhibits HUVEC proliferation (C) and migration (D) and smooth muscle cells (SMC) migration (E) induced by apelin-13 and 36 peptides. (F) Relative expression of APLN and APLNR mRNA in Huvec and SMC expressed relative to APLN mRNA abundance in HUVEC assigned 100%. (G) Western blotting analysis of apelin receptor expression in HUVEC, CT-26 and MC-38 cells, expressed relative to apelin receptor protein abundance in HUVEC assigned 100%. (H) Representative images of apelin-dm peptide effect alone or on apelin-mediated aortic ring microvessel sprouting. (I) Quantification of aortic ring vascular sprout surface per aortic ring relative to control untreated aorta (100%). (J) Representative images of apelin-dm peptide or Bevacizumab effect on VEGF-mediated aortic ring microvessel sprouting. Scale bar indicates 250 μm. (K) Quantification of number of sprouts per field. The data are representative of three independent experiments. n.s. not significant. All values represent the mean ± s.e.m. Significant differences P were determined by two-way ANOVA.

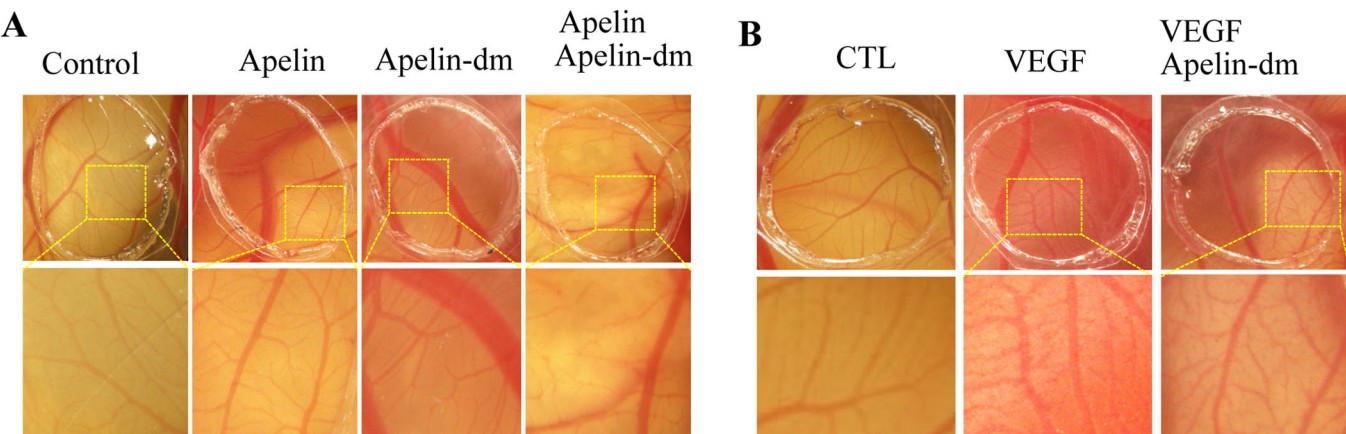

**Figure EV3.  Effects of apelin-dm on apelin- and VEGF-induced vessel formation in the CAM Assay.**

(**A**, **B**) Representative images of apelin-dm effect alone, on apelin (**A**) or on VEGF-mediated (**B**) vessel formation in the CAM assay (corresponding to Fig. 3D, H). On day 9, the CAM received either vehicle (Control), 100 nM of apelin, 100 nM of apelin-dm or both. For A, the photographs shown were taken after treatments (24 h) and are representative of the results obtained in an additional 5 eggs per group. For (**B**), on day 9, the CAM received either vehicle (Control), 20 ng VEGF, or VEGF and apelin-dm. The photographs shown were taken after treatments (24 h) and are representative of the results obtained in an additional 5 eggs per group. high-magnification pictures of CAMs are also indicated.

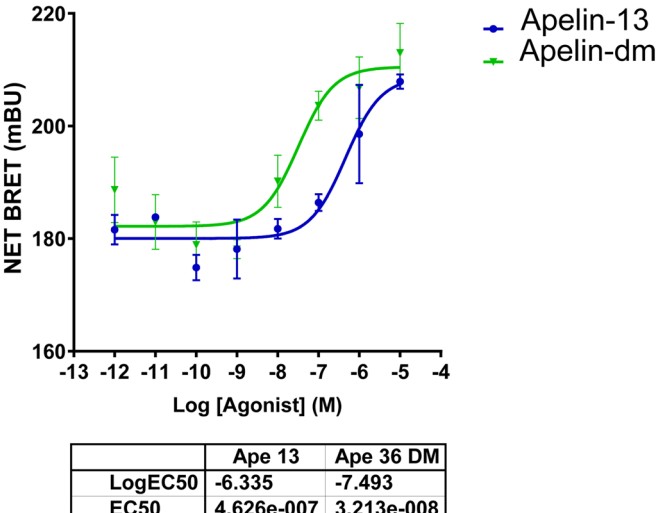

| | Ape 13 | Ape 36 DM |
|---|---|---|
| LogEC50 | -6.335 | -7.493 |
| EC50 | 4.626e-007 | 3.213e-008 |

**Figure EV4. Dose-dependent increase in BRET Signal by apelin and apelin-dm, Indicating apelin receptor internalization.**

HEK293T cells transiently transfected with apelin receptor-Rluc and Rab5-YFP were used to measure the Bioluminescence Resonance Energy Transfer (BRET) signal, indicative of apelin receptor internalization in the presence of various concentration of apelin and apelin-dm. Net BRET signals were determined by subtracting the BRET signal from cells expressing only Rluc-tagged apelin receptor from those co-expressing Rluc-tagged apelin receptor and YFP-tagged Rab5.

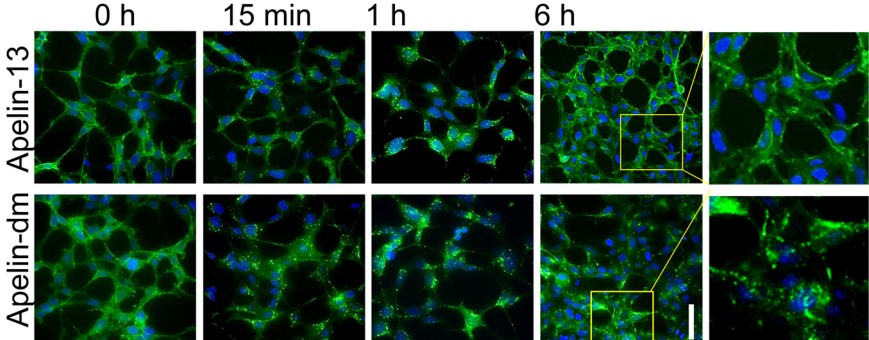

**Figure EV5.   Representative images of apelin-13 and apelin-dm peptides effect on apelin receptor internalization in HEK293A cells stably expressing apelin receptor-EGFP fusion protein.**

The data are representative of three independent experiments. Six hours after peptides washout, the apelin receptor seemed to be mainly returned to the cell surface in apelin-13-treated cells. In contrast, in cells treated with apelin-dm, the intracellular vesicles of the apelin receptor remained internalized. Scale bar, 100 μm.

