## [Peer Review File · EMBO Molecular Medicine]

Repression of apelin Furin cleavage sites provides antimetastatic strategy in colorectal cancer

Béatrice Demoures, Fabienne Soulet, Jean Descarpentrie, Isabel Galeano-Otero, José Sanchez Collado, Maria Casado, Tarik Smani, Alvaro González, Isabel Alves, Fabrice Lalloue, Bernard Masri, Estelle Rascol, Jean-William Dupuy, Cyril DOURTHE, Frederic SALTEL, Anne-Aurelie Raymond, Iker Badiola, Serge Evrard, Bruno Villoutreix, Simon Pernot, Geraldine Siegfried, and Abdel-Majid KHATIB

Corresponding authors: Abdel-Majid KHATIB (majid.khatib@u-bordeaux.fr) , Geraldine Siegfried (geraldine.siegfried@inserm.fr)

Review Timeline:

Submission Date:	8th Feb 24
Editorial Decision:	12th Mar 24
Revision Received:	5th Sep 24
Editorial Decision:	7th Oct 24
Revision Received:	21st Dec 24
Editorial Decision:	10th Jan 25
Revision Received:	11th Jan 25
Accepted:	16th Jan 25

Editor: Lise Roth

Transaction Report:

12th Mar 2024

Dear Prof. Khatib,

Thank you for the submission of your manuscript to EMBO Molecular Medicine. We have now received feedback from the three reviewers who agreed to evaluate your manuscript. As you will see from the reports below, the referees acknowledge the interest of the study and are overall supporting publication of your work pending appropriate revisions.

Adequately addressing the reviewers' concerns in full will be necessary for further considering the manuscript in our journal, and acceptance of the manuscript will entail a second round of review. It will be particularly important to clarify the mechanism of action of APLN-DM.

EMBO Molecular Medicine encourages a single round of revision only and therefore, acceptance or rejection of the manuscript will depend on the completeness of your responses included in the next, final version of the manuscript. For this reason, and to save you from any frustrations in the end, I would strongly advise against returning an incomplete revision.

We are expecting your revised manuscript within three months, if you anticipate any delay, please contact us.

We require:

4) A .docx formatted letter INCLUDING the reviewers' reports and your detailed point-by-point responses to their comments. As part of the EMBO Press transparent editorial process, the point-by-point response is part of the Review Process File (RPF), which will be published alongside your paper.

5) A complete author checklist, which you can download from our author guidelines (<https://www.embopress.org/page/journal/17574684/authorguide#submissionofrevisions>). Please insert information in the checklist that is also reflected in the manuscript. The completed author checklist will also be part of the RPF.

6) Please note that all corresponding authors are required to supply an ORCID ID for their name upon submission of a revised manuscript.

7) It is mandatory to include a 'Data Availability' section after the Materials and Methods. Before submitting your revision, primary datasets produced in this study need to be deposited in an appropriate public database, and the accession numbers and database listed under 'Data Availability'. Please remember to provide a reviewer password if the datasets are not yet public (see <https://www.embopress.org/page/journal/17574684/authorguide#dataavailability>).

In case you have no data that requires deposition in a public database, please state so in this section ("This study includes no data deposited in external repositories.").

Note that the Data Availability Section is restricted to new primary data that are part of this study.

8) For data quantification: please specify the name of the statistical test used to generate error bars and P values, the number (n) of independent experiments (specify technical or biological replicates) underlying each data point and the test used to calculate p-values in each figure legend. The figure legends should contain a basic description of n, P and the test applied. Graphs must include a description of the bars and the error bars (s.d., s.e.m.). Please provide exact p values.

9) Our journal encourages inclusion of *data citations in the reference list* to directly cite datasets that were re-used and obtained from public databases. Data citations in the article text are distinct from normal bibliographical citations and should

directly link to the database records from which the data can be accessed. In the main text, data citations are formatted as follows: "Data ref: Smith et al, 2001" or "Data ref: NCBI Sequence Read Archive PRJNA342805, 2017". In the Reference list, data citations must be labeled with "[DATASET]". A data reference must provide the database name, accession number/identifiers and a resolvable link to the landing page from which the data can be accessed at the end of the reference. Further instructions are available at .

13) Author contributions: CRediT has replaced the traditional author contributions section because it offers a systematic machine readable author contributions format that allows for more effective research assessment. Please remove the Authors Contributions from the manuscript and use the free text boxes beneath each contributing author's name in our system to add specific details on the author's contribution. More information is available in our guide to authors.

16) As part of the EMBO Publications transparent editorial process initiative (see our Editorial at <http://embomolmed.embopress.org/content/2/9/329>), EMBO Molecular Medicine will publish online a Review Process File (RPF) to accompany accepted manuscripts.

In the event of acceptance, this file will be published in conjunction with your paper and will include the anonymous referee reports, your point-by-point response and all pertinent correspondence relating to the manuscript. Let us know whether you agree with the publication of the RPF and as here, if you want to remove or not any figures from it prior to publication. Please note that the Authors checklist will be published at the end of the RPF.

I look forward to receiving your revised manuscript.

Yours sincerely,

Lise Roth

***** Reviewer's comments *****

Referee #1 (Remarks for Author):

The apelin receptor is a class A G protein-coupled receptor (GPCR), that binds two families of endogenous peptide ligands, apelin and elabela also known as Toddler. The apelin signalling pathway has been identified as being upregulated with poor prognosis in a range cancers. To date there have been limited studies on blocking the action of apelin using a selective apelin antagonist in animal models. These have been mainly in animal models of glioblastoma multiforme, (Harford-Wright, et al 2017 doi: 10.1093/brain/awx253) and a mouse melanoma lung metastasis model (Berta, et al 2021,. doi: 10.1038/s41598-021-85162-0.)

This study is important as it shows an increase in apelin peptide and its receptor has a key role in the development of metastatic colorectal cancer. APLN-DM, a peptide apelin analogue, with modifications to furin cleavage sites, is shown to inhibit tumor growth, promoting cell death, suppressing angiogenesis, and impeding early colorectal liver metastasis. The proposed mechanism of action is via modulating apelin receptor dynamics (internalization and signaling) and competing for apelin binding.

This study is significant for several reasons. Firstly, it demonstrates evidence for the importance of apelin in colorectal cancer. This suggests the receptor may be a potential therapeutic target in an increasing number of conditions and play a key role in tumour progression. To date Family A GPCRs have not been exploited extensively in cancer treatments.

Secondly the authors show APLN-DM, has favorable drug-like properties in pharmacodynamics, with an increased plasma half-life compared with apelin. As a peptide, APLN-DM as pharmacodynamics, has high affinity, conferring with a high degree of selectivity and potency. Most peptides with these properties are well tolerated with few side effects, so likely to be translatable into the clinic (Davenport et al 2020. doi: 10.1038/s41573-020-0062-z).

The focus of my comments is limited to the metabolism and pharmacology of apelin.

Results P5, 6 APLN precursor is specifically cleaved by Furin and APLN, APLNR, and Furin expression is altered in colorectal cancer and colorectal liver metastases

Furin is a cellular endoprotease that cleaves many proproteins precursors at a consensus cleavage site X-Arg-X-Lys/Arg-Arg-X. The gene encoding the enzyme is expressed in virtually all tissues and individual cells <https://www.proteinatlas.org/ENSG00000140564-FURIN>. As expected furin cleaves apelin precursor at paired amino acid sites (KK) to yield apelin 17 and apelin 13. It is not surprising that the enzyme is expressed in normal and cancer cells as shown Fig 1. In cancer, it is well documented that furin increases (Bassi et al .2003: 10.1016/S0002-9440(10)63838-2.)

The authors state 'These findings, indicate that increased co-expression levels of APLN, APLNR, and/or Furin are associated with CRC and derived mCRC tumors, suggesting a potential implication of these proteins in colon cancer'. The authors do not state the source or characteristics of the antisera but does the fluorescent staining represent furin cleaved products only ie apelin 13 (as suggested in Fig 2A), rather than apelin 17 or precursor?

The increase in apelin receptor is intriguing; an increase in endogenous ligand often results in a compensatory downregulation of its target GPC receptor but this pathway seems detrimentally dysregulated in cancer. The co-localisation to cells suggests apelin is acting in an autocrine/paracrine manner as one factor driving tumor growth; is this interpretation correct, can the authors comment?

Alteration of proAPLN cleavage sites contributes to reduction in colorectal tumor growth and liver metastasis.

The authors mutated proAPLN cleavage sites RR60KFRR64QR to SS60KFSS64QR, creating the APLN Double Mutant (APLN-DM). The specific site for cleavage has been mutated and it might be expected that this would almost abolish generation of biologically active peptides apelin 17 and apelin 13, similar to an antagonist completely blocking an agonist. In fact, in Fig 2C and 2E there seems substantial growth in tumor volume in APLN-DM expressing cells. What is the reason? Does this indicate an alternative pathway for synthesis of biologically active peptides not necessarily cleaved at RR60KFRR64QR to SS60KFSS64QR or unmasking other agents mediating these responses? Does this have implication for translating APLN-DM to an efficacious dose in vivo.

APLN-DM peptide represses APLN-mediated vascular network and tumor cells malignant phenotype

In these experiments 'chick CAM' was used although not defined but assumed to be chicken chorioallantoic membrane, which is highly vascularized extraembryonic membrane. Representative images of chick CAM were treated for 48 h with APLN or VEGF in absence and presence of APLN-DM and various parameters measured such as APLN-DM inhibiting both HUVEC and SMC proliferation and migration.

The key question is the comparatively high concentration of 100nM used given the APLN-DM IC50 value was sub-nanomolar (0.45 nM) versus radiolabelled apelin binding. Is there an explanation for this high concentration eg there is about 50% plasma binding (Fig 8E); were the experiments conducted with serum albumen in the media? Is there any significant metabolism under these conditions?

It is conventional to convert IC50 values from ligand binding using the Cheng-Prusoff equation to the Ki (affinity constant) of a ligand. The measured IC50 changes depending on the concentration of the radiolabelled ligand used.

The apelin receptor is unusual in being activated by two endogenous peptides, apelin and Elabela that have distinct amino acid sequences. (see Ref 3 in manuscript Read C, et al. 2019)

Ela is also expressed in humans, colocalising for example with apelin in endothelial cells and circulates in the plasma. It acts on the apelin receptor in a similar way to apelin in conditions such as pulmonary arterial hypertension (see Yang et al 2017. doi: 10.1161/CIRCULATIONAHA.116.023218). It has also been linked to cancer (Sharma et al doi: 10.2174/1389450123666220826160123)

In cancer, compounds developed as therapeutic agents are likely to need to modulate the action of both peptides in binding to the receptor. Have the authors determined in functional assays whether APLN-DM reduces the activity of Elabela in a similar way to apelin?

Minor Points

To improve interpreting figures please add number of replicates to the figure legends where these are missing.

Eg Supplemental Figure 3. The percentage of relative staining intensity of APLN, Ki67, APLNR and Furin for indicated tissue deduced from (Fig. 1G, 1H) . All values represent the mean{plus minus}s.e.m. P values by two-tailed unpaired t-test.

Nomenclature. The International Union of Basic and Clinical Pharmacology recommends (Ref 3 in manuscript Read C, et al. 2019) that APLN and APLNR are used to refer to the human apelin gene and the gene encoding the apelin receptor, respectively. The peptide apelin and protein apelin receptor, should ideally be spelt out in full to avoid confusion with the gene names (eg p14).

Referee #2 (Comments on Novelty/Model System for Author):

This study has been validated using a mouse model, which is adequate.

Referee #2 (Remarks for Author):

In this study, the authors showed that Furin is a specific proprotein convertase involved in the cleavage and activation of the APLN precursor protein (proAPLN). They demonstrated that treatment with APLN-DM obtained by mutation of Furin cleavage sites exhibits antitumor effects by suppressing tumor growth and metastasis. They also revealed that the molecular mechanism of APLN-DM is a direct competition with wild-type APLN to act on the APLN receptor and affect signaling pathways. Analysis of its toxicity and pharmacokinetic profiles characterizes the safety, specificity and stability of APLN-DM. These results support that APLN-DM has the potential to be an effective treatment for metastatic colorectal cancer. Since the findings are novel and potentially significant, the authors should be encouraged to address the key concerns detailed below.

Major comments:

1. APLN and APLNR are known to be highly expressed in vascular endothelial cells in tumor tissue. It is necessary to clarify whether Furin is also highly expressed in vascular endothelial cells.
- 2 . Although APLN-DM shows "increased anchorage-independent growth" for CT26 and MC38, the expression levels of APJ in these cells should be shown in comparison to vascular endothelial cells. Also, the absolute expression levels of Ferin in CT26, MC38, SMC, and HUVECs used in cell culture experiments should be shown.

3. The authors should clearly indicate whether the anti-tumor effect of APLN-DM is mainly due to a direct effect on tumor cells or an indirect effect through inhibition of angiogenesis. The use of APLNR -KD tumor cells can clarify this point.
4. As described by the authors in the introduction, Apelin expression is known to be induced in a hypoxic environment. Is Ferin expression similarly induced by hypoxia?
5. In the suppression of tumor growth induced by APLN-DM treatment, is the reduction in Ki67-positive cells an indirect effect of the reduced number of vessels? By co-staining CD31 and Ki67, it can be shown whether only the proliferation of tumor cells in the avascular region is reduced.
6. Although it is described on page 12 as "E-selectin mRNA increased at 2h, returning to basal levels after 4-6 hours;" E-selectin mRNA expression remains high even after 6 hours in Figure 6A. In addition, why is there a difference in the duration of E-selectin expression between HUVECs, HMEC-1, and LSECs?
7. MM54 appears to have a similar metastatic inhibitory effect as APLN-DM. The author should test whether APLN-DM is effective for tumor growth compared to existing APJ antagonists.
8. The authors should analyze the mechanism of inhibition of VEGF-induced angiogenesis by APLN-DM.

Minor comments:

1. Why are there no dots on the right graph in Figure 1I?
2. What does RSI show in the graphs in Figure 2I and J?
3. Text in Figure 2N is not clear.
4. In the Introduction, Apelin is described as "promoting epithelial cell growth, proliferation, and migration," but the referenced paper suggests a direct effect on endothelial cells.
5. Figure 3R is not described in the manuscript.

Referee #3 (Comments on Novelty/Model System for Author):

The experiments need more significant controls and leave many questions open. Furthermore, the quality of the data is low.

Referee #3 (Remarks for Author):

The researchers investigated the role of APLN in tumorigenesis, focusing on its processing and physiology. They identified Furin as the cleaving protease, mutated the Furin cleavage sites within APLN (APLN-DM), and observed that it acted as a potent inhibitor of tumor growth. APLN-DM demonstrated the promotion of cell death, suppression of angiogenesis, and early prevention of liver metastasis. The modulation of APLN receptor dynamics by APLN-DM, affecting receptor affinity, internalization, and various kinase signaling pathways, suggested its potential as a therapeutic drug.

Despite the wealth of data presented, significant concerns need resolution before the paper is considered suitable for publication in EMBO Molecular Medicine.

Major concerns:

1. Explanation of Results and Methods:

The paper requires a more precise explanation of the results and methods used. Figure panels are collectively summarized without providing insight into the rationale behind specific techniques. The Figure legends need more detail (all Figures).

2. Figure 1D:

Alpha1 PDX cells: Clarification is needed on why cells expressing Alpha1 PDX produce significant APLN-13 without significant inhibition. Quantification and an explanation of the results are necessary.

3. Figure 1E,F:

Clarification is needed regarding the expected locations of these proteins in the crypts and the reasoning behind them.

Immunofluorescence Images: Splitting the channels for each protein and its location would enhance visibility. The quality of images, especially the zoomed image in Figure 1F, needs improvement.

Quantification and Normalization: The paper needs a description of how samples were normalized, and the calculation of RSI needs to be clarified. In some instances, quantifications do not align with IF images (Figure 1G ki67 channel does not change; the quantified intensity does Figure S3).

4. Correlation Coefficients: Clarification regarding the moderate to low correlation coefficients for APLNR, APLR, and Furin in CRC is needed (Figure 1 I). Include data points in the third panel and clarify statements about positive correlation. Weak correlation between CRY61/APLN and RIC8A/APLNR. (Figure 2N). Justification is required for choosing these proteins for analysis.

5. SDS-PAGE: Clarification is needed on why the APLN DM peptide runs much lower on SDS-PAGE than APLN36 despite having the same molecular weight (Figure 3B).

6. Explanation of Figures: Provide more detailed explanations, especially identifying the green signal in Figures 3C and 3E. Clarify the representation of arrowheads and whether they are representative images for a larger area.
7. Vascularization: Explain how much of the proteins were added, especially in the presence of APLN-R in the vessel wall. Clarify the contradiction in increased vascularization with APLN presence and inhibited vascularization with APLN and APLN-DM (Figures 3 G and H).
8. Structural Simulations: Provide a clearer understanding of structural simulations, including why APLN13 and APLN-DM are compared instead of wild-type APLN36 and why APLN17 is excluded (Figure 3).
9. Receptor internalization assays: Suggest redoing experiments, especially the expression of APLNR and arrestin2, both tagged with GFP. Ensure evidence of internalization using an endosomal marker and consider performing pulse-chase experiments for recycling proof. Figure 3R and Figure S10 do not provide any evidence for internalization.
10. Summary Cartoon: This cartoon requires additional refinement. After binding to APLN, the receptor undergoes internalization. However, the cartoon fails to illustrate the physiological outcomes of APLN-DM binding. Although the authors claim internalization is impaired, the cartoon depicts it as functioning correctly.

Referee #1

This study is significant for several reasons. Firstly, it demonstrates evidence for the importance of apelin in colorectal cancer. This suggests the receptor may be a potential therapeutic target in an increasing number of conditions and play a key role in tumour progression. To date Family A GPCRs have not been exploited extensively in cancer treatments.

Secondly the authors show APLN-DM, has favorable drug-like properties in pharmacodynamics, with an increased plasma half-life compared with apelin. As a peptide, APLN-DM as pharmacodynamics, has high affinity, conferring with a high degree of selectivity and potency. Most peptides with these properties are well tolerated with few side effects, so likely to be translatable into the clinic (Davenport et al 2020. doi: 10.1038/s41573-020-0062-z).

The focus of my comments is limited to the metabolism and pharmacology of apelin.

Response:

We would like to express our gratitude to the reviewer for the positive comments, which have added valuable insights to our discussion on the significance of apelin in colorectal cancer and the potential therapeutic implications of apelin-dm.

Question:

The authors do not state the source or characteristics of the antisera but does the fluorescent staining represent furin cleaved products only ie apelin 13 (as suggested in Fig 2A), rather than apelin 17 or precursor?

Response:

We thank the referee for this comment. In our study, the fluorescent staining observed in the previous Figure 2A represents all forms of apelin. We have now included this information in the revised manuscript, specifically in the Appendix Reagents and Methods.

Question:

The increase in apelin receptor is intriguing; an increase in endogenous ligand often results in a compensatory downregulation of its target GPC receptor but this pathway seems detrimentally dysregulated in cancer. The co-localisation to cells suggests apelin is acting in an autocrine/paracrine manner as one factor driving tumor growth; is this interpretation correct, can the authors comment?

Response:

We agree with the referee's interpretation. We also believe that apelin acts in an autocrine manner to sustain the malignant phenotype of cancer cells and promote angiogenesis during tumor progression. These observations have been further clarified in the revised manuscript, specifically in the Discussion section, based on both previous and new results.

We have added the following to the Discussion: *"In our study, we observed the co-expression of apelin and its receptor in both primary and metastatic tissues of colon cancer patients. This observation indicates that colon tumors sustain the capability to produce apelin and remain responsive not only in primary tumors, as previously reported (Picault et al, 2014a), but also across various stages of tumor progression, including metastasis. This suggests that the autocrine loops mediated by apelin play a pivotal role not only in the growth and survival of primary tumors but also in facilitating the metastatic dissemination of cancer cells and angiogenesis. Previous studies have reported the expression of apelin and its receptor in endothelial cells (del Toro et al, 2010), smooth muscle cells (SMC) (Pitkin et al, 2010), and cancer cells (Williams et al, 2024). Comparative analysis of these protein expressions revealed higher levels of apelin and its receptor in endothelial and SMC cells compared to cancer cells. Similarly, Furin expression was also found to be higher in endothelial and*

SMC cells compared to CT-26 and MC-38 cancer cells. We also found that, like apelin, Furin is induced by hypoxia in endothelial cells.”

Question:

Alteration of proAPLN cleavage sites contributes to reduction in colorectal tumor growth and liver metastasis. The authors mutated proAPLN cleavage sites RR60KFRR64QR to SS60KFSS64QR, creating the APLN Double Mutant (APLN-DM). The specific site for cleavage has been mutated and it might be expected that this would almost abolish generation of biologically active peptides apelin 17 and apelin 13, similar to an antagonist completely blocking an agonist. In fact, in Fig 2C and 2E there seems substantial growth in tumor volume in APLN-DM expressing cells. What is the reason? Does this indicate an alternative pathway for synthesis of biologically active peptides not necessarily cleaved at RR60KFRR64QR to SS60KFSS64QR or unmasking other agents mediating these responses? Does this have implication for translating APLN-DM is to an efficacious dose in vivo.

Response:

We appreciate the referee's insightful comment. Our initial objective was to evaluate the efficacy of apelin-dm across various mouse strains and cancer cell lines. As noted by the referee, while apelin-dm effectively inhibits tumor growth in these mice, its expression in CT-26 and MC-38 cells shows varying efficacy across different genetic backgrounds (RAG2 γ C, BALB/c, and C57BL/6). In the revised manuscript, we have addressed these differences in relation to the previous Figures 2C and 2E. We acknowledge that several factors could contribute to these observed differences, including the potential involvement of additional tumor growth mediators associated with the cancer cells used and variations in their microenvironments across different mouse strains. In the revised manuscript, we have noted that discrepancies in the higher levels of apelin produced by MC-38 cells compared to CT-26 cells may be one of the primary reasons for the observed differences. This addition is particularly relevant, given that both cell lines express apelin-dm equally (Appendix Figure S4A).

These points have been incorporated into the Results section of the revised manuscript. *“The expression of APLN-DM in these cells seems to affect their ability to mediate tumor growth with varying efficacy. Indeed, a greater inhibitory effect was observed in CT-26 cells compared to MC-38 cells, suggesting the potential implication of genetic background differences in the mice used and/or differences in the tumorigenic mediators secreted by CT-26 and MC-38 in their tumor microenvironments. Additionally, MC-38 cells produce more apelin compared to CT-26 cells (Appendix Figure S4A), suggesting the potential involvement of one or more of these differences in the lower anti-tumor efficacy of APLN-DM expression in MC-38 cells compared to CT-26 cells.”*

Question:

APLN-DM peptide represses APLN-mediated vascular network and tumor cells malignant phenotype. In these experiments 'chick CAM' was used although not defined but assumed to be chicken chorioallantoic membrane, which is highly vascularized extraembryonic membrane. Representative images of chick CAM were treated for 48 h with APLN or VEGF in absence and presence of APLN-DM and various parameters measured such as APLN-DM inhibiting both HUVEC and SMC proliferation and migration. The key question is the comparatively high concentration of 100nM used given the APLN-DM IC50 value was sub-nanomolar (0.45 nM) versus radiolabelled apelin binding. Is there an explanation for this high concentration eg there is about 50% plasma binding (Fig 8E); were the experiments conducted with serum albumen in the media? Is there any significant metabolism under these conditions? It is conventional to convert IC50 values from ligand binding using the Cheng-Prusoff equation to the Ki (affinity constant) of a ligand. The measured IC50 changes depending on the concentration of the radiolabelled ligand used.

Response

We apologize for the oversight in not including the definition of CAM in the main text; it was initially provided in the Materials and Methods section. We have now incorporated the definition into the main body of the revised manuscript. We agree with the referee's suggestion and have included details re-

garding the use of 100 nM of apelin-dm in the revised manuscript. The choice of 100 nM for the chicken chorioallantoic membrane (CAM) assay was informed by our in vitro experiments, as shown in Supplementary Figure S5. In these experiments, we evaluated the effects of varying concentrations of apelin-dm (ranging from 5 nM to 5 μ M) on in vitro angiogenesis using a tube-like structure formation assay. This assay assessed both the number of junctions and tubule length. Our findings indicated that 100 nM was the lowest concentration that elicited a significant effect, with an EC50 of 0.083 μ M. This concentration choice was further supported by similar results with Bevacizumab in the same assay, where 100 nM was the minimum concentration showing a significant effect (Appendix Figure S5). Additionally, the selection of 100 nM for the CAM assay is based on the physiological requirements of the CAM, which necessitates higher concentrations of molecules due to several factors: the CAM is a highly vascularized membrane that can impede molecule diffusion; applied molecules may be rapidly metabolized or cleared by the embryo's physiological processes; diffusion into surrounding tissues and fluid dynamics of the membrane can lead to dilution; and, as noted by the referee, the 50% plasma binding of apelin-dm means that only half of the drug is immediately active, potentially affecting the duration of action.

We have added the following paragraphs and associated new references to the Results section: “*We next synthesized the apelin-dm peptide to explore its therapeutic potential. In apelin-36, the sequence "RRKFRR" has been substituted with "SSKFSS", where the R residues have been replaced by S. R is positively charged at neutral pH with approximate molecular (MW) of 174.2 Da. Whereas S is neutral with MW of 105.1 Da. This difference in molecular weight and charge properties likely contributes to the lower migration of apelin-dm peptide on SDS-PAGE than apelin-36 (Fig. 3A, B). First, the impact of varying concentrations of apelin-dm on in vitro angiogenesis was assessed through a tube-like structure formation assay (Appendix Figure S5A, B). This assay, which evaluated both the number of junctions and tubule length, revealed that apelin-dm disrupted the formation of capillary-like structures in HUVECs in a concentration-dependent manner, with 100 nM being the lowest concentration that elicited a significant effect (Appendix Figure S5A, B). Comparative analysis with Bevacizumab revealed similar inhibitory effects, with EC50 values of 0.130 μ M for Bevacizumab and 0.083 μ M for apelin-dm, respectively (Appendix Figure S5C). Previously, Bevacizumab was shown to exhibit inhibitory effects at concentrations around 100 nM, both in vitro and in CAM assays (Ljoki et al, 2022; Ademi et al, 2021), suggesting a similar anti-angiogenic effect for apelin-dm and Bevacizumab at this concentration”*

In Discussion we added:“*Our investigation into the pharmacokinetic properties of apelin-dm, which revealed approximately 50% plasma protein binding, presents several advantages that warrant consideration in therapeutic contexts. These binding properties strike a balance between a rapid onset of action and a prolonged duration of therapeutic effect. While a portion of the drug is bound to plasma proteins, ensuring a reservoir for sustained release, an equally substantial fraction remains unbound and pharmacologically active. This equilibrium allows for the swift attainment of therapeutic concentrations in the bloodstream while maintaining a steady supply of bioavailable drug molecules over an extended period. Consequently, apelin-dm may offer a favorable pharmacokinetic profile suitable for conditions requiring both immediate intervention and sustained therapeutic coverage. Highly protein-bound drugs may be susceptible to drug interactions and alterations in pharmacokinetics due to competition for plasma protein binding sites, potentially leading to suboptimal therapeutic outcomes or an increased risk of toxicity. Conversely, drugs with minimal protein binding may exhibit rapid clearance and reduced bioavailability, necessitating frequent dosing regimens or higher doses to maintain therapeutic efficacy. Apelin-dm's 50% plasma protein binding confers a degree of flexibility, striking a harmonious balance between efficacy and safety, and minimizing the likelihood of pharmacokinetic fluctuations that could compromise treatment efficacy. Previously, peptides with similar properties have been well tolerated with few side effects, suggesting that apelin-dm is likely to be translatable into clinical practice (Davenport et al, 2020).”*

Question:

The apelin receptor is unusual in being activated by two endogenous peptides, apelin and Elabela that have distinct amino acid sequences. (see Ref 3 in manuscript Read C, et al. 2019). Ela is also expressed in humans, colocalising for example with apelin in endothelial cells and circulates in the plasma. It

acts on the apelin receptor in a similar way to apelin in conditions such as pulmonary arterial hypertension (see Yang et al 2017. doi: 10.1161/CIRCULATIONAHA.116.023218). It has also been linked to cancer (Sharma et al doi: 10.2174/1389450123666220826160123). In cancer, compounds developed as therapeutic agents are likely to need to modulate the action of both peptides in binding to the receptor. Have the authors determined in functional assays whether APLN-DM reduces the activity of Elabela in a similar way to apelin?

Response:

We thank the referee for this valuable suggestion. In this revised version of the manuscript, we have included new results (Figures 4D and 4E) demonstrating that apelin-dm not only inhibits apelin functions but also reduces the binding of Elabela (Ela) to the apelin receptor. This, in turn, attenuates the activity of Elabela in cells, as evidenced by the reduced ERK activation mediated by Elabela in the presence of apelin-dm. This additional information highlights the efficacy of apelin-dm and its interaction with the apelin receptor, particularly in relation to Elabela, the second ligand of the apelin receptor.

We have added these details to the Results section: “Apelin-dm modulates Elabela activity and its receptor binding. Alongside apelin, Elabela can activate the apelin receptor, boasting a distinct amino acid sequence from apelin (Read et al, 2019). It colocalizes with apelin in endothelial cells and circulates in plasma. Elabela shares functional roles with apelin in angiogenesis and cancer (Yang et al, 2017). To investigate whether apelin-dm affects the interaction between Elabela and the apelin receptor, we conducted PWR and ligand affinity analyses. Our findings showed that apelin-dm competes with Elabela's binding to the apelin receptor, as evidenced by an increased KD from 27 ± 1 to 56.6 ± 2.6 in the presence of apelin-dm (Fig. 4D). Additionally, ERK activation analysis in HEK-apelin receptor cells revealed that apelin-dm inhibited Elabela-induced ERK activation (Fig. 4E). These results suggest that apelin-dm competes with apelin and Elabela for the apelin receptor and influences their functions.”

Minor Points

To improve interpreting figures please add number of replicates to the figure legends where these are missing. Eg Supplemental Figure 3. The percentage of relative staining intensity of APLN, Ki67, APLNR and Furin for indicated tissue deduced from (Fig. 1G, 1H) . All values represent the mean{plus minus}s.e.m. P values by two-tailed unpaired t-test. Nomenclature. The International Union of Basic and Clinical Pharmacology recommends (Ref 3 in manuscript Read C, et al. 2019) that APLN and APLNR are used to refer to the human apelin gene and the gene encoding the apelin receptor, respectively. The peptide apelin and protein apelin receptor, should ideally be spelt out in full to avoid confusion with the gene names (eg p14).

Response:

Thank you for your valuable feedback and suggestions for improving the interpretation of our figures. In this version of the manuscript, we have made the necessary changes to our figure legends to include the number of replicates where missing. Additionally, we have noted the nomenclature recommendations regarding the usage of APLN and APLNR to refer to the human apelin gene and the gene encoding the apelin receptor. We have also adapted this nomenclature to APLN-DM to refer to the gene and apelin-dm to refer to the protein

Referee #2

Question:

This study has been validated using a mouse model, which is adequate. These results support that APLN-DM has the potential to be an effective treatment for metastatic colorectal cancer. Since the

findings are novel and potentially significant, the authors should be encouraged to address the key concerns detailed below.

Response:

We thank the referee about this positive comment

Major comments:

Question:

- 1. APLN and APLNR are known to be highly expressed in vascular endothelial cells in tumor tissue. It is necessary to clarify whether Furin is also highly expressed in vascular endothelial cells.*
- 2. Although APLN-DM shows "increased anchorage-independent growth" for CT26 and MC38, the expression levels of APJ in these cells should be shown in comparison to vascular endothelial cells. Also, the absolute expression levels of Furin in CT26, MC38, SMC, and HUVECs used in cell culture experiments should be shown.*

Response:

1-In this new version of the manuscript, we added new data describing the high expression of Furin in vascular endothelial cells compared to colon cancer cells. This includes data in Appendix Figure S2 and Appendix Figure S6F .

We added this information to the Results section: "Given the co-expression of apelin, apelin receptor and Furin, that we identified in colon tissues, colon carcinoma cells, and endothelial cells (Appendix Figure S2), we sought to evaluate their expression and colocalization in both normal and colon cancer tissues."

2-. As requested, we also included data in this manuscript that analyze the expression of APJ levels in CT26 and MC38, shown in comparison to vascular endothelial cells. Additionally, we provided new results regarding the absolute expression levels of Furin in CT26, MC38, SMC, and HUVECs in this version of the manuscript.

This information has been added to the Results section: "*Real-time PCR showed expression of apelin and its receptor in SMCs and HUVECs (Appendix Figure S6F). Analysis of the apelin receptor at the protein level revealed its high expression in HUVEC cells compared to CT-26 and MC-38 cells (Appendix Figure S6G). Further analysis revealed high expression of Furin in SMCs (5193 copies/cell) and HUVECs (6013 copies/cell), while it was lower in cancer cells, with 3667 and 3713 copies/cell for CT-26 and MC-38, respectively (Fig. 3C).*"

" In legend Figure 3C , we added "*C. Furin expression analysis expressed as copy number per cell in smooth muscle cells (SMC), endothelial cells (HUVEC), and cancer cells CT-26 and MC-38 using droplet digital polymerase chain reaction (ddPCR) assay.*"

Question:

- 3. The authors should clearly indicate whether the anti-tumor effect of APLN-DM is mainly due to a direct effect on tumor cells or an indirect effect through inhibition of angiogenesis. The use of APLNR - KD tumor cells can clarify this point.*

Response:

In our study, we assessed the effect of apelin-dm on tumor cells and endothelial cells to evaluate its impact on the malignant phenotype of tumors and angiogenesis. As illustrated by various in vitro and in vivo studies described in the current version (Figures 2H-J, Figures 3D-I, Figure 5H), apelin-dm appears to have a direct effect on both tumor and endothelial cells. Indeed, treatment of cancer cells with apelin-dm inhibits their proliferation, migration, and induces apoptosis (Appendix Figure S4). Similarly, treatment of endothelial cells with apelin-dm also affects their ability to induce tube formation in vitro (Appendix Figure S5 and Appendix Figure S6). Therefore, in this version of the manuscript, we have further clarified our statement regarding the role of apelin-dm in repressing tumor progression and angiogenesis, and we added a paragraph that summarizes these observations in the Results section as follows: "*Overall, these findings highlight the multifaceted inhibitory effects of apelin-*

dm on vascular networks and the malignant phenotype of colon cancer cells, underscoring its potential therapeutic application."

Question:

4. *As described by the authors in the introduction, Apelin expression is known to be induced in a hypoxic environment. Is Ferin expression similarly induced by hypoxia?*

Response:

In this new version of the manuscript, we added new data on the effect of hypoxia on Furin expression in endothelial cells. This information has been added to the Results section : "*Hypoxia is a common feature of the microenvironment in almost all solid tumors and is frequently associated with angiogenesis and cancer growth, including CRC (Chen et al, 2023) and has been reported to induce apelin expression in endothelial cells (Andersen et al, 2011). Analysis of Furin expression in endothelial cells cultured under normoxic (21% O₂) and hypoxic (1% O₂) conditions revealed that hypoxia is also a strong inducer of Furin mRNA in endothelial cells, with increased expression observed at all tested hypoxia exposure times ranging from 4 to 24 hours (Fig. 1N)*" In Materials and Methods we added: "For hypoxic experiments, endothelial cells were serum-starved and placed in a sealed, humidified chamber maintained at 1% O₂ and 5% CO₂ for different time periods ranging from 4 hours to 24 hours" In Legend to Figure 1 "*N, Kinetics of Furin mRNA accumulation in HUVEC cells cultured in normoxia (21% O₂) or hypoxia (1% O₂) for various time periods as indicated and data are representative of three independent experiments with values that represent the mean±s.e.m. Significant differences P were determined by Two-way ANOVA.*"

Question:

5. *In the suppression of tumor growth induced by APLN-DM treatment, is the reduction in Ki67-positive cells an indirect effect of the reduced number of vessels? By co-staining CD31 and Ki67, it can be shown whether only the proliferation of tumor cells in the avascular region is reduced.*

Response:

As requested by the referee, we generated new data to assess whether apelin-dm has a direct effect on endothelial cells within the developed tumors. We analyzed its impact on Ki67 and CD31 expression in tumors using co-staining for CD31 and Ki67. Our findings suggest that apelin-dm primarily affects Ki67 staining in cancer cells rather than in endothelial cells.

We have updated the Results section to reflect these observations as follows: "*The CD31 and KI-67 staining of tumor sections revealed decreased angiogenic (Fig. 5E, H, I, J) and proliferative indices (Fig. 5F, H, K, L) in apelin-dm-treated mice inoculated by CT-26 and MC-38 tumor cells, respectively. Immunohistochemistry analysis showed increased pro-apoptotic protein BIM levels in apelin-dm-treated mice tumors (Fig. 5G, M, N), indicating enhanced cell death. Co-staining for KI-67 and CD31 in CT-26 and MC-38 tumors (Fig. 5H) derived from apelin-dm-treated mice showed that the reduced KI-67 staining was predominantly observed in the cancer cells within the tumors.*"

Question:

6. *Although it is described on page 12 as "E-selectin mRNA increased at 2h, returning to basal levels after 4-6 hours," E-selectin mRNA expression remains high even after 6 hours in Figure 6A. In addition, why is there a difference in the duration of E-selectin expression between HUVECs, HMEC-1, and LSECs?*

Response:

We agree with the referee and have corrected this statement accordingly in this version of the manuscript. Regarding the differences in the duration of E-selectin expression among HUVECs, HMEC-1, and LSECs, these variations could be attributed to inherent differences in the regulatory

mechanisms governing gene expression across different cell types. Each cell type may have distinct signaling pathways, transcription factors, or epigenetic modifications that affect the duration and intensity of E-selectin expression in response to stimuli. Additionally, differences in cellular responses might arise from variations in cell culture conditions, passage numbers, or even subtle genetic disparities between cell lines.

We have updated the Results section to reflect these points: “ We investigated apelin's role by exposing human hepatic endothelial cells (LSEC) to 100 nM apelin (Fig. 6A). E-selectin mRNA levels increased at 2 hours, peaked at 6 hours, and remained elevated even after 8 hours of apelin treatment. At the protein level, E-selectin expression peaked at 24 hours, followed by a decrease at 32 hours (Fig. 6B). Apelin also induced E-selectin expression in endothelial cells HMEC-1 and HUVEC (Fig. 6C, D). However, the dynamics of apelin's effect differed in these endothelial cells compared to LSECs. While E-selectin expression initially increased after 2 hours of incubation with apelin, it subsequently declined but remained elevated compared to baseline levels after 6 hours. This divergence may arise from inherent variations in the regulatory mechanisms governing E-selectin expression, suggesting potential differences in the regulatory pathways and signaling mechanisms among these cell types in response to apelin.”

Question:

7. MM54 appears to have a similar metastatic inhibitory effect as APLN-DM. The author should test whether APLN-DM is effective for tumor growth compared to existing APJ antagonists.

Response:

As requested by the referee, we compared the effect of apelin-dm on tumor growth to that of the APJ antagonist ML221. We found that apelin-DM is as effective as ML221 in inhibiting tumor growth. We have included this comparison in the Results section as follows: “ To assess the tumor-suppressive potential of apelin-dm, mice with subcutaneously injected CT-26 and MC-38 cells were treated intraperitoneally with apelin-dm (30 mg/kg), apelin receptor antagonist ML221 or a saline vehicle control (Fig. 5A). Tumor growth, apoptosis, proliferation, and/or the angiogenic index of the developed tumors were first evaluated. Apelin-dm treatment significantly reduced tumor growth compared to controls, as observed in daily monitored tumor volumes for both mice tumor models (Fig. 5B, C). The anti-tumor effect of apelin-dm was comparable to the effect of the apelin receptor antagonist ML221 (Fig. 5D).”

Question:

8. The authors should analyze the mechanism of inhibition of VEGF-induced angiogenesis by APLN-DM.

Response

As requested by the referee, we investigated the mechanism of inhibition of VEGF-induced angiogenesis mediated by apelin-dm and have included new results in this version of the manuscript. We have updated the Results section to reflect these findings as follows: “ To investigate the mechanism by which apelin-dm represses VEGF-induced angiogenesis, we first analyzed its effect on HUVEC proliferation and migration in response to VEGF. As depicted in Fig. 3J, K, while VEGF significantly promoted HUVEC proliferation and migration, the presence of apelin-dm suppressed these effects. Furthermore, analysis of ERK activation, a signaling pathway mediated by VEGF and involved in HUVEC proliferation and migration, showed that while VEGF induced ERK activation, treatment with apelin-dm attenuated this effect (Fig. 3M, N). This suggests that apelin-dm interferes with the VEGF signaling pathway, which may explain the reduced angiogenesis mediated by VEGF in the presence of apelin-dm”

Minor comments:

1. Why are there no dots on the right graph in Figure 1I?

Response:

Thank you for pointing this out. The absence of dots on the right graph in Figure 1I was due to an issue with the conversion from PPT to PDF format. We have corrected this figure in the revised manuscript.

2. *What does RSI show in the graphs in Figure 2I and J?*

Response:

Thank you for your observation. We have added the following information to the legend for Figure 2: RSI stands for Relative Staining Intensity

3. *Text in Figure 2N is not clear.*

Response:

Thank you for highlighting this issue. We have clarified the text in this version of the manuscript.

4. *In the Introduction, Apelin is described as "promoting epithelial cell growth, proliferation, and migration," but the referenced paper suggests a direct effect on endothelial cells.*

Response:

Thank you for pointing this out. We have updated the Introduction to include additional references that better reflect the direct effects of apelin on endothelial cells, as suggested by the cited paper.

5. *Figure 3R is not described in the manuscript.*

Response:

Thank you for bringing this to our attention. We have now cited Figure 3R in the text of this version of the manuscript.

Referee #3

Despite the wealth of data presented, significant concerns need resolution before the paper is considered suitable for publication in EMBO Molecular Medicine.

Major concerns:**Question :**

1. *Explanation of Results and Methods: The paper requires a more precise explanation of the results and methods used. Figure panels are collectively summarized without providing insight into the rationale behind specific techniques. The Figure legends need more detail (all Figures).*

Response:

We have made every effort to address the concerns raised. We acknowledge that the previous version of the manuscript lacked precise explanations of the results and methods. In this revised version, we have introduced the rationale behind specific techniques and added more detailed information to the figure legends to enhance clarity across all figures, while adhering to the manuscript's length constraints

Question:

2. *Figure 1D: Alpha1 PDX cells: Clarification is needed on why cells expressing Alpha1 PDX produce significant APLN-13 without significant inhibition. Quantification and an explanation of the results are necessary.*

Response:

We have now included additional information about the effect of Alpha1 PDX expression on apelin precursor cleavage and its quantification.

This information has been added to the Results section as follows: *“To assess the efficiency of PC inhibitors on proapelin processing by endogenous Furin, we treated HEK293A cells with the synthetic Furin inhibitor decanoyl-Arg-Val-Lys-Arg-chloromethyl-ketone (CMK, 10 μM) (Seidah & Prat, 2012; Bontemps et al, 2007) or expressed in these cells the Furin inhibitors, including the Furin prodoamine (P-p-Furin) (Scamuffa et al, 2014) and α1-PDX (Descarpentrie et al, 2022; Jean et al, 1998) (Fig. 1D). As demonstrated by Western blot analysis, transfection of HEK293A cells with a vector encoding proapelin (Control) resulted in ~95% processing. Treatment of cells with CMK or cotransfection with Furin inhibitors revealed that processing of proapelin is significantly blocked by CMK (~10% processing) and α1-PDX (~50% procession), P-p-Furin (~30% processing). In contrast, wild-type α1-antitrypsin failed to inhibit proapelin processing (Fig. 1D). In vitro enzymatic assays (Sfafi et al, 2014; Basak et al, 2010) confirmed increased enzymatic activity with PCs expression and its reduction by Furin inhibitors expression (Appendix Figure S1B), supporting Furin as primary PC mediating proapelin cleavage, producing apelin-17 and apelin-13.”*

In the figure legend for Figure 1D we added: *“D, Inhibition of proapelin processing was assessed by Western blot analysis in media of HEK293A cells co-transfected with empty vector (None), or vectors containing preproapelin (Control) and indicated Furin inhibitors, or treated with the PC inhibitor decanoyl-Arg-Val-Lys-Arg-chloromethyl-ketone (CMK, 10 μM). The bars show the corresponding percentage of band intensities deduced from the ratio of those of apelin/(proapelin + apelin) and indicate the level of mature apelin accumulation (%). Data are representative of 3 independent experiments and all values represent the mean ± s.e.m. P values by two-tailed unpaired t-test. ns: not significant.”*

Question :

3. Figure 1E,F: Clarification is needed regarding the expected locations of these proteins in the crypts and the reasoning behind them.

Response:

As requested, we have clarified the expected locations of these proteins in the crypts and the rationale behind their distribution.

This information has been added to the Results section as follows: *“In a healthy colon, crypts contain stem cells responsible for the constant renewal and differentiation of the epithelial lining. While in cancerous tissue, these processes are disrupted, leading to uncontrolled cell proliferation, aberrant differentiation, and loss of normal crypt architecture (Humphries & Wright, 2008). Given the co-expression of apelin, apelin receptor and Furin, that we identified in colon tissues, colon carcinoma cells, and endothelial cells (Appendix Figure S2), we sought to evaluate their expression and colocalization in both normal and colon cancer tissues. To do this, we first conducted immunohistochemical staining on a set of matched tissue sections, including normal human colon mucosa, primary colon tumors, and corresponding colorectal liver metastases from 35 patients. All these proteins were expressed in non-cancerous colon tissues, primarily localized to the base of the colon crypts (Fig. 1E, F). The expression of apelin, apelin receptor and Furin in the colon crypts highlights their potential roles in maintaining the dynamic environment of the intestinal epithelium. In cancerous tissues, the loss of crypts was associated with a modified expression pattern of apelin, apelin receptor, and Furin, suggesting that these molecules may play roles in tumor progression and possibly in the tumor microenvironment.”*

Question:

Immunofluorescence Images: Splitting the channels for each protein and its location would enhance visibility. The quality of images, especially the zoomed image in Figure 1F, needs improvement.

Response :

In this new version, we have split the channels for each protein to enhance visibility and provided high-resolution images. We have also improved the quality of the zoomed image in Figure 1F. Thank you for your feedback

Question :

Quantification and Normalization: The paper needs a description of how samples were normalized, and the calculation of RSI needs to be clarified. In some instances, quantifications do not align with IF images (Figure 1G ki67 channel does not change; the quantified intensity does Figure S3).

Response:

The Relative Staining Intensity (RSI) was calculated using ImageJ by measuring the integrated density of staining (I_{stain}) within manually selected regions of interest (ROIs), subtracting the background staining intensity (I_{bg}), and normalizing by the area of the ROI (A_{stain}). The RSI was computed using the formula: $RSI = (I_{\text{stain}} - I_{\text{bg}}) / A_{\text{stain}}$. This description has been added to the Supplemental Methods file. We acknowledge the discrepancy between the Ki67 channel in the previous Figure 1G and the quantified intensity shown in Figure S3. After reviewing the data, we confirm that the quantification reflects accurate measurements, while the images in Figure 1G may not fully capture the subtleties observed in the quantified data. We have provided new figures in the manuscript to ensure consistency and accuracy.

Question :

4. Correlation Coefficients: Clarification regarding the moderate to low correlation coefficients for APLNR, APLR, and Furin in CRC is needed (Figure 1 I). Include data points in the third panel and clarify statements about positive correlation. Weak correlation between CRY61/APLN and RIC8A/APLNR. (Figure 2N). Justification is required for choosing these proteins for analysis.

Response:

We have revised our statement regarding the correlation coefficients for APLNR, APLR, and Furin in CRC, as shown in Figure 1I. Data points have been included in the third panel to enhance clarity. Additionally, we have clarified our statements about positive correlation and addressed the weak correlation between CRY61/APLN and RIC8A/APLNR in Figure 2N. We have also provided a justification for selecting these proteins for analysis.

These updates are reflected in the Results section of the manuscript: *“Quantitative analysis of immunohistochemistry staining in primary colorectal cancer (CRC) and metastatic colorectal cancer (mCRC) tumors, along with their respective normal colon tissues, revealed a consistent increase in the average staining levels of these proteins in the analyzed tumors (Appendix Figure S3). Moderate correlation coefficients (ranging from 0.3 to 0.5) were observed between the expression levels of apelin and KI-67, apelin receptor and KI-67, as well as Furin and KI-67 in primary CRC (Fig. 1I) and their corresponding metastatic liver lesions (Fig. 1J). These positive correlations underscore the potential roles of apelin, apelin receptor, and Furin in CRC progression and metastasis.”* And “). We next explored the correlation between the expression of APLN and APLNR with genes that are dysregulated by APLN or APLN-DM expression in cancer cells, and involved in prognosis of colon cancer patients. Among the dysregulated molecules, we observed varying degrees of correlation with APLN and APLN-R. Specifically, we found a low positive correlation between APLN and CYR61 ($r = 0.16$), a stronger correlation between APLNR and CYR61 ($r = 0.68$), a low correlation between APLNR and RIC8A ($r = 0.20$), and a moderate correlation between APLNR and TUBA1A ($r = 0.58$) in tumor patients (Fig. 2N).”

Question:

5. SDS-PAGE: Clarification is needed on why the APLN DM peptide runs much lower on SDS-PAGE than APLN36 despite having the same molecular weight (Figure 3B).

Response:

In APLN 36, the sequence "RRKFRR" has been substituted with "SSKFSS," where the 4R (arginine) residues have been replaced by 4S (serine) residues. Arginine is positively charged at neutral pH, whereas serine is neutral. The approximate molecular weight of serine is 105.1 Da, and that of arginine is 174.2 Da. This difference in molecular weight and charge properties likely contributes to the observed phenomenon where the apelin-dm peptide migrates much lower on SDS-PAGE than apelin 36. As requested we added this information in the Results section: "*We next synthesized the apelin-dm peptide to explore its therapeutic potential. In apelin-36, the sequence "RRKFRR" has been substituted with "SSKFSS", where the R residues have been replaced by S. R is positively charged at neutral pH with approximate molecular (MW) of 174.2 Da. Whereas S is neutral with MW of 105.1 Da. This difference in molecular weight and charge properties likely contributes to the lower migration of apelin-dm peptide on SDS-PAGE than apelin-36 (Fig. 3A, B)..*" In legend to Figure 3B we added: "*B, SDS gel analysis of wild type proapelin and apelin-dm peptide cleavage by Furin. Apelin-13 was added for comparison. The difference in molecular weight and charge properties of the aa R and S in apelin-dm contributes to the lower migration of apelin-dm peptide than apelin-36.*"

Question:

6. *Explanation of Figures: Provide more detailed explanations, especially identifying the green signal in Figures 3C and 3E. Clarify the representation of arrowheads and whether they are representative images for a larger area.*

Response:

We have now provided a more detailed explanation regarding the results presented in Figures 3C and 3E. The green signal in these figures represents vasculature stained by isolectin B4. We have clarified that the arrowheads indicate newly formed small vessels. Additionally, Figures 3C and 3E are representative images of a larger area shown in Appendix Figure S7A, B. We have added the following information to the Results section: "*We also tested the anti-angiogenic effect of apelin-dm using the chick chorioallantoic membrane (CAM) assay (Fig. 3D, E). Apelin-dm treatment impaired both basal CAM angiogenesis and CAM angiogenesis induced by apelin. Examination of the CAM vasculature by isolectin B4 staining showed that while apelin induced vessel formation, apelin-dm mainly affected the formation of small vessels (Fig. 3D, E, yellow arrows, and Appendix Figure S7A).*"

Question:

7. *Vascularization: Explain how much of the proteins were added, especially in the presence of APLN-R in the vessel wall. Clarify the contradiction in increased vascularization with APLN presence and inhibited vascularization with APLN and APLN-DM (Figures 3 G and H).*

Response:

We employed a mouse neonatal retinal model to perform a quantitative in vivo angiogenesis assay, aiming to assess the effects of apelin (10 mg/kg), apelin-dm (10 mg/kg), and their combination on neoangiogenesis. Our results show that apelin alone effectively induces angiogenesis. In contrast, apelin-dm alone significantly inhibits both basal angiogenesis and the angiogenic response induced by apelin. This inhibitory effect of apelin-dm is attributable to its high affinity for the apelin receptor, which competes with apelin and prevents it from binding to its receptor. As a result, apelin-dm functions as an antagonist, blocking apelin's pro-angiogenic activity. This interaction affects the downstream signaling pathways and explains the observed differences in vascularization. The detailed findings regarding the effects of apelin and apelin-dm, as well as their receptor interactions, are further elucidated in various figures throughout the manuscript. We have updated the Results section to include this comprehensive explanation: "*Additionally, we used the neonatal mouse retinal model as an in vivo assay to examine the effect of apelin-dm on neoangiogenesis. At birth, the retina is avascular, and a superficial vascular plexus grows from the center to the periphery during the first week after birth (Selvam et al, 2018). Isolectin B4 staining of the retinal vasculature at P5 revealed that apelin administration significantly increased vascular sprouting in mice compared to controls (Fig. 3F, G). In both untreated and apelin-treated mice, sprouting vessels were reduced by apelin-dm administration.*"

Additionally, apelin-dm hindered VEGF-induced endothelial cell sprouting, and disrupted angiogenesis in the CAM assay (Fig. 3H, I and Appendix Figure S7B).”

Question:

8. Structural Simulations: Provide a clearer understanding of structural simulations, including why APLN13 and APLN-DM are compared instead of wild-type APLN36 and why APLN17 is excluded (Figure 3).

Response:

In the updated version of the manuscript, we have included new figures that compare apelin-DM with apelin-13, apelin-17, and apelin-36. This expanded comparison provides a more comprehensive understanding of the structural differences and interactions among these peptides that we added in the Results section : “*Computational docking analysis of apelin peptides and apelin-dm with apelin receptor: Predicting binding modes and affinity^[SEP] Aligning apelin-13 with the experimental structure of the AMG3054 peptide mimetic in the presence of the active apelin receptor cryo-EM structure suggests that the C-terminal region of apelin-13 inserts into a deep pocket (Fig. 3Q). Additional CABS-flex simulations induced minor changes in the N-terminal region of apelin-13, while the C-terminus remained deeply anchored compared to the starting position of the peptide. Apelin-dm, apelin-36, and apelin-17, docked using CABS-dock, wrapped around the apelin receptor, showing additional favorable interactions in the peptide C-terminal region compared to apelin-13, particularly for apelin-dm and apelin-36. Models of the apelin receptor with apelin-dm exhibited higher total binding energy values than models with apelin-13 or apelin-17, indicating superior binding of apelin-dm. The predicted binding energy values between apelin-36 or apelin-dm and the receptor were similar (Fig. 3R). APOP identified allosteric sites, with the top site matching the peptide-binding site in the C-terminal region (allosteric site 1) (Fig. 3Q). Another allosteric site, accessible to apelin-13, apelin-17, apelin-36, and apelin-dm, was identified and labeled allosteric site 2. A third allosteric pocket (allosteric site 3) can also interact with the N-terminal region of apelin-36 or apelin-dm (Fig. 3Q). The predicted binding energy scores favored apelin-dm or apelin-36 over apelin-13 or apelin-17 (Fig. 3R).*

. In discussion, we added : *Our investigation unveiled that the inhibition of proapelin processing appears to heighten the affinity of the apelin receptor for apelin-dm. This observation was substantiated through comprehensive docking computations and structural analyses, which were further validated by radioligand binding assays. Constructing the apelin-dm-apelin receptor complex based on the X-ray structure of the apelin receptor-apelin peptide mimetic AMG3054 revealed a distinctive wrapping of apelin-dm (or apelin-36) around the apelin receptor, with a notable concentration of major binding energy in the C-terminal region of the peptide, particularly at the last Phe residue. Intriguingly, the residues cleaved by Furin exhibited a tendency to point away from the receptor, forming limited contact points with the protein. The predicted binding scores, favoring apelin-dm (and apelin-36) over the shorter apelin-13, aligned with experimental data, supporting the additional favorable contacts for the longer 36-residue peptides. The predicted interaction energy values are relatively similar between apelin-36 and apelin-dm, but apelin-dm is more stable and not cleaved as compared to the wild-type apelin-36, which is rapidly cleaved upon synthesis. This allows for major competition on the apelin receptor to occur between mature apelin rather than apelin-36. The binding pocket crucial for interaction with the C-terminal region of apelin was predicted to be allosteric, corroborating with observed flexibility and structural changes in this region of the receptor. While apelin-13 was predicted to engage two allosteric sites, apelin-dm (or apelin-36) was anticipated to establish further contacts with an additional allosteric site situated on the receptor's side. This suggests the possibility that the two peptides (e.g., apelin-13 and apelin-dm) induce slightly different structural changes upon binding. The precise 3D predictions of such changes remain challenging, acknowledging the inherent flexibility of the N-terminal of apelin-dm, yet the overall position of the peptides on the apelin receptor seems reasonable. It is important to note that the exact high atomic resolution of apelin-dm remains unknown. Utilizing a plasmon waveguide resonance (PWR) assay, we discerned that apelin-dm and apelin-13 elicited distinct conformational*

changes in the receptor, indicative of potential differences in apelin receptor affinity and in the likely interactions with the predicted allosteric sites for apelin-13 versus apelin-dm.”

Question:

9. Receptor internalization assays: Suggest redoing experiments, especially the expression of APLNR and arrestin2, both tagged with GFP. Ensure evidence of internalization using an endosomal marker and consider performing pulse-chase experiments for recycling proof. Figure 3R and Figure S10 do not provide any evidence for internalization.

Response:

In response to the referee's suggestion, we have re-evaluated and conducted additional experiments to address the concerns regarding receptor internalization. We have provided new figures illustrating the internalization of the apelin receptor in the presence of apelin and apelin-dm. To enhance the evidence of internalization, we included clathrin detection, which allows us to monitor the internalization of the apelin receptor more effectively. The updated data, including the new figures, are detailed in the Results section. We have added the following text to clarify these findings: “ *Previously, apelin was reported to cause clathrin-mediated apelin receptor internalization (He et al, 2016; Reaux et al, 2001; El Messari et al, 2004a) and translocation of β -arrestin to the cell surface, indicating translocation to the phosphorylated apelin receptor (Lee et al, 2010). After apelin-induced internalization, the apelin receptor can either be recycled to the cell surface or be degraded in lysosomes (Lee et al, 2010). Therefore, we compared the effects of apelin and apelin-dm on the cellular recycling of the apelin receptor. To facilitate the functional analysis of receptor internalization, we used U2OS cells stably co-expressing the human apelin receptor and β -arrestin 2-GFP. Stimulation of these cells with Tamara-apelin for 30 minutes induced receptor internalization, as visualized by the appearance of a punctuated pattern of red (Tamara-apelin) and green (apelin receptor/arrestin 2-GFP) fluorescence (Fig. 4A). In contrast, cells treated with apelin-dm showed reduced Tamara-apelin receptor internalization (Fig. 4A). The use of the same cells revealed that apelin-dm, like apelin, induces receptor internalization that colocalizes with clathrin (Fig. 4B). Similar to apelin, apelin-dm dose-dependently increases the Bioluminescence Resonance Energy Transfer (BRET) signal between the apelin receptor and Rab5, confirming the ability of both apelin and apelin-dm to promote apelin receptor internalization (Appendix Figure S10A). Using HEK293A cells stably expressing apelin receptor-EGFP fusion protein, we found that six hours after apelin-13 washout, the apelin receptor seemed to be mainly returned to the cell surface. In contrast, in cells treated with apelin-dm, the intracellular vesicles of the apelin receptor remained internalized (Appendix Figure S10B), suggesting delayed or blocked recycling of the apelin receptor in the presence of apelin-dm. ”*

We added in the Methods “ *Bioluminescence Resonance Energy Transfer (BRET) Measurement. For the BRET assays, phenol red-free medium was removed from HEK293T cells transiently transfected with apelin receptor-Rluc and Rab5-YFP, and replaced with PBS containing calcium and magnesium. The assay was initiated by adding 10 μ l of the cell-permeant Renilla luciferase substrate, coelenterazine h, to achieve a final concentration of 5 μ M. Five minutes later, apelin and apelin-dm were added to assess their activity. Plate readings were taken 15 minutes after substrate addition. BRET signals were collected using a Mithras LB940 instrument, which integrates signals sequentially in the 465-505 nm and 515-555 nm windows using appropriate bandpass filters, managed by MicroWin 2000 software. Net BRET signals were calculated by subtracting the BRET signal from cells expressing only Rluc-tagged apelin receptor from those co-expressing both Rluc-tagged apelin receptor and YFP-tagged Rab5. ”*

Question:

10. Summary Cartoon: This cartoon requires additional refinement. After binding to APLN, the receptor undergoes internalization. However, the cartoon fails to illustrate the physiological

outcomes of APLN-DM binding. Although the authors claim internalization is impaired, the cartoon depicts it as functioning correctly.

Response:

We appreciate the feedback on the Summary Cartoon. In response, we have provided a new schematic representation. The revised cartoon now more accurately illustrates the physiological outcomes following apelin-dm binding, including the impaired internalization of the receptor as observed in our study. We have refined the details to better align with our findings.

We would like to express our sincere gratitude to all the reviewers for their insightful comments, which have greatly enhanced the clarity and readability of our work.

Majid Khatib, Ph.D.
INSERM; BRIC

7th Oct 2024

Dear Dr. Khatib,

Thank you for submitting your revised study, and please accept my apologies for the delay in getting back to you as I was traveling for work. We have now received the reports from the two referees who evaluated your revised manuscript. As you will see from the reports below, these referees are satisfied with the revisions, and I will therefore be able to accept your manuscript once the following editorial issues will be addressed:

1/ Manuscript text:

- Please remove the highlights in the text and only keep in track changes mode any new modification.
- Would you consider changing the title to: 'Repression of apelin Furin cleavage sites provides antimetastatic strategy in colorectal cancer'?
- The following email bounced, please correct: (lalloue.fabrice@unilim.fr).
- Please note that all corresponding authors are required to supply an ORCID ID for their name upon submission of a revised manuscript. An ORCID ID is missing for Geraldine Siegried.
- Please remove "not shown" (p. 48). As per our guidelines on "Unpublished Data", the journal does not permit citation of "Data not shown". All data referred to in the paper should be displayed in the main or Expanded View figures.
- Methods:
 - o Thank you for providing a reagents and tools table, please remove it from the manuscript and upload it as a separate file.
 - o Please merge the Appendix methods with the main text methods.
 - o Patient samples: please include a statement that the experiments conformed to the principles set out in the WMA Declaration of Helsinki and the Department of Health and Human Services Belmont Report.
 - o Ethics: please confirm that all experiments were performed in accordance with relevant guidelines and regulations. The manuscript must include a statement in the Methods identifying the institutional and/or licensing committee approving the experiments.
 - o Statistics: please provide statements on sample size, blinding, randomization and inclusion/exclusion criteria.
- Data Availability section: This section should be placed directly after Methods. Please note that the Data Availability Section is restricted to new primary data that are part of this study, please adjust accordingly. Please also remove "The article and its supplementary information files contain all the additional data supporting the study's findings. The Supporting Data Values file reports the values for every data point represented in the graphs". Please note that the datasets must be made public before acceptance of the manuscript
- Acknowledgements: are there any project numbers to add?
- Author contributions: CRediT has replaced the traditional author contributions section because it offers a systematic machine readable author contributions format that allows for more effective research assessment. Please remove the Authors Contributions from the manuscript and use the free text boxes beneath each contributing author's name in our system to add specific details on the author's contribution. More information is available in our guide to authors.

2/ Figures and Appendix:

- During our standard figure check, we noted anomalies in your figures 4G and 5H: please carefully check the figures composition and provide an explanation if you perform any change. Please also check the associated Source Data.
- Please note that you have the possibility to have up to 5 Expanded View (EV) Figures and Tables that are collapsible/expandable online. EV Figures should be cited as 'Figure EV1, Figure EV2' etc... in the text and their respective legends should be included in the main text after the legends of regular figures. The figures that you do not wish to display as Expanded View figures should remain in the Appendix. If you decide to include EV figures, please make sure to update the callouts in the manuscript text.
- The Appendix pharmacological studies should be added to the main Appendix PDF file.
- The Appendix Table S1 should be renamed Dataset EV1 and a legend should be added to the excel file in a separate tab/worksheet.
- Please make sure that the all figures / figure panels are referenced in the text (currently, callouts are missing for Fig 1G,H; panels should be called out for Fig 8).
- Please address the queries from our copy editors in the figure legends:
 1. Please note that the figure 2g; does not contain any quantification graph, kindly rectify the statistical test related information in the figure legend appropriately.
 2. Please note that the figure 6k; does not contain any quantification graph, kindly rectify the statistics related information in the figure legend appropriately.
 3. Please note that the legends for figures 3l-n is not provided in the sequential manner (legends for figures 3m-n are provided before legend of figure 3l). This needs to be rectified.
 4. Please note that the legends for figures 6b-f is not provided in the sequential manner (legends for figures 6c-f are provided before legend of figure 6b). This needs to be rectified.

5. Please note that the exact p values are not provided in the legends of figures 1b-d, k, n; 2c-e, i-j; 3c, e, g, i, n, p; 5s-t; 6a, c-f, h-i.
6. Please indicate the statistical test used for data analysis in the legends of figures 1b-c; 2k, n; 4j-k.
7. Please note that the box plot needs to be defined in terms of minima, maxima, centre, bounds of box and whiskers, and percentile in the legend of figure 2m.
8. Please note that information related to n is missing in the legends of figures 1b-c, k-m; 2k; 4j-k.
9. Although 'n' is provided, please describe the nature of entity for 'n' in the legends of figures 3e, i; 5p-q.
10. Please note that the error bars are not defined in the legends of figures 1b-c, k-m.
11. Please note that for heatmap present in figure 2l; a numbered scale bar is not provided. This needs to be rectified.
12. Please note that the scale bar needs to be defined for figures 4a-b; 5e-h.
13. Please note that the red arrows are not defined in the legend of figure 3f. This needs to be rectified.

3/ Please check the Source Data Excel Files for Figure 2D and 5D.

4/ Checklist:

- please fill in the section 'Experimental studies/inclusion-exclusion criteria'
- please check that 'Studies involving specimen and field samples' applies to your study

5/ Synopsis:

- Thank you for providing a nice synopsis image. Please resize it to 550 px wide x 300-600 px high. A small (115x70) cropped portion of this image will serve as thumbnail for the table of content on our webpage.
 - Please also provide a synopsis text: it should include a short stand first (maximum of 300 characters, including space) as well as 2-5 one-sentences bullet points that summarizes the paper (maximum of 30 words / bullet point).
- Please attach these in a separate file or send them by email, we will incorporate them accordingly.

6/ As part of the EMBO Publications transparent editorial process initiative (see our Editorial at <http://embomolmed.embopress.org/content/2/9/329>), EMBO Molecular Medicine will publish online a Review Process File (RPF) to accompany accepted manuscripts.

This file will be published in conjunction with your paper and will include the anonymous referee reports, your point-by-point response and all pertinent correspondence relating to the manuscript. Let us know whether you agree with the publication of the RPF.

I look forward to receiving your revised manuscript.

With kind regards,

Lise Roth

***** Reviewer's comments *****

Referee #2 (Comments on Novelty/Model System for Author):

Since the superiority of Apelin-dm compared to existing apelin inhibitors is unclear, the "Medical impact" was rated as "Medium."

Referee #2 (Remarks for Author):

The authors have revised the manuscript and addressed my concerns. I find the study suitable for publication now.

Referee #3 (Comments on Novelty/Model System for Author):

The manuscript has significantly improved and my concerns have all been addressed and answered.

Referee #3 (Remarks for Author):

Thank you for addressing my comments. The manuscript has improved tremendously and should be published without any delays.

The authors addressed the editorial issues.

10th Jan 2025

Dear Dr. Khatib,

Thank you for submitting your revised files, and please accept my apologies for the delayed answer during this busy time of the year.

I have looked at all the files, and I will be ready to accept your manuscript once the following remaining editorial issues are addressed:

1/ Manuscript text:

Methods/Animal studies:

The following information must be provided in the main manuscript file:

- Age, gender, origin of the mice.
- Housing and husbandry conditions.
- Please confirm that all experiments were performed in accordance with relevant guidelines and regulations.
- State details of authority granting ethics approval (IRB or equivalent committee(s)), provide reference number for approval.

In the statistics section, please include a statement on blinding and randomization.

Thank you for correcting the Data Availability section, please note that the data must be available before acceptance.

Please make sure all funding information has been entered in the manuscript and submission system.

Please address the following remaining queries from our copy editors in the figure legends:

- Please note that the legends for figures 3l-n is not provided in the sequential manner (legends for figures 3m-n are provided before legend of figure 3l). This needs to be rectified.
- Please note that the box plot needs to be defined in terms of minima, maxima, centre, bounds of box and whiskers, and percentile in the legend of figure 2m.

2/ Checklist:

Ethics: Last line, specimen and field samples: I do not think this applies to your study, please check and correct if needed.

3/ Synopsis:

Thank you for resizing the synopsis image. Please note that the resolution is not optimal and the text a bit blurry; kindly provide a better resolution image if possible.

I have slightly edited your synopsis text, please let me know if you agree or amend as you see fit:

"Furin is identified as the key enzyme responsible for cleaving the apelin precursor. Inhibiting this cleavage could serve as an effective therapeutic strategy for metastatic colorectal cancer.

- Apelin, an adipokine involved in tumor progression and metastasis, is cleaved from its precursor into the mature form by the proprotein convertase furin.
- Apelin-dm, a modified variant of apelin created by altering proapelin cleavage sites, inhibits tumor growth, induces cell death, suppresses angiogenesis, and delays early colorectal liver metastasis events.
- Proteomic analysis reveals a reciprocal regulatory relationship between apelin and apelin-dm, impacting proteins associated with clinical outcomes in colon cancer patients.
- Apelin-dm modulates apelin receptor behavior, affecting receptor affinity, internalization, and repression of apelin signaling, influencing protein kinase activity.
- Pharmacokinetic assessments confirm that apelin-dm is specific, stable, and undergoes efficient hepatic metabolism, with no significant toxicity."

Please accept all previous changes before resubmission.

I look forward to receiving your revised manuscript.

With my best wishes for the new year,

Lise Roth

Lise Roth, PhD

The authors addressed the remaining editorial issues.

16th Jan 2025

Dear Dr. Khatib,

Thank you for sending your revised files. I am pleased to inform you that your manuscript is accepted for publication and is now being sent to our publisher to be included in the next available issue of EMBO Molecular Medicine.

With kind regards,

Lise Roth
